# A STEP TO DECOUPLE OPTIMIZATION IN 3DGS

**Renjie Ding**[1]   **Yaonan Wang**[1*]   **Min Liu**[1]   **Jialin Zhu**[2]   **Jiazheng Wang**[1]
**Jiahao Zhao**[1]   **Wenting Shen**[1]   **Feixiang He**[3]   **Xiang Chen**[1]

[1]National Engineering Research Center of Robot Visual Perception and
Control Technology, School of Artificial Intelligence and Robitics, Hunan University
[2] Baidu Inc., [3] Central South University
{regi, yaonan, liu_min}@hnu.edu.cn, misaliet@outlook.com
{wjiazheng, zhaojiahao, wenting}@hnu.edu.cn
drfxhe@gmail.com, xiangc@hnu.edu.cn
https://eliottdjay.github.io/adamwgs/

## ABSTRACT

3D Gaussian Splatting (3DGS) has emerged as a powerful technique for real-time novel view synthesis. As an explicit representation optimized through gradient propagation among primitives, optimization widely accepted in deep neural networks (DNNs) is actually adopted in 3DGS, such as synchronous weight updating and Adam with the adaptive gradient. However, considering the physical significance and specific design in 3DGS, there are two overlooked details in the optimization of 3DGS: (i) update step coupling, which induces optimizer state rescaling and costly attribute updates outside the viewpoints, and (ii) gradient coupling in the moment, which may lead to under- or over-effective regularization. Nevertheless, such a complex coupling is under-explored. After revisiting the optimization of 3DGS, we take a step to decouple it and recompose the process into: Sparse Adam, Re-State Regularization and Decoupled Attribute Regularization. Taking a large number of experiments under the 3DGS and 3DGS-MCMC frameworks, our work provides a deeper understanding of these components. Finally, based on the empirical analysis, we re-design the optimization and propose AdamW-GS by re-coupling the beneficial components, under which better optimization efficiency and representation effectiveness are achieved simultaneously.

## 1 INTRODUCTION

Novel view synthesis is a fundamental task in both computer graphics and computer vision. Owing to its highly parallelized design and efficient use of GPUs, 3DGS (Kerbl et al., 2023) leverages explicit primitive-based representations to achieve significantly higher efficiency than implicit representation adopted in NeRF (Mildenhall et al., 2020), while still delivering competitive reconstruction quality. 3DGS has sparked a wave of research and led to rapid extensions across various downstream applications (Matsuki et al., 2024; Chen et al., 2024; Dai et al., 2024).

As a representation directly optimized via backpropagation, 3DGS commonly adopts Adam (Kinga et al., 2015) as its optimizer, following the standard practice in DNN training. By default, the attributes of all primitives are updated simultaneously at each iteration, which we refer to as **synchronous** optimization in this work. Given the viewpoint relationship, an ideal scenario is that the optimization of primitives is guided by gradients from visible viewpoints in different directions. However, upon closer examination, we observe that even when a primitive is invisible from a given viewpoint and receives zero gradients, the **update-step coupling** induced by Adam under synchronous optimization still evolves its optimizer state and can produce effective updates driven by historical moments. From the perspective of optimization efficiency, Mallick et al. (2024) introduce Sparse Adam, which restricts updates to primitives visible from the current viewpoint while excluding invisible primitives with zero gradients, thereby enabling **asynchronous** optimization. Nevertheless, Sparse Adam has not been widely used, as its efficiency gains often come at the cost

---

*Corresponding authors.

of degraded reconstruction performance. We argue that such viewpoint-related optimizer remains promising, and that its current limitation stems from insufficient recognition of update-step coupling.

Furthermore, a previous study (Loshchilov & Hutter, 2017) on DNN optimization points out that the **gradient coupling** in Adam makes the regularization unstable. In current 3DGS, a variety of regularization losses (Yu et al., 2024) have been introduced to improve reconstruction quality or mitigate redundancy[1]. Since regularization gradients are jointly preconditioned with photometric gradients by Adam, a natural question arises as to *whether existing regularization losses are exerting their intended effects*. In practice, regularization losses are typically assumed to be controlled by hyperparameters, yet it remains unclear whether they provide sufficient flexibility. This further motivates the question of whether additional decoupling can enable more precise control over regularization. In this work, we focus on opacity and scaling regularization, which are closely tied to reconstruction quality in the 3DGS-MCMC framework (Kheradmand et al., 2024) and redundancy removal in the vanilla 3DGS framework (Kerbl et al., 2023). Prior studies have shown that the densification stage often produces a large number of redundant primitives (Liu et al., 2025b), while the framework lacks a native mechanism to automatically remove them. Even with opacity regularization, existing pipelines usually rely on additional pruning operations or direct modifications to the densification process. However, *if regularization can be properly decoupled and recomposed, is it possible to achieve redundancy removal without resorting to such extra pruning operations?*

Considering the coupling in 3DGS optimization, this work takes a step to decouple the coupled optimization factors and recomposing them in a better manner. Building on our analysis, we redesign the optimization process to improve efficiency while preserving or enhancing reconstruction quality. To summarize, our contributions are:

- By reanalyzing the complex coupling in 3DGS optimization, we decouple and recompose the optimization process into three effective components: Sparse Adam, Re-State Regularization, and Decoupled Attribute Regularization.
- We further investigate the effects of these components, including the behavior of Sparse Adam, the activation of regularization, the role of regularization hyperparameters, the risks of implicit updates, and the necessity of controllable regularization.
- Based on the experiments and analysis, we propose AdamW-GS with controllable attribute regularization and adopt it into vanilla 3DGS, 3DGS-MCMC and more variants. Experiments show that AdamW-GS improves reconstruction quality and optimization efficiency, while significantly reducing primitive redundancy without introducing additional pruning operations.

## 2 RELATED WORK

**"Coupling" in 3DGS Optimization:** 3DGS commonly adopts the Adam optimizer as in DNNs. However, its synchronous optimization induces overlooked update steps for primitives that are invisible from the current viewpoint. Sparse Adam (Mallick et al., 2024) introduces asynchronous updates to improve efficiency, yet the insufficient recognition of update-step coupling results in degraded performance. Regularization gradients in 3DGS are also coupled with photometric gradients through adaptive preconditioning of Adam, analogous to the coupling between L2 regularization and adaptive gradient preconditioning in DNN optimization. As analyzed in AdamW (Loshchilov & Hutter, 2017), such coupling makes the effective strength of regularization depend on the second-moment estimates, thereby motivating weight decay decoupling. A similar idea is reflected in (Rota Bulò et al., 2025), which, although not explicitly designed for decoupling, replaces opacity reset in vanilla 3DGS with a constant opacity decay during densification, effectively constructing an AdamW-style optimizer. Nevertheless, this approach neglects the physical properties of opacity and the varying significance of individual primitives. The central contribution of this work is to further decouple and recompose 3DGS optimization, thereby reinterpreting primitive redundancy in vanilla 3DGS and reconstruction quality in 3DGS-MCMC from an optimization perspective.

**Redundancy during Optimization:** 3DGS typically requires careful treatment of primitive generation and pruning (or death (Kheradmand et al., 2024; Zhu et al., 2025)). Current studies have observed that optimization often produces abundant redundant primitives, while pruning in adaptive density control (Kerbl et al., 2023) is insufficient to remove them. Four main strategies have been

---

[1]Redundancy is strictly defined as minimizing active primitives without sacrificing reconstruction quality.

proposed to address this redundancy: (1) Densification refinement (Fang & Wang, 2024; Mallick et al., 2024; Wang et al., 2025); (2) Hand-crafted criteria (Fan et al., 2024; Niemeyer et al., 2025; Hanson et al., 2025); (3) Learning-based pruning (Lee et al., 2024; Liu et al., 2025b; Zhang et al., 2024a). More details can be found in Sec. H. (4) Optimization-based operations: Optimization together with additional opacity L1 regularization has become a common technique for redundancy removal, even extending beyond Efficient 3DGS tasks (Papantonakis et al., 2024; Lee et al., 2024; Kheradmand et al., 2024; Liu et al., 2025a; Svitov et al., 2024).

## 3 PRELIMINARY: 3DGS(-MCMC) AND ADAM

**3DGS** approximates the radiance field of a target scene using a set of Gaussian primitives parameterized by location, opacity ($o$), scale ($s$), and other attributes, and renders images from given viewpoints via alpha blending. Further details are provided in Sec. D of the Appendix.

Typically, the training pipeline can be divided into **densification** and **pure optimization** (**P-Op**). During densification, new primitives are generated to enhance scene representation while redundant ones are pruned (low-opacity primitives are removed (Kerbl et al., 2023)), or new primitives are sampled from existing primitives while dead primitives are reallocated to new locations (Kheradmand et al., 2024). After densification, there is a P-Op stage, during which only gradient propagation and attribute updates occur. In this training pipeline, the photometric loss term in Eq.1 is adopted with the Adam optimizer. For 3DGS-MCMC, extra regularization losses are used to promote respawning.

$$\mathcal{L} = \underbrace{(1 - \lambda_1)\mathcal{L}_1 + \lambda_1\mathcal{L}_{DSSIM}}_{\text{photometric loss: } \ell} + \underbrace{\lambda_o|o|_1 + \lambda_s|s|_1}_{\text{regularization loss:} \mathcal{R}} \tag{1}$$

In this work, $\ell$ and $\mathcal{R}$ denote photometric loss and any regularization, while $\nabla\ell$ and $\nabla\mathcal{R}$ are the corresponding gradient.

**Adam** is a first-order stochastic optimization method that adaptively estimates moment statistics of the gradients. Specifically, it maintains exponential moving averages of the first- and second-moment estimates, $m_t(\theta)$ and $v_t(\theta)$, which are bias-corrected to $\hat{m}(\theta)_t$ and $\hat{v}(\theta)_t$. The parameter update is then performed as Eq. 3, where $\eta$ denotes the learning rate and $\epsilon$ ensures numerical stability.

$$m(\theta)_t = \beta_1 \times m(\theta)_{t-1} + (1 - \beta_1) \times g(\theta)_t \quad v(\theta)_t = \beta_2 \times v(\theta)_{t-1} + (1 - \beta_2) \times g(\theta)_t^2 \tag{2}$$

$$\theta_{t+1} = \theta_t - \eta \times \frac{\hat{m}(\theta)_t}{\sqrt{\hat{v}(\theta)_t} + \epsilon} \tag{3}$$

We omit the expression of bias correction procedure. In this paper, $\theta$ and $g$ denote an arbitrary primitive attribute and the gradient information respectively.

As in DNN training (Goodfellow et al., 2016), Adam by default updates all attributes synchronously, including both primitives visible from the current viewpoint and those that are not. Even when invisible primitives receive zero gradients, they are still included in the update step, during which their moments are rescaled and attributes are still updated. In this paper, we refer to this phenomenon as **Implicit Update**, which is induced by historical optimizer states.

## 4 METHODOLOGY

### 4.1 CHARACTERISTICS OF SPARSE ADAM

As discussed in Sec. 1, using **Sparse Adam** directly improves optimization efficiency but causes degraded performance. Let $\mathcal{V}$ denote a visibility filter that takes value 1 for primitives visible from the current viewpoint and 0 otherwise. Sparse Adam can be obtained by replacing $\beta$ in Eq. 2 via:

$$\beta' = \beta \times \mathcal{V} + (1 - \mathcal{V}) \tag{4}$$

Table 1: Quantitative results on MipNerf360 for 3DGS with different optimizers. The definitions of the metrics can be found in Sec.5. (m: million)

| Cite | PSNR | SSIM | LPIPS | $N_p$/m | $N_d$/m |
|------|------|------|-------|---------|---------|
| GS1 | 27.507 | 0.815 | 0.216 | 3.331 | 0.232 |
| GS2 | 27.285 | 0.809 | 0.228 | 2.532 | 0.039 |
| GS3 | 27.567 | 0.816 | 0.216 | 3.342 | 0.048 |

However, from the perspective of inter-viewpoint updates, Sparse Adam is effectively viewpoint-stable: the updates of primitives are influenced only by steps from visible viewpoints and does not undergo implicit updates from invisible viewpoints.

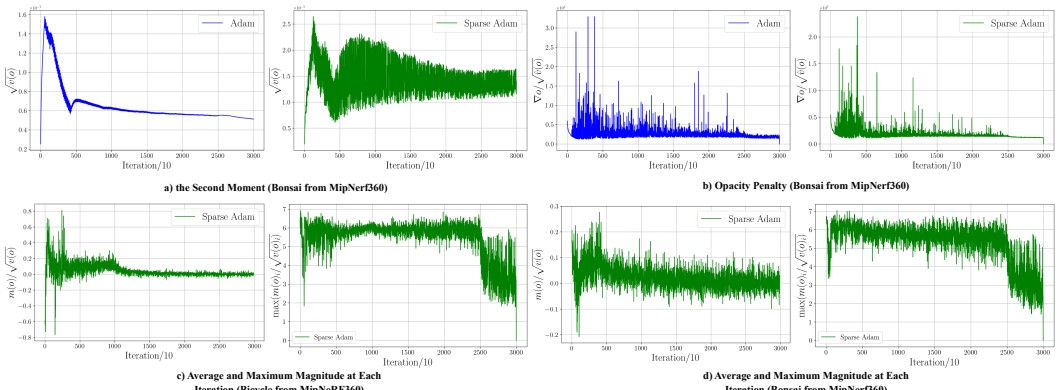

Figure 1: **a**: The $\sqrt{v(o)}$ in 3DGS-MCMC with different optimizers. More examples can be found in Appendix Figure 5. **b**: The opacity regularization decisive term in 3DGS-MCMC with different optimizers. **c-d**: The average and maximum magnitudes of $m(o)/\sqrt{v(o)}$ at every iteration.

To conduct a preliminary investigation, three experiments are performed on vanilla 3DGS: (GS1) training with the original Adam optimizer, (GS2) replacing Adam with Sparse Adam, and (GS3) employing Adam during the densification stage while applying Sparse Adam in the P-Op stage. Based on the comparisons in Table 1, we make the following observations: **Observation 1** *Sparse Adam is more stable*: Since densification is governed by gradient magnitudes, the smaller number of primitives observed under Sparse Adam suggests that more primitives fail to meet the gradient threshold. Moreover, Sparse Adam retains more active primitives, especially in GS3 of Table 1 (0.048 million dead primitives with Sparse Adam *vs.* 0.232 million with Adam), which implies that certain components in Adam are potential for pruning redundant primitives. **Observation 2** *Sparse Adam is less explorative*: Although Sparse Adam quickly drives primitives toward stability and results in fewer primitives overall, its performance degrades noticeably. Experiments on 3DGS-MCMC show similar observations in some scenes, where Sparse Adam leads to fewer reallocated primitives, as shown in Figure 3 f.

## 4.2 Update-step decoupling: Re-State Regularization

The key question is which components differentiate Adam from Sparse Adam. To answer this, we revisit synchronous optimization with Adam introduced in Sec. 3. The overlooked update-step coupling arises as zero-gradient update steps rescale the moments ($m(\theta)_t = \beta_1 m(\theta)_{t-1}$, $v(\theta)_t = \beta_2 v(\theta)_{t-1}$) and subsequently induce implicit updates based on the rescaled moments. This first implies that the optimizer states of primitives invisible from the current viewpoint continue to evolve. As visualized in Figure 1 a, the second moment in 3DGS-MCMC with Adam is notably smaller than that with Sparse Adam, suggesting a potential amplification of the first moment. To verify this, we compare the effective strength of opacity regularization in Figure 1 b, $\nabla o/\sqrt{\hat{v}(o)}$, and observe that Adam induces stronger regularization, consistent with our discussion in Sec. 4.1. *Moment rescaling facilitates the activation of regularization*. The comparison in Figure 1 a also shows that the magnitude remains consistently high for Sparse Adam, underscoring the importance of moment rescaling: *when inappropriate gradients accumulate from a specific viewpoint, they are difficult to dissipate, thereby hindering optimization*. The role of implicit updates is direct: primitives invisible from the current viewpoint are continuously updated based on the past moments. However, no prior work has explicitly examined the impact of these components on 3DGS optimization. Motivated by this, we construct two decoupled variants for Sparse Adam.

The implementation of moment rescaling is straightforward. We define an optimization interval and introduce a milestone style **State Sampling Schedule** (**StSS**), which uniformly samples the existing primitives at fixed intervals. The sampled primitives are then processed as in Eq. 5 to deliberately attenuate their optimizer states. This process is referred to as **Re-State Regularization** (**RSR**).

$$m(\theta)_t^{\text{new}} = \alpha_1 \times m(\theta)_t^{\text{old}} \quad v(\theta)_t^{\text{new}} = \alpha_2 \times v(\theta)_t^{\text{old}} \quad 0 \leq \alpha_1 < 1, \ 0 \leq \alpha_2 < 1 \qquad (5)$$

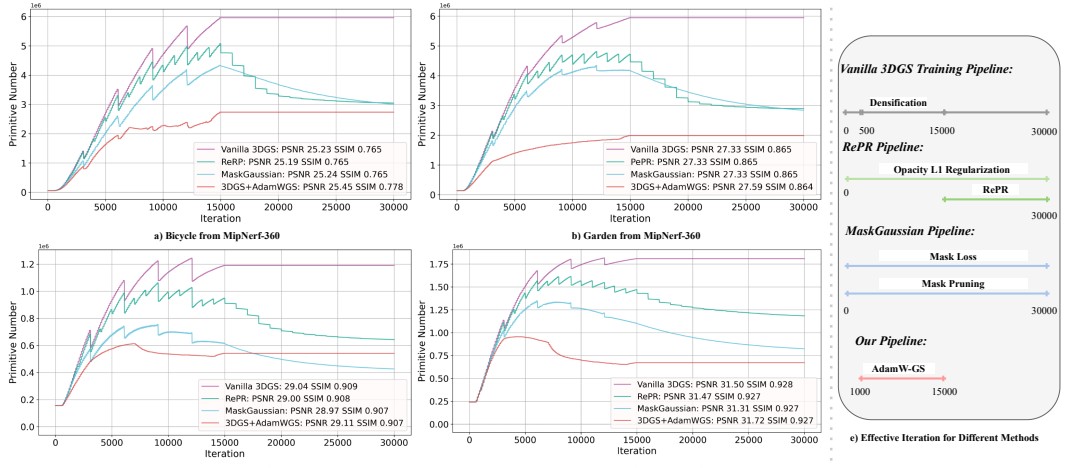

Figure 2: **a-d**: Changes in the number of primitives during training for four methods: vanilla 3DGS, Redundant Primitives Removal (RePR), MaskGaussian, and 3DGS with our proposed AdamW-GS. **e**: Iteration ranges over which different components affect the primitive number.

Here, the subscript $t$ follows an asynchronous indexing scheme rather than the synchronous step in Eq. 2, meaning that each primitive has its own distinct $t$. A detailed discussion of hyperparameter selection is provided in Sec. J.2.

We also construct an artificial implicit update variant to study its role. While implicit updates can be beneficial in some cases, they may introduce negative effects, potentially linked to primitive redundancy (see Appendix Sec. I.4). In the experiment section in the main text, we use implicit updates only as a tool to amplify attribute regularization under Sparse Adam, so as to study the influence of regularization: when $\lambda \nabla \mathcal{R}(\theta)/N_v$ in Eq. 6 is large, maintaining implicit updates further strengthens regularization. Concretely, our **Artificial Implicit Update (AIU)** is implemented as a uniform sampler that selects a random subset of invisible primitives for extra updates, while keeping their moments unchanged when they are invisible from the current viewpoint, as in Sparse Adam.

$$m(\theta)'_t = \beta'_1 m(\theta)'_{t-1} + (1 - \beta'_1)(\frac{\nabla \ell(\theta)}{N_I} + \frac{\lambda \nabla \mathcal{R}(\theta)}{N_v}) \tag{6}$$

where $N_I$ and the $N_v$ are the scale of image pixels and the primitive number visible from the current viewpoint, respectively, following the implementation of the codebase[2].

### 4.3 GRADIENT DECOUPLING: DECOUPLED ATTRIBUTE REGULARIZATION

We view attribute regularization as an essential component of optimization. In efficient 3DGS and related tasks, the widely used opacity L1 regularization exemplifies this idea, analogous to weight regularization in DNNs, though with a different role. As discussed in Sec. 3, our understanding of how regularization constrains individual primitives remains limited. Implicit updates may further magnify the effect of regularization: when the magnitude of $\lambda \nabla \mathcal{R}(\theta)/N_v$ in Eq. 6 approaches or exceeds that of $\nabla \ell(\theta)/N_I$, the regularization term continues to influence the optimization via the adaptive gradient or implicit updates. The coupling between photometric and regularization losses in Eq. 6 therefore raises a natural question: *is the current regularization over- or under-effective?*

First, we revisit the factors that actually govern the effect of the current attribute regularization loss, *i.e.*, L1 loss under gradient coupling. The hyperparameters maintain the balance between different loss terms, ensuring that the regularization remains relatively small during sensitive optimization stages without interfering with the photometric loss. Relevant results are presented in Appendix Sec. I.1. Regarding the modulation by optimizer moments, the regularization is activated when $\lambda \nabla \mathcal{R}(\theta)/N_v$ becomes dominant or when the moment is rescaled during synchronous optimization in Adam or reset in 3DGS-MCMC.

$$m(\theta)'_t = \beta'_1 \times m(\theta)'_{t-1} + (1 - \beta'_1) \times \frac{\nabla \ell(\theta)}{N_I} \quad v(\theta)'_t = \beta'_2 \times v(\theta)'_{t-1} + (1 - \beta'_2) \times (\frac{\nabla \ell(\theta)}{N_I})^2 \tag{7}$$

---

[2]https://github.com/DerThomy/3dgs-mcmc

Using Sparse Adam or Adam in 3DGS-MCMC alters the effect of attribute regularization. As illustrated in Figure 3 e–f, different optimizers yield different numbers of reallocated primitives and different PSNR values, indicating over- or under-effective regularization. This suggests that stronger regularization may be required to improve reconstruction quality in some cases. However, due to the gradient coupling in the optimizer moments (Eq. 6), directly amplifying $\nabla \mathcal{R}$ can easily lead to update steps that become overly attribute-dependent and less guided by the photometric gradient, while the coupling also sustains the effect of regularization across iterations. In extreme cases, such as when hyperparameters are scaled by $10\times$, optimization fails entirely. When amplification relies on the second moment, the effect of regularization depends on its scale relative to $\nabla \ell$. Furthermore, if $\nabla \mathcal{R}$ aligns with $\nabla \ell$, the second moment grows rapidly; if they are opposed, it grows slowly, which is contrary to expectations. This coupling prevents effective control of regularization. As shown in Figure 1 c,d, the order-of-magnitude gap between the average and maximum $m(o)/\sqrt{v(o)}$ further illustrates that improper amplification under coupling can easily result in over-effective regularization. Decoupling $\nabla \ell$ and $\nabla \mathcal{R}$ and recomposing them enables more reliable control without destabilizing optimization. Further experiments in Sec. 5 also show the side effects of coupling.

Therefore, the first step is to decouple the gradients in the moments in Eq. 6. This step is analogous to AdamW and readily yields Eq. 7, which defines a moment update term containing only the gradient of the photometric loss $\nabla \ell$. For $\nabla \mathcal{R}(\theta)$, AdamW constructs updates that are not controlled by the second moment. This **AdamW-style decoupling** is effectively equivalent to a constant penalty on opacity (Rota Bulò et al., 2025). The experiments are presented in Sec. I.2. Clearly, this approach is not optimal. Unlike DNNs, each attribute in 3DGS has physical meaning, and each primitive carries a distinct importance. Penalizing all primitives with the same strength fails to remove redundant primitives when the penalty is too weak, while overly strong penalties hinder optimization.

AdamW-style decoupling suggests that effective regularization should assign different penalties to different primitives. Such assignment, however, cannot be arbitrary: (i) it must not interfere with normal optimization, particularly in under-constructed regions where $\nabla \ell$ is large; (ii) it must still fulfill its core function of suppressing overfitting when $\nabla \ell$ is small (Srivastava et al., 2014). This implies that regularization should adapt to the geometry of the data (Pascanu & Bengio, 2013), naturally linking back to adaptive gradient (Duchi et al., 2011; Kinga et al., 2015) where adjustments are made via $\sqrt{v}$. The empirical success of vanilla regularization compared with the constant penalty further highlights its potential. Since we have already disentangled the moment in Eq. 7, $\sqrt{\hat{v}}$ now provides a more faithful estimate of the optimization geometry. Motivated by this, we introduce the regularization form $\nabla \mathcal{R}/\sqrt{\hat{v}}$. This design offers four benefits: (1) updates are preserved in under-constructed regions with large $\nabla \ell$; (2) when a primitive lies near a saddle point, regularization facilitates escape, consistent with saddle-point analyses in (Wang et al., 2025) and long-axis primitive gradients in (Zhang et al., 2024b); (3) regularization remains small in general but adaptively amplified when needed; (4) it enables explicit control over regularization via moment modulation, e.g., through RSR. Building on this rationale, we extend this approach and propose Eq. 8:

$$\theta_{t+1} = \theta_t - \eta \times [\frac{\hat{m}(\theta)'_t}{\sqrt{\hat{v}(\theta)'_t} + \epsilon} + \min(\lambda_\theta \frac{\nabla \mathcal{R}(\theta)/N_I}{\sqrt{\hat{v}(\theta)'_t} + \epsilon}, \mathcal{C}_t)] \tag{8}$$

We balance the loss terms with $\lambda_\theta$ and normalize the regularization scale by $N_I$. Regularization preconditioned by $\sqrt{\hat{v}(\theta)'_t}$ improves stability while removing the dependence on $N_v$, which can vary by orders of magnitude across training, views, and scenes. $\mathcal{C}_t$ remains on the same order as the maximum step size of Adam and matches the max values observed in Figure 1 c–d, while extensive results confirm its robustness. This recomposed regularization is referred to as **Decoupled Attribute Regularization** (**DAR**). Detailed hyperparameter selection is provided in Appendix Sec. J.2.

## 4.4 RECOUPLING: ADAMW-GS

Building on Sec. 4.1, 4.2, and 4.3, we propose **AdamW-GS** for better 3DGS optimization by recoupling Sparse Adam, RSR, and DAR. In detail, primitive attributes are optimized asynchronously with Sparse Adam, guided by photometric loss and DAR. At fixed training intervals, RSR uniformly samples primitives and rescales their moments via Eq. 5 to better activate regularization. The rescaled moments replace the previous estimates and directly participate in subsequent optimization.

We consider two DAR variants—opacity and scaling regularization—both of which are already employed in 3DGS-MCMC. Considering the limited transport in vanilla 3DGS (Jung et al., 2024),

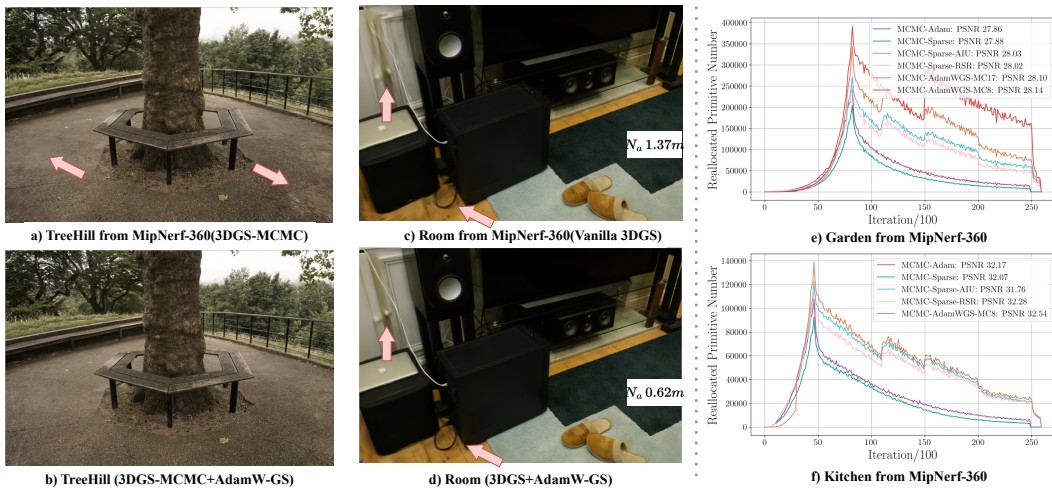

Figure 3: **a-d**: Reconstruction results visualization. More can be found in Appendix Sec.K. **e-f**: The Reallocated Primitive Number in 3DGS-MCMC Framework. For outdoor scenes, MC17 and MC8 differ only in the StSS sampling ratio, where MC8(StSSMC3)>MC17(StSSMC1)=MCMC-Sparse-RSR. For indoor scenes, MC8 uses StSSMC1. More information can be checked in Table 2.

**exploration** here denotes improved primitive mobility during optimization. We directly use the noise regularization from 3DGS-MCMC as an additional position regularization, which is similar to DAR but controlled by opacity and primitive shape rather than the second moment. However, due to the sensitivity to noise (Jung et al., 2024), the noise term is excluded for MipNerf360 indoor scenes with fewer primitives. To evaluate the effectiveness of exploration, we provide comparative experiments with and without opacity reset (Kerbl et al., 2023), which encourages exploration by lowering the opacity. Opacity Correction is adopted for cloning during densification, which is thought to be necessary. We adopt opacity operations similar to those in (Rota Bulò et al., 2025).

## 5 EXPERIMENTS

**Dataset and Metric** Following existing research (Kerbl et al., 2023), we employ 13 scenes from 3 datasets, including nine scenes from Mip-Nerf360 (Barron et al., 2022), two from Tanks&Temples dataset (Knapitsch et al., 2017) and two provided by Hedman et al. (Hedman et al., 2018).

We follow the train/test split suggested by Mip-NeRF 360, taking every 8th photo for testing. This protocol enables consistent and meaningful comparisons under standard reconstruction metrics, including PSNR, SSIM, and LPIPS(VGG), which are widely used in the literature. We also present the total primitive number $N_p$, the active primitives number $N_a$ and the dead primitives number $N_d$. Given the significant variation in primitive number across methods, often differing by millions, we employ normalized changes relative to the vanilla ($\Delta N_x = \frac{N_x - N_x^{\text{vanilla}}}{N_x^{\text{vanilla}}}$, where $x \in \{a, p, d\}$) to better illustrate primitive number variations. A primitive is defined as active if its opacity is larger than $\frac{1}{255}$, and as dead otherwise. This definition, differing from (Kheradmand et al., 2024), follows the 3DGS[3] implementation, where primitives with opacity below $\frac{1}{255}$ are excluded from rendering.

**Baselines** To evaluate the impact of different components on the training pipeline, we conduct extensive experiments under both the vanilla 3DGS and 3DGS-MCMC frameworks. These two baselines respectively represent approaches without and with an $N_p$ maximum. Without explicitly introducing pruning operations, we observe that a substantial number of redundant primitives can be automatically removed in the vanilla 3DGS with AdamW-GS. For comparison, we include 2 adaptive pruning methods, the learning-based MaskGaussian(Liu et al., 2025b) and Redundant Primitive Removal (RePR) with the hand-crafted criterion in (Papantonakis et al., 2024). Unlike methods with a predefined pruning rate, these methods are reported to identify redundant primitives automatically with negligible performance degradation. *All methods are run on a single A6000 GPU.*

---

[3]https://github.com/graphdeco-inria/gaussian-splatting

Table 2: Quantitative results on MipNerf-360 with different components. Detailed descriptions of Sparse Adam, AIU, RSR, and DAR are provided in Sec. 4.1, Sec. 4.2, and Sec. 4.3, respectively. All RSR and DAR settings remain fixed across experiments, except for the StSS schedule. The StSS sampling ratios used in this table are as follows: for outdoor scenes, MC8 (StSSMC3) > MC7 (StSSMC2) > others (StSSMC1), while for indoor scenes, MC8 = MC7 = others (StSSMC1). The complete StSS schedules for each configuration are illustrated in Figure 9. Appendix Sec. K provides per-scene experimental results, including detailed configurations and additional experiments with a broader range of settings.

|  | Cite | Sparse Adam | AIU | RSR | $\mathcal{R}_o$ | $\mathcal{R}_s$ | PSNR | SSIM | LPIPS | $\Delta N_a$ | $N_p$/m |
|---|---|---|---|---|---|---|---|---|---|---|---|
|  | MC1 | x | x | x | L1 | L1 | 27.948 | 0.833 | 0.199 | -3.75% | 3.313 |
|  | MC2 | ✓ | x | x | L1 | L1 | 27.998 | 0.832 | 0.199 | +4.28% | 3.313 |
|  | MC3 | ✓ | ✓ | x | L1 | L1 | 28.050 | 0.833 | 0.198 | +3.62% | 3.313 |
| 3DGS-MCMC | MC4 | ✓ | ✓ | ✓ | L1 | L1 | 28.017 | 0.834 | 0.191 | +0.51% | 3.313 |
|  | MC19 | ✓ | x | ✓ | L1 | L1 | 28.075 | 0.837 | 0.190 | +2.97% | 3.313 |
|  | MC7 | ✓ | x | ✓ | DAR | DAR | 28.185 | 0.839 | 0.182 | +4.72% | 3.313 |
|  | MC8 | ✓ | x | ✓ | DAR | DAR | 28.219 | 0.840 | 0.182 | +4.52% | 3.313 |

## 5.1 RESULTS AND ANALYSIS

**Over- or Under-Effective Regularization** To examine the controllability of regularization, we introduce AIU during densification as a direct tool to amplify the effect of regularization. Its role is straightforward: when the moment of a primitive places it under the regularization, AIU preserves the corresponding update, yielding more dead primitives and thus strengthening regularization. Figure 3 e–f visualizes changes in reallocated primitives in 3DGS-MCMC with AIU. Enhanced regularization promotes exploration and improves reconstruction quality in some cases (MC3 and MC4 in Table 2), but the effect is scene-dependent. For instance, in Kitchen, as shown in Figure 3 f, the rapid growth of reallocated primitives degrades performance. This suggests that regularization can be **under-effective** in some scenes, requiring amplification, e.g., via AIU or other methods, but can also be **over-effective** when its strength is excessive or inapposite.

**Activated via the Moment** Within the 3DGS-MCMC framework, we evaluate the effect of our proposed RSR on regularization, *a way of activating regularization via rescaling the moment*. Whether using the original L1 form with gradient coupling or our proposed DAR, attribute regularization remains tied to the second moment. Figure 3 e–f shows that applying RSR substantially increases the number of dead primitives, thereby strengthening attribute regularization. This directly demonstrates the effectiveness of RSR as a component for amplifying regularization. However, as highlighted by the comparison between MCMC-Sparse-RSR and MCMC-AdamWGS-MC17 in Figure 3 e, the influence of RSR with the same StSS is weaker under L1 regularization with gradient coupling than under DAR. By decoupling the gradient in the moment and recomposing the attribute regularization, RSR more effectively amplifies the regularization effect, which further translates into improved reconstruction quality under stronger regularization.

**Side Effect in Coupling** To further examine this effect, *we increase the sampling ratio of RSR* in MidNeRF-360 indoor scenes, which typically involve fewer primitives. The results are summarized in Appendix, from Sec. K.6 to Sec. K.9, as MC20 and MC21. Specifically, we replace the low-ratio StSSMC1 in both the original 3DGS-MCMC with L1 regularization and 3DGSMCMC with our proposed AdamW-GS by the higher-ratio StSSMC2. As the sampling ratio increases, original 3DGS-MCMC with L1 regularization exhibits significant drops in reconstruction quality across all four scenes, e.g., Room: 32.514→31.179 dB; Kitchen: 32.289→31.924 dB. The same phenomenon is also observed in the experiments on outdoor scenes. In contrast, 3DGS-MCMC with DAR shows little to no degradation in Room, Counter, and Bonsai, and only a minor drop in Kitchen (32.546→32.298 dB), while consistently outperforming L1 regularization under the same settings. These results confirm that *gradient coupling hinders effective control of regularization and can easily introduce negative effects on reconstruction quality*. Thus, decoupling the gradient from the moment and recomposing regularization are necessary for stable and effective optimization.

Experiments with RSR also highlight the issue of over- or under-effective regularization. For scenes with more primitives, such as MipNeRF360 outdoor scenes, we adopt a higher sampling ratio for RSR. As the ratio increases, the number of reallocated primitives grows further, as shown in Figure 3 e, strengthening regularization and enhancing the exploration capability of 3DGS-MCMC. A

Table 3: Quantitative results of different methods on MipNerf360. MC8 and GS8/GS7 denote our proposed AdamW-GS variants. More information of MC8 is provided in Table 2. All variants share the same hyperparameters except for the StSS schedule. Following the design used in 3DGS-MCMC, outdoor scenes for vanilla 3DGS use a high-ratio StSS, while indoor scenes use a low-ratio StSS. As discussed in Sec. 4.4, GS7 is the noise without opacity reset version to study the effectiveness of exploration. A per-scene organization of results, including detailed configurations and additional experiments, is presented in Sec. K.

| Methods | All | | | | | Outdoor | | | | Indoor | | | |
|---|---|---|---|---|---|---|---|---|---|---|---|---|---|
| | PSNR↑ | SSIM↑ | LPIPS↓ | $\Delta N_a$ | time/mins↓ | PSNR↑ | SSIM↑ | LPIPS↓ | $\Delta N_a$ | PSNR↑ | SSIM↑ | LPIPS↓ | $\Delta N_a$ |
| Original MCMC | 27.948 | 0.833 | 0.199 | -3.75% | 46.81 | 25.105 | 0.755 | 0.212 | -4.38% | 31.502 | 0.930 | 0.182 | -2.97% |
| +AdamW-GS(MC8) | **28.219** | **0.840** | **0.182** | +4.52% | **39.77** | **25.247** | **0.764** | **0.191** | +3.07% | **31.934** | **0.935** | **0.172** | +6.33% |
| vanilla 3DGS | 27.506 | 0.815 | 0.216 | (3.098m) | 30.58 | 24.648 | 0.728 | 0.239 | (4.512m) | 31.080 | **0.925** | 0.189 | (1.331m) |
| +AdamW-GS(GS8) | 27.678 | **0.822** | 0.220 | 49.3% | **18.53** | 24.854 | **0.740** | 0.243 | **-48.4%** | 31.209 | 0.925 | 0.191 | 50.4% |
| +AdamW-GS(GS7) | **27.730** | 0.820 | 0.222 | -46.9% | 19.18 | **24.949** | 0.737 | 0.248 | -44.0% | | | | |
| RePR | 27.503 | 0.815 | 0.218 | -41.1% | 26.73 | 24.661 | 0.728 | 0.241 | -41.5% | 31.055 | 0.924 | 0.190 | -40.6% |
| MaskGaussian | 27.485 | 0.815 | 0.219 | **-53.1%** | 26.80 | 24.683 | 0.728 | 0.240 | 46.4% | 30.988 | 0.924 | 0.192 | **-61.5%** |

direct comparison between MC7 and MC8 shows that higher sampling ratios improve reconstruction quality. This further demonstrates the effectiveness of RSR in controlling regularization.

**AdamW-GS in 3DGS-MCMC** With properly tuned hyperparameters, we evaluate our proposed AdamW-GS within the 3DGS-MCMC framework. As shown in Table 3, our approach outperforms original 3DGS-MCMC in terms of PSNR, SSIM, and LPIPS. By decoupling the gradient coupling and recomposing regularization, AdamW-GS yields more stable regularization. Meanwhile, by decoupling update-step coupling, the proposed RSR provides finer control of regularization. Furthermore, eliminating numerous ineffective zero-gradient updates via Sparse Adam improves overall optimization efficiency.

**Robust Hyperparameter Test and Autonomously Redundancy Removal** Given that AdamW-GS, equipped with DAR and RSR, introduces many hyperparameters, we test their robustness by directly applying it to vanilla 3DGS with the same hyperparameters. According to the training division of vanilla 3DGS, we design several StSS configurations, as shown in Figure 9. Similar to our observations in 3DGS-MCMC with AdamW-GS, where moment rescaling and opacity DAR encourage a large number of dead primitives for exploration, our experiments show that *AdamW-GS automatically removes a large number of redundant primitives in vanilla 3DGS without introducing additional pruning components, while the improved reconstruction quality further indicates enhanced exploration* (Observation 2 issue in Sparse Adam in Sec.4.1).

We compare vanilla 3DGS optimized with AdamW-GS against two adaptive pruning methods. Importantly, our approach contains no extra pruning component; instead, it relies solely on opacity DAR. In terms of primitive reduction, AdamW-GS achieves results comparable to MaskGaussian and even outperforms it by 2% on outdoor scenes. Beyond pruning efficiency, our method also improves reconstruction quality: in outdoor scenes, it reduces the number of primitives by 48.4% while increasing PSNR by 0.2 dB and SSIM by 0.01. In indoor scenes, it removes 50% of primitives while still improving PSNR by 0.1 dB. By contrast, MaskGaussian removes 61% of primitives but degrades PSNR by 0.1 dB, suggesting a potential risk of reconstruction-quality degradation, as shown in Figure 2 c. The visualization in Figure 3 c-d also demonstrate better detail reconstruction by our methods compared with vanilla 3DGS. This partially explains why our method retains more primitives than MaskGaussian. Ultimately, our objective is not simply to minimize the number of active primitives, but to strike a preferable balance—*achieving higher reconstruction quality with as few primitives as necessary.*

The proposed method is further evaluated on additional datasets. As shown in Table 4, experiments on Deep Blending and Tanks&Temples further demonstrate the superiority of our approach. In particular, on Deep Blending, 3DGS equipped with AdamW-GS even achieves higher PSNR than the original 3DGS-MCMC.

Table 4: Quantitative results on Deep Blending and Tank & Temples. (m: million.)

| | Deep blending | | | | Tank & Temples | | | |
|---|---|---|---|---|---|---|---|---|
| | PSNR↑ | SSIM↑ | LPIPS↓ | $\Delta N_a$ | PSNR↑ | SSIM↑ | LPIPS↓ | $\Delta N_a$ |
| 3DGSMCMC | 30.089 | 0.914 | 0.239 | -24% | 24.563 | 0.869 | 0.160 | -3.1% |
| +AdamW-GS | **30.417** | **0.916** | **0.228** | 2.8% | **24.726** | **0.875** | **0.150** | 6.7% |
| 3DGS | 29.694 | 0.904 | 0.247 | 2.60m | 23.677 | 0.848 | 0.178 | 1.60m |
| +AdamW-GS | **30.260** | **0.912** | **0.245** | -60% | **24.303** | **0.855** | 0.181 | -40% |
| MaskGaussian | 29.895 | 0.908 | 0.248 | -65% | 23.607 | 0.846 | 0.181 | -50% |

Appendix Sec. E.4 further reports results on long-sequence datasets for vanilla 3DGS and 3DGS-

MCMC, both with and without the proposed AdamW-GS, demonstrating that AdamW-GS continues to provide clear advantages in these settings.

Table 5: Quantitative results of MaskGaussian.

| | Indoor | | | | Outdoor | | | |
|---|---|---|---|---|---|---|---|---|
| | PSNR | SSIM | LPIPS | $\Delta N_a$ | PSNR | SSIM | LPIPS | $\Delta N_a$ |
| MaskGaussian | 30.988 | 0.924 | 0.192 | -61.5% | 24.683 | 0.728 | 0.240 | -46.4% |
| +AdamW-GS | **31.199** | **0.925** | 0.193 | **-68.6%** | **24.939** | **0.739** | 0.244 | **-48.1%** |

Beyond dataset-level evaluation, we examine the compatibility of AdamW-GS with additional pipelines. As shown in Table 3, MaskGaussian exhibits a potential reconstruction-quality risk on indoor scenes, consistent with Observation 1 from the Adam-vs.-Sparse-Adam experiments in Sec. 4.1. We attribute this issue to the *synchronous updating of mask scores in MaskGaussian*; MaskGaussian is briefly introduced in Sec. E. Results in Table 5 further support this hypothesis: *when optimized with AdamW-GS, MaskGaussian no longer suffers from this quality risk and prunes about 7% more redundant primitives on indoor scenes*. This also demonstrates that *AdamW-GS is compatible with additional pruning operations*. In Appendix Sec. E, we provide results for more pipeline variants with AdamW-GS, further confirming its robustness.

To better understand how our method penalizes redundant primitives, we visualize the dynamics of primitive number during training, as shown in Figure 2. PePR, which relies solely on opacity L1 regularization during densification, shows limited effectiveness in reducing redundancy. In contrast, under our proposed regularization, primitive number remains consistently lower. Even when final $\Delta N_a$ is larger than that of MaskGaussian, our method, i.e., vanilla 3DGS optimized with AdamW-GS, still outperforms MaskGaussian in reducing redundancy during densification.

**Extra Exploration is Necessary** Previous studies have highlighted the role of exploration in 3DGS (Jung et al., 2024; Kheradmand et al., 2024), and here we investigate its interplay with redundancy. AdamW-GS benefits from the noise-based position regularization, yielding improved pruning efficiency and higher SSIM. As shown in GS7 and GS8 of Table 2, removing opacity reset weakens the noise-based regularization, leading to smaller SSIM gains and weaker redundancy removal. We attribute the effectiveness of the noise-based regularization in AdamW-GS to its more appropriate attribute regularization. Overall, our results underscore that additional exploration is necessary. However, the noise-based position regularization from 3DGS-MCMC cannot be directly applied to scenes with relatively few primitives, e.g., indoor scenes, where the smaller primitive set may make optimization more sensitive to noise and can lead to degraded reconstruction quality. In Appendix Sec. F, we provide a more detailed analysis of the exploration ability induced by our method. We also introduce two additional exploration strategies that can be combined with the current AdamW-GS and yield further improvements on certain indoor scenes.

**Efficiency Analysis** As shown in Table 3, 3DGS-MCMC achieves further optimization efficiency gains from Sparse Adam. Vanilla 3DGS enjoys the same benefit, while our method achieves additional gains by substantially reducing redundancy during the densification stage, thereby further enhancing optimization efficiency. More detailed analysis is provided in Appendix Sec. C.

## 6 DISCUSSION AND CONCLUSION

This work takes a step to decouple the complex 3DGS optimization and analyzes its key mechanisms. AdamW-GS is then proposed, which improves efficiency and enables more controllable regularization. The method enhances reconstruction quality within the 3DGS-MCMC framework and achieves notable redundancy removal in vanilla 3DGS, with slight quality gains and even without pruning. Our results indicate that efficiency and effectiveness can be jointly improved with appropriate regularization. While this work focuses on opacity and scaling, more general DAR variants for other attributes or alternative strategies may yield further improvements. Although the current hyperparameters and hand-crafted StSS configurations are robust, adaptive parameter adjustment during training remains a promising direction. Finally, although noise-driven exploration shows necessary, it can harm performance in certain scenes. The additional exploration strategies considered in this work provide only limited improvements. A more systematic investigation of exploration strategies may be a promising direction for future research.

**Acknowledgments** This work was supported by China Mobile Hunan Company Limited, China Mobile Communications Group Co., Ltd., and by the National Natural Science Foundation of China under Grant U22B2050, 62425305, 62221002, and 62503161. This work was conducted as part of the project "Networked Robotic System for Major Equipment Manufacturing (5G+Robotics)".

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

## A  REPRODUCIBILITY

The datasets are publicly available from (Kerbl et al., 2023). Methodological steps and formulas are detailed in Secs. 4.4 and J.1, and all hyperparameters are reported in Secs. J.2 and K.

More rendering visualization can be found in Figure 10.

## B  THE USE OF LARGE LANGUAGE MODELS (LLMs)

In this work, LLMs are used solely to assist in writing and polishing the manuscript; they do not contribute to the discovery or experimental design.

## C  TIME COST ANALYSIS

Within the 3DGS-MCMC framework, Sparse Adam substantially reduces the step time by approximately 50% (see Table 6) by eliminating implicit updates. The fluctuations in forward and backward costs can be attributed to variations in the number of active primitives during training. Although DAR encourages a larger proportion of dead primitives, Sparse Adam exhibits greater stability and tends to retain more active primitives.

Table 6: Time cost of different methods, which is divided into time cost of densification, forward, backward and update step. In all experiments, the loss function is implemented consistently. Baseline methods employ the native Adam optimizer in PyTorch, whereas our AdamW-GS follows the implementation described in this work.

| Methods | | Avg. | Bicycle | Flowers | Garden | Stump | Treehill | Room | Counter | Kichen | Bonsai |
|---|---|---|---|---|---|---|---|---|---|---|---|
| vanilla 3DGS | Densification | 0.37 | 0.43 | 0.38 | 0.49 | 0.39 | 0.37 | 0.32 | 0.34 | 0.34 | 0.32 |
| | Forward | 6.91 | 8.35 | 6.09 | 8.47 | 6.44 | 6.08 | 6.70 | 6.67 | 7.61 | 5.86 |
| | Backward | 13.72 | 14.26 | 11.27 | 15.75 | 11.54 | 11.38 | 14.69 | 14.62 | 17.38 | 12.59 |
| | Step | 9.55 | 16.30 | 10.25 | 17.08 | 13.65 | 10.53 | 4.63 | 3.75 | 5.69 | 4.12 |
| | All | 30.58 | 39.36 | 28.01 | 41.81 | 32.04 | 28.38 | 26.35 | 25.40 | 31.03 | 22.91 |
| RePR | Densification | 0.95 | 1.40 | 0.99 | 1.29 | 1.54 | 1.07 | 0.58 | 0.60 | 0.54 | 0.62 |
| | Forward | 6.07 | 6.55 | 5.41 | 6.85 | 5.73 | 5.40 | 6.00 | 6.23 | 7.04 | 5.50 |
| | Backward | 12.75 | 12.23 | 10.52 | 13.82 | 10.73 | 10.69 | 13.72 | 14.24 | 16.79 | 12.05 |
| | Step | 6.92 | 11.20 | 7.39 | 11.51 | 11.05 | 7.84 | 3.10 | 2.75 | 4.51 | 3.01 |
| | All | 26.73 | 31.40 | 24.33 | 33.48 | 29.06 | 25.02 | 23.41 | 23.83 | 28.90 | 21.20 |
| MaskGaussian | Densification | 0.37 | 0.46 | 0.36 | 0.49 | 0.41 | 0.38 | 0.31 | 0.30 | 0.32 | 0.33 |
| | Forward | 5.91 | 7.42 | 5.26 | 7.40 | 5.46 | 5.39 | 5.30 | 5.60 | 6.41 | 4.96 |
| | Backward | 13.66 | 14.19 | 11.2 | 15.66 | 11.20 | 11.45 | 14.15 | 14.81 | 17.61 | 12.69 |
| | Step | 6.83 | 12.50 | 7.64 | 12.52 | 9.38 | 8.12 | 2.39 | 2.29 | 3.94 | 2.74 |
| | All | 26.8 | 34.59 | 24.48 | 36.09 | 26.46 | 25.36 | 22.16 | 23.01 | 28.30 | 20.75 |
| 3DGS +AdamWGS(GS8) | Densification | 0.33 | 0.34 | 0.33 | 0.34 | 0.35 | 0.33 | 0.32 | 0.31 | 0.34 | 0.33 |
| | Forward | 5.19 | 5.21 | 4.45 | 4.81 | 4.75 | 4.49 | 5.54 | 5.69 | 6.27 | 5.56 |
| | Backward | 10.80 | 9.76 | 8.59 | 9.70 | 9.07 | 8.61 | 12.1 | 12.73 | 14.25 | 12.47 |
| | Step | 2.17 | 3.90 | 2.89 | 3.76 | 4.19 | 3.05 | 0.34 | 0.40 | 0.58 | 0.43 |
| | All | 18.53 | 19.23 | 16.28 | 18.62 | 18.37 | 16.50 | 18.39 | 19.15 | 21.45 | 18.81 |
| 3DGSMCMC | Densification | 0.02 | 0.04 | 0.03 | 0.04 | 0.04 | 0.03 | 0.02 | 0.02 | 0.02 | 0.02 |
| | Forward | 9.03 | 9.13 | 8.15 | 9.79 | 8.02 | 7.72 | 9.67 | 9.47 | 10.45 | 8.87 |
| | Backward | 18.95 | 17.39 | 15.49 | 19.19 | 14.85 | 14.39 | 22.55 | 22.20 | 24.54 | 19.97 |
| | Step | 21.09 | 31.51 | 19.45 | 33.76 | 24.68 | 20.46 | 10.23 | 8.12 | 12.17 | 8.75 |
| | All | 46.81 | 58.08 | 43.13 | 62.79 | 47.60 | 42.61 | 42.49 | 39.81 | 47.19 | 37.63 |
| 3DGSMCMC +AdamWGS(MC8) | Densification | 0.03 | 0.05 | 0.03 | 0.05 | 0.04 | 0.03 | 0.02 | 0.02 | 0.02 | 0.02 |
| | Forward | 9.18 | 9.50 | 8.10 | 10.15 | 8.51 | 7.91 | 9.55 | 9.26 | 10.84 | 8.85 |
| | Backward | 11.46 | 18.33 | 15.25 | 21.32 | 16.02 | 14.41 | 21.18 | 19.91 | 25.61 | 19.66 |
| | Step | 11.46 | 18.75 | 12.04 | 20.64 | 14.84 | 12.35 | 6.20 | 5.06 | 7.89 | 5.40 |
| | All | 39.77 | 46.65 | 35.44 | 52.17 | 39.41 | 34.71 | 36.97 | 34.27 | 44.39 | 33.94 |

For vanilla 3DGS, the reduction in time cost benefits from both the decrease in update-step overhead achieved by Sparse Adam and the reduced forward and backward costs resulting from fewer primitives. Overall, our method achieves more than a 40% reduction in total runtime. In indoor scenes with fewer primitives, the update-step cost becomes negligible, as illustrated in Figure 4. Due to the additional loss used by MaskGaussian for pruning, it yields almost no improvement in the backward time cost.

## D  PRELIMINARY

**3DGS(-MCMC)**  3DGS approximates the radiance field of targeted scenes via a group of Gaussian primitives $\{\mathcal{G}_i\}^n$ parameterized with $\boldsymbol{\theta} = \{\boldsymbol{\theta}_i \in \Theta\}_{i=1}^n$ where $\boldsymbol{\theta}_i \triangleq (\boldsymbol{\mu}_i, \boldsymbol{\Sigma}_i, o_i, \boldsymbol{c}_i)$, $\boldsymbol{\mu}_i \in \mathbb{R}^3$ denotes the primitive position, $\boldsymbol{\Sigma}^{(i)} \in \mathbb{S}_+^{3\times3}$ is the positive semi-definite covariance matrix decoupled into quaternion and a scaling vector, $o^{(i)}$ is considered as the opacity value. The color attribute $\boldsymbol{c}^{(i)} \in \mathbb{R}^{(3)}$ is typically stored as spherical harmonics coefficients and converted into RGB when rendering.

Given a viewpoint $k$ and transformation (Kerbl et al., 2023), the pixel $\boldsymbol{p}$ on the screen is rendered via $\alpha$-blending according to the projected depth of primitives:

$$\mathbf{C}^k(\boldsymbol{p}; \boldsymbol{\theta}) = \sum_{i=1}^{N_{\boldsymbol{p}}^k} \boldsymbol{c}_i^k \underbrace{o_i \mathcal{G}_i^k(\boldsymbol{p}; \boldsymbol{\theta}_i)}_{\alpha_i^k(\boldsymbol{p}; \boldsymbol{\theta}_i)} \underbrace{\prod_{j=1}^{i-1}(1 - \alpha_j^k(\boldsymbol{p}; \boldsymbol{\theta}_j))}_{\text{Transmittance } T_i^k(\boldsymbol{p}; \boldsymbol{\theta}_i)} \tag{9}$$

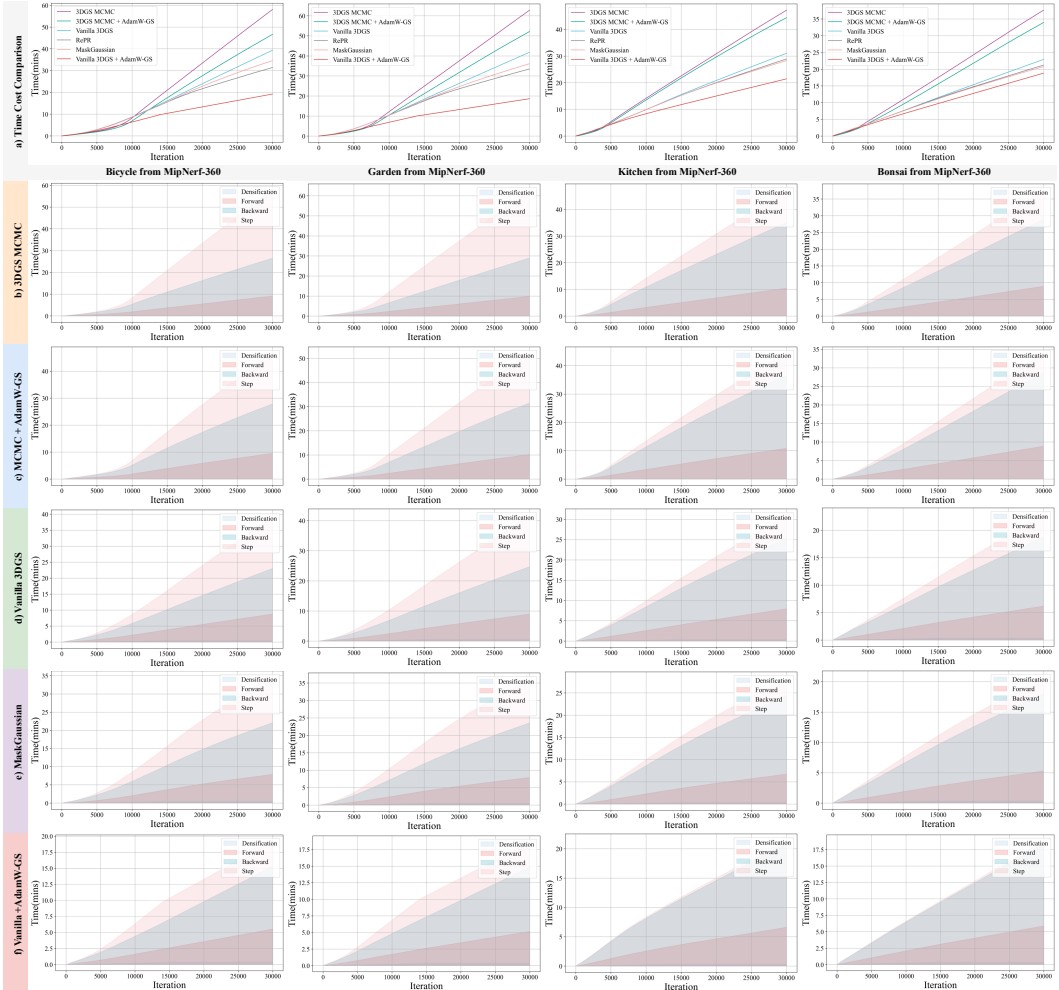

Figure 4: Comparison of Training Time Costs.

Here the superscript $k$ denotes the corresponding parameters under the given viewpoint after the respective transformation, whereas $\mathcal{G}_i^k = \exp(-\frac{1}{2}(\boldsymbol{\mu}_i^k - \boldsymbol{p})^\top \boldsymbol{\Sigma}_i^k (\boldsymbol{\mu}_i^k - \boldsymbol{p}))$.

**Optimization** The 3DGS training pipeline typically initializes from Structure-from-Motion (SfM) (Schonberger & Frahm, 2016) and then trained with several warm-up iterations (Kerbl et al., 2023; Jung et al., 2024). The primitives then undergo **densification**, wherein new primitives are generated to enhance scene representation while redundant ones are pruned. In the vanilla pipeline shown in Figure 9 a, both operations are governed by adaptive density control: new primitives are created via cloning or splitting according to gradient magnitude, whereas low-opacity primitives are removed. After densification, optimization proceeds in a **pure optimization** (**P-Op**) stage, during which only gradient propagation and attribute updates are performed. Throughout warm-up, densification, and P-Op, training employs the photometric term in Eq. 1 and uses the Adam optimizer. Building on Stochastic Gradient Langevin Dynamics (SGLD), 3DGS-MCMC (Kheradmand et al., 2024) follows a similar stage division, as shown in Figure 9 b: new primitive sampling and dead primitive real-location are performed during densification, followed by a shortened P-Op. It also adopts Adam, augmented with the regularization loss term in Eq. 1 to promote primitive respawning, together with additional positional noise for exploration and consistency with the SGLD formulation.

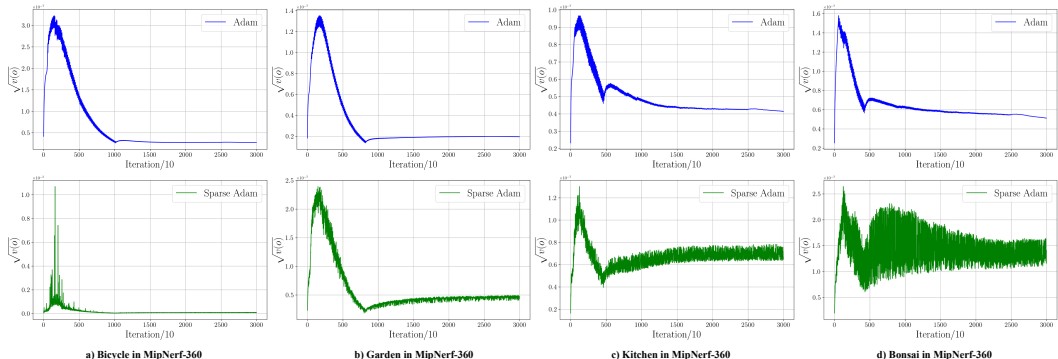

**a) Bicycle in MipNerf-360**   **b) Garden in MipNerf-360**   **c) Kitchen in MipNerf-360**   **d) Bonsai in MipNerf-360**

Figure 5: The average of the second moment in valid primitives.

Table 7: Quantitative results of vanilla 3DGS and MaskGaussian (Liu et al., 2025b) with and without AdamW-GS on the Mip-NeRF 360 dataset.

| Pipeline | AdamW-GS | All | | | | Outdoor | | | | Indoor | | | |
|---|---|---|---|---|---|---|---|---|---|---|---|---|---|
| | | PSNR↑ | SSIM↑ | LPIPS↓ | $\Delta N_a$ | PSNR↑ | SSIM↑ | LPIPS↓ | $\Delta N_a$ | PSNR↑ | SSIM↑ | LPIPS↓ | $\Delta N_a$ |
| vanilla 3DGS | x | 27.506 | 0.815 | 0.216 | (3.098m) | 24.648 | 0.728 | 0.239 | (4.512m) | 31.080 | 0.925 | 0.189 | (1.331m) |
| | ✓ | 27.678 | 0.822 | 0.220 | -49.3% | 24.854 | 0.740 | 0.243 | -48.4% | 31.209 | 0.925 | 0.191 | -50.4% |
| MaskGaussian | x | 27.485 | 0.815 | 0.219 | -53.1% | 24.683 | 0.728 | 0.240 | -46.4% | 30.988 | 0.924 | 0.192 | -61.5% |
| | ✓ | 27.721 | 0.821 | 0.221 | -57.2% | 24.939 | 0.739 | 0.244 | -48.1% | 31.199 | 0.925 | 0.193 | -68.6% |

# E  MORE PIPELINE VARIANTS WITH ADAMW-GS OR EXPERIMENTS ON DIFFERENT DATASETS

## E.1  MASKGAUSSIAN WITH ADAMW-GS: MORE STABLE UPDATING OF MASK SCORE

MaskGaussian introduces a probabilistic formulation of 3D Gaussian primitives, where each primitive is assigned a learnable probability of existence that governs a dynamic sampling process during rendering. This mechanism enables adaptive pruning of redundant primitives throughout the 3DGS optimization. Specifically, each primitive is associated with a mask score $\pi_i$, from which a binary mask $\mathcal{M}_i$ is stochastically sampled using the Gumbel-Softmax reparameterization. The standard rendering equation in Eq. 9 is thus modified to incorporate the mask in both color accumulation and transmittance updates, as shown in Eq. 10. Detailed derivations and implementation are provided in (Liu et al., 2025b).

$$\mathbf{C}^k(\boldsymbol{p};\boldsymbol{\theta}) = \sum_{i=1}^{N_{\boldsymbol{p}}^k} \mathcal{M}_i \cdot \boldsymbol{c}_i^k \cdot \alpha_i^k(\boldsymbol{p};\boldsymbol{\theta}_i) \cdot T_i^k(\boldsymbol{p};\boldsymbol{\theta}_i) \quad \text{where} \ \mathcal{M}_i \sim \mathrm{GumbelSoftmax}(\boldsymbol{\pi}) \quad (10)$$

$$\boldsymbol{\pi}_{t+1} = \boldsymbol{\pi}_t - \eta \times \left[\frac{\hat{m}(\boldsymbol{\pi})_t'}{\sqrt{\hat{v}(\boldsymbol{\pi})_t'} + \epsilon}\right] \quad (11)$$

When MaskGaussian is equipped with AdamW-GS, the observed PSNR improvement aligns with that of vanilla 3DGS using AdamW-GS. Additionally, for indoor scenes, it enables an extra pruning of approximately 7% of primitives. This suggests that synchronous updates of mask scores may lead to potentially "destructive pruning behavior". In contrast, our method cannot only be jointly used with such pruning strategies to stabilize their inherent training dynamics, but also further improve overall efficiency, pruning performance, and representation quality simultaneously.

## E.2  TAMING-3DGS WITH ADAMW-GS

Taming-3DGS (Mallick et al., 2024) constrains the total number of primitives by restricting the number of new primitives added at each densification step, as defined by Eq. 12. During every densification step, Taming-3DGS computes a score for each primitive to determine its probability of being selected for densification. Based on these scores, a fixed number of primitives are sampled for densification. Additionally, Taming-3DGS incorporates further optimizations, such as enhanced

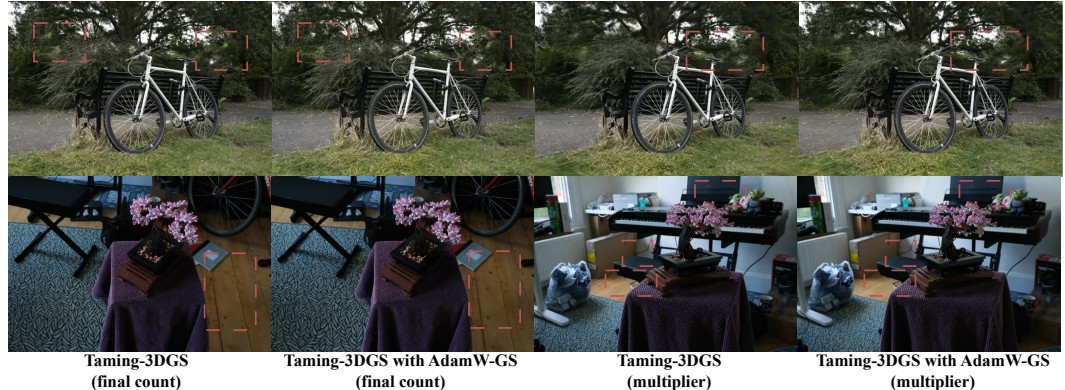

| Taming-3DGS
(final count) | Taming-3DGS with AdamW-GS
(final count) | Taming-3DGS
(multiplier) | Taming-3DGS with AdamW-GS
(multiplier) |

Figure 6: Rendering visualization of Taming-3DGS with or without AdamW-GS.

Table 8: Quantitative results of Taming-3DGS (Mallick et al., 2024) with and without AdamW-GS on the Mip-NeRF 360 dataset. A brief introduction to Taming-3DGS is provided in Appendix Sec. E.2. All primitive counts ($N_p$ and $N_a$) are reported in millions, and time cost is reported in minutes.

| Pipeline | AdamW-GS | All | | | | | | Indoor | | | | | Outdoor | | | | |
|---|---|---|---|---|---|---|---|---|---|---|---|---|---|---|---|---|---|
| | | PSNR↑ | SSIM↑ | LPIPS↓ | $N_p$ | $N_a$ | Time | PSNR | SSIM | LPIPS | $N_p$ | $N_a$ | PSNR | SSIM | LPIPS | $N_p$ | $N_a$ |
| Taming-3DGS (multiplier) | x | 27.386 | 0.796 | 0.258 | 0.668 | 0.620 | 7.44 | 31.025 | 0.918 | 0.205 | 0.357 | 0.329 | 24.479 | 0.698 | 0.299 | 0.916 | 0.853 |
| | ✓ | 27.537 | 0.799 | 0.254 | 0.646 | 0.575 | 5.27 | 31.268 | 0.920 | 0.200 | 0.357 | 0.313 | 24.552 | 0.703 | 0.297 | 0.877 | 0.785 |
| Taming-3DGS (final count) | x | 27.912 | 0.822 | 0.207 | 3.205 | 2.609 | 20.30 | 31.603 | 0.928 | 0.181 | 1.377 | 1.144 | 24.959 | 0.736 | 0.228 | 4.667 | 3.776 |
| | ✓ | 28.034 | 0.826 | 0.207 | 3.109 | 1.847 | 10.46 | 31.720 | 0.928 | 0.180 | 1.408 | 0.759 | 25.085 | 0.744 | 0.229 | 4.469 | 2.717 |
| Taming-3DGS-p (final count) | ✓ | 28.038 | 0.828 | 0.205 | 2.160 | 2.160 | 8.44 | 31.724 | 0.930 | 0.177 | 0.804 | 0.804 | 25.089 | 0.745 | 0.227 | 3.246 | 3.246 |

parallelization. For detailed implementation and algorithmic design, please refer to (Mallick et al., 2024).

$$A(\text{step}_x) = \frac{\Omega_{\text{densi}} - \Omega_0 - 2\text{step}_{\text{densi}}}{\text{step}_{\text{densi}}^2}\text{step}_x^2 + 2\text{step}_x + \Omega_{\text{densi}} \tag{12}$$

Where $\text{step}_x$ denotes the current number of densification steps, $\text{step}_{\text{densi}}$ represents the total number of densification iterations, $\Omega_{\text{densi}}$ refers to the upper bound on the number of primitives after densification, and $\Omega_0$ indicates the initial number of primitives.

Two operating modes are provided in Taming-3DGS[4]: final count and multiplier, which correspond to a higher and a lower upper bound on the final total primitive number $N_p$, respectively. Experiments were conducted under both modes for Taming-3DGS as well as Taming-3DGS with AdamW-GS. DAR in this setting only includes opacity and scaling regularization as well, following the same configuration as in Sec. 4.4. Noise regularization is omitted because the growth rate in Taming-3DGS is constrained. Considering that AdamW-GS imposes stronger penalties on redundant primitives, leading to the generation of a large number of dead primitives that are rapidly pruned and thus reduce the total primitive count significantly, we propose to replace the conservative pruning strategy in Taming-3DGS with the pruning method used in vanilla 3DGS when integrating AdamW-GS. We denote this modified pipeline as Taming-3DGS-p. Related experimental results are summarized in Table 8.

As shown in Table 8, under the same constraint on the final $N_p$, the variants trained with AdamW-GS consistently outperform their original counterparts in reconstruction quality while also improving training speed. Visual comparisons in Figure 6 further demonstrate that our method more faithfully preserves fine scene details. When a relatively large final primitive budget is given, i.e., under the final-count setting in Table 8, AdamW-GS reduces the total training time by nearly half, from 20.30 min in the original Taming-3DGS to 10.46 min in Taming-3DGS with AdamW-GS. Since AdamW-GS effectively penalizes redundant primitives at an early stage, we further incorporate the pruning

---

[4]https://github.com/humansensinglab/taming-3dgs

Table 9: Quantitative results of Deformable Beta Splatting (DBS) (Liu et al., 2025a) with and without AdamW-GS on the Mip-NeRF 360 dataset. A brief introduction to DBS is provided in Appendix Sec. E.3. The Treehill scene from Mip-NeRF 360 is excluded.

| AdamW-GS | All | | | Outdoor | | | Indoor | | |
|---|---|---|---|---|---|---|---|---|---|
| | PSNR↑ | SSIM↑ | LPIPS↓ | PSNR↑ | SSIM↑ | LPIPS↓ | PSNR↑ | SSIM↑ | LPIPS↓ |
| x | 29.362 | 0.864 | 0.165 | 26.029 | 0.787 | 0.187 | 32.696 | 0.940 | 0.143 |
| ✓ | 29.643 | 0.871 | 0.158 | 26.108 | 0.798 | 0.175 | 33.178 | 0.945 | 0.140 |

Table 10: Quantitative results for vanilla 3DGS and 3DGS-MCMC with and without AdamW-GS on the OMMO dataset (Lu et al., 2023). All primitive counts ($N_p$ and $N_a$) are reported in millions.

| AdamW-GS | 3DGSMCMC | | | | | 3DGS | | | | |
|---|---|---|---|---|---|---|---|---|---|---|
| | PSNR↑ | SSIM↑ | LPIPS↓ | $N_p$ | $N_a$ | PSNR↑ | SSIM↑ | LPIPS↓ | $N_p$ | $N_a$ |
| x | 30.359 | 0.925 | 0.135 | 1.960 | 1.673 | 30.040 | 0.914 | 0.154 | 1.878 | 1.640 |
| ✓ | 30.716 | 0.930 | 0.126 | 1.960 | 1.765 | 30.351 | 0.914 | 0.154 | 1.245 | 1.211 |

strategy from vanilla 3DGS to construct Taming-3DGS-p. With our optimizer, this variant further reduces the peak primitive count during training, from 3.205 million in the original Taming-3DGS to 2.160 million in Taming-3DGS with AdamW-GS, thereby yielding additional GPU memory savings.

### E.3 DEFORMABLE BETA SPLATTING WITH ADAMW-GS

Deformable Beta Splatting (DBS) (Liu et al., 2025a) shares a similar pipeline design and the same loss as 3DGS-MCMC (Kheradmand et al., 2024), as both are formulated within the Stochastic Gradient Langevin Dynamics framework. Unlike 3DGS-MCMC, DBS replaces the Gaussian kernel with the Beta kernel $\mathcal{B}(\boldsymbol{p}; \boldsymbol{\theta}_i)$ and introduces a Spherical Beta function to better represent complex geometries and diverse appearance attributes. Consequently, the rendering process is reformulated as shown in Eq. 13.

$$\mathbf{C}^k(\boldsymbol{p}; \boldsymbol{\theta}) = \sum_{i=1}^{N_{\boldsymbol{P}}^k} \boldsymbol{c}_i^k o_i \mathcal{B}_i^k(\boldsymbol{p}; \boldsymbol{\theta}_i) \prod_{j=1}^{i-1} (1 - o_i \mathcal{B}_i^k(\boldsymbol{p}; \boldsymbol{\theta}_i)) \tag{13}$$

Consistent with our experiments on 3DGS-MCMC, we apply AdamW-GS to DBS, and the results are summarized in Table 9. Across eight of the nine scenes, excluding Treehill, we observe clear improvements in reconstruction quality when using AdamW-GS. For the Treehill scene, however, we find that DBS suffers from pronounced overfitting and triggers early stopping prematurely. Meanwhile, we observe that during the densification stage, $N_d$ has been closed early to zero, indicating that DAR cannot effectively exert its intended effect in this scenario. Our method may be less effective in such heavily overfitting scenario.

### E.4 VANILLA 3DGS OR 3DGSMCMC WITH ADAMW-GS ON OMMO DATASETS

We provide results on the OMMO dataset (Lu et al., 2023), which contains large-scale outdoor scenes with long-range sequences. Our data processing follows the settings described in (Kheradmand et al., 2024) and its associated Github repository [5]. However, the image preprocessing failed for scene #10; thus, we report the average results for the left scenes. All results are summarized in Table 10. Our method demonstrates effectiveness on long sequence datasets as well.

## F EXPLORATION STRATEGIES

In Sec. 4.1, we observe that Sparse Adam exhibits is less explosive. This behavior is partly due to the stability described in Observation 1 and partly attributable to the inherent poor transport property of 3DGS (Jung et al., 2024), which we briefly discussed in Sec. 4.4. This phenomenon causes a larger portion of primitives to remain in a stable state during optimization with Sparse Adam, neither triggering further primitive generation nor contributing to continued reconstruction improvement. As a

---

[5]https://github.com/ubc-vision/3dgs-mcmc

Table 11: Quantitative results of vanilla 3DGS under different modifications on Mip-NeRF 360. More details are provided in Sec. F. "+ Only RSR" denotes the variant that uses only Sparse Adam and RSR in the 3DGS pipeline.

| Pipeline | All | | | | Outdoor | | | | Indoor | | | |
|---|---|---|---|---|---|---|---|---|---|---|---|---|
| | PSNR↑ | SSIM↑ | LPIPS↓ | $\triangle N_a$ | PSNR↑ | SSIM↑ | LPIPS↓ | $\triangle N_a$ | PSNR↑ | SSIM↑ | LPIPS↓ | $\triangle N_a$ |
| vanilla 3DGS | 27.506 | 0.815 | 0.216 | (3.098m) | 24.648 | 0.728 | 0.239 | (4.512m) | 31.080 | 0.925 | 0.189 | (1.331m) |
| + Only RSR (GS0) | 27.483 | 0.818 | 0.217 | -28.6% | 24.685 | 0.733 | 0.238 | -27.9% | 30.981 | 0.924 | 0.190 | -29.5% |
| +AdamW-GS (GS8) | 27.678 | 0.822 | 0.220 | -49.3% | 24.854 | 0.740 | 0.243 | -48.4% | 31.209 | 0.925 | 0.191 | -50.4% |
| +AdamW-GS (GS8) + ABE | 27.751 | 0.822 | 0.220 | -41.1% | 24.909 | 0.740 | 0.243 | -38.5% | 31.304 | 0.925 | 0.191 | -44.3% |
| +AdamW-GS (GS8) + Longer Densi | 27.715 | 0.824 | 0.218 | -48.4% | 24.857 | 0.744 | 0.240 | -44.6% | 31.288 | 0.925 | 0.190 | -53.2% |

result, the final primitive set has a smaller $N_a$ but inferior reconstruction quality. In Sec. 5 (Extra Exploration is Necessary), we examine the necessity of additional exploration through experiments that introduce noise regularization. Here, exploration is deliberately to encourage primitives to undergo larger movements, which is consistent with the notion of bad transport described in (Jung et al., 2024). We have given the discussion in Sec. 4.4.

**Exploration in Current AdamW-GS** In Sec. 5, we argue that the current AdamW-GS improves exploration. This enhancement arises from several factors: (1) **RSR**. We provide additional ablation studies showing that, when using only Sparse Adam together with RSR, training the vanilla 3DGS pipeline on Mip-NeRF 360 yields reconstruction quality comparable to vanilla 3DGS with Adam. The corresponding results are reported in Table 11 under "+ Only RSR (GS0)". (2) **The influence of DAR around saddle points**. The analysis in (Wang et al., 2025) indicates that many primitives become trapped near saddle points, preventing further effective optimization. The presence of DAR encourages these primitives to continue participating in optimization rather than stagnating. (3) **Improved gradient flow after removing redundant primitives**. When a large number of redundant primitives are eliminated, the gradient flow within primitive groups becomes more coherent and effective. (4) **Noise regularization for outdoor scenes**. For outdoor datasets, we intentionally introduce noise regularization to further encourage extra exploration.

Existing work provides only limited discussion of exploration strategies for the 3DGS pipeline. Noise regularization, while helpful in certain cases, has inherent limitations and is not applicable to all datasets. Nevertheless, we argue that developing additional exploration strategies is a promising direction, as such mechanisms encourage primitives to explore a broader region of the space. Building on our current work, we introduce two additional exploration strategies: **Adaptive Bound-Expanding Split** (Jung et al., 2024) and **Densification Extending**. For outdoor scenes, the following experiments still employ noise regularization; we further demonstrate that these strategies can be used jointly.

**Adaptive Bound-Expanding Split (ABE-Split)** divides each Gaussian into three, where the location of the third cross-region primitive is initialized using a constant factor proportional to the scene extent. The corresponding results are reported in Table 11 "+ AdamW-GS (GS8) + ABE". ABE-Split consistently improves reconstruction quality across both indoor and outdoor scenes. In some cases, the gains are particularly clear—for example, on the Room scene from Mip-NeRF 360, the PSNR/SSIM increases from 31.500 dB / 0.920 (vanilla 3DGS) to 32.121 dB / 0.923.

**Densification Extending:** We extend the original densification phase from 15,000 to 25,000 iterations to allow primitives to continue being generated and refined. Because our method quickly penalizes redundant primitives, this extension does not introduce the risk of memory explosion. The corresponding results are reported in Table 11 under "+ AdamW-GS (GS8) + Longer Densi". Extending the densification phase further benefits indoor scenes and additionally reduces the number of active primitives.

## G  FAILURE CASES

Although AdamW-GS yields improved reconstruction quality—both quantitatively and qualitatively in terms of rendering visualization—compared with the original method, it only offers an

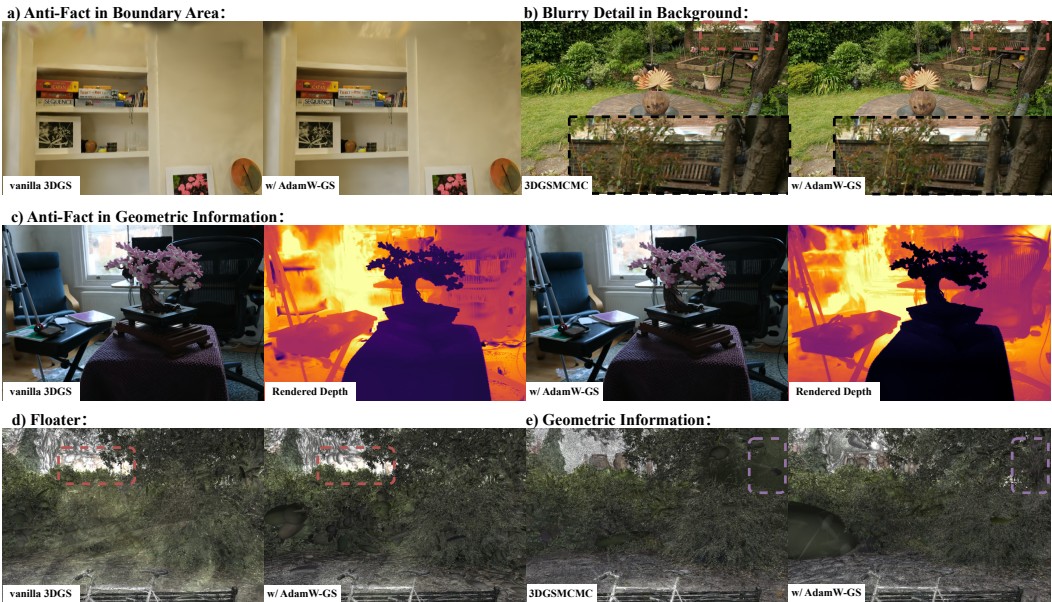

Figure 7: More examples to show the failure cases: **a**) the rendering results from Room to show the anti-fact boundary area; **b**) the rendering results from Garden to show the blurriness in background; **c**) the rendering results and normalized depth maps from bonsai to show anti-fact in geometric information; **d-e**) the ellipsoid visualization from Bicycle.

optimization-level enhancement and does not fundamentally resolve the inherent limitations of the pipeline itself.

- Severe artifacts in boundary regions with insufficient view coverage. As shown in Figure 7 a), boundary areas on walls exhibit substantial artifacts, floaters, and even incorrect colors due to the lack of sufficient multi-view constraints.
- Insufficient refinement of background regions. These areas often remain under-optimized and may suffer from blurriness. We give an example in Figure 7 b).
- Geometric inconsistency despite reasonable rendering results. Our current optimization does not incorporate any additional geometric priors. From the normalized depth map computed using Eq. 14, as shown in the rendered-depth visualization in Figure 7 c, many regions can be observed to violate geometric plausibility, such as reflective book regions and shadowed chair regions. Without explicit geometric priors, AdamW-GS cannot fully recover the correct underlying geometry. Nonetheless, certain improvements in geometry can be observed—for instance, the tree trunk region highlighted by the purple box in Figure 7 e).
- Persisting floater issues when the original pipeline suffers from floaters. Even with AdamW-GS, floaters may remain, as illustrated in Figure 7 d).
- Failure under severe overfitting. When the reconstruction pipeline significantly overfits the scene, AdamW-GS may break down. We encountered this issue when using DBS to reconstruct Treehill from the Mip-NeRF 360 dataset. Overfitting can prematurely trigger early stopping, and the sharp decreasing of $N_p$ leads to the failure of DAR.

$$\mathbf{D}^k(\mathbf{p}) = \sum_{i=1}^{N_p^k} d_i^k \cdot \alpha_i^k(\boldsymbol{p}) \cdot T_i^k(\boldsymbol{p}) \tag{14}$$

where $d_i^k$ is the depth of the primitive in the corresponding camera coordinations.

## H MORE ABOUT RELATED WORK

**Redundancy during Optimization** (1) Densification refinement: Several works refine the densification rule of 3DGS (Fang & Wang, 2024; Mallick et al., 2024; Wang et al., 2025). SteepGS (Wang

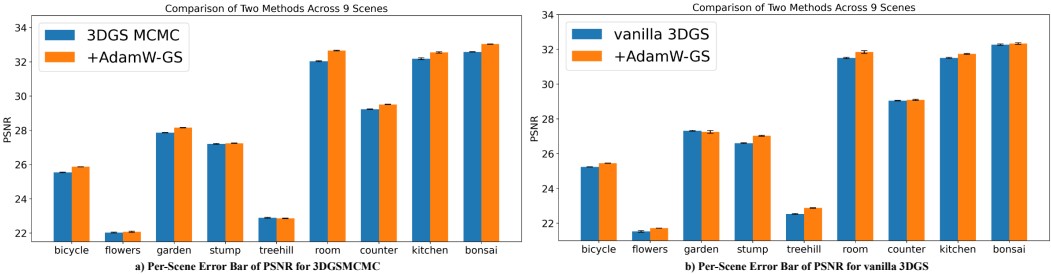

Figure 8: The PSNR error bar plot across nine scenes is shown for both methods, while the other metrics exhibit no notable variation. A more detailed per-scene result is provided in Sec. K.

et al., 2025) mitigates redundancy and alleviates under-optimization by introducing optimization-conditioned densification, but its reconstruction quality and pruning effectiveness remain inferior to those of learning-based methods. (2) Hand-crafted criteria: These methods design importance or redundancy criteria to guide pruning after densification. LightGaussian (Fan et al., 2024) scores each Gaussian using the product of opacity, transmittance, scale, and ray contribution. RadSplat (Niemeyer et al., 2025) uses the maximum alpha–transmittance product across views. (Papantonakis et al., 2024) propose a pipeline to prune primitives based on the degree of primitive overlap. PUP-3DGS(Hanson et al., 2025) proposes a principled sensitivity pruning score which is computed as a second-order approximation of the reconstruction error on the training views. (3) Learning-based pruning: Learning-based methods adaptively remove redundant primitives, often through learnable masks. Compact3DGS (Lee et al., 2024) introduces Gaussian masks with L1 regularization on active mask count. MaskGaussian (Liu et al., 2025b) models the existence of primitives, or measures the importance via learning. LP-3DGS(Zhang et al., 2024a) redesigns the masking function to leverage the Gumbel-Sigmoid method. (4) Optimization-related operations: Optimizing together with additional opacity L1 regularization has become a common technique for redundancy removal, even extending beyond Efficient 3DGS tasks (Papantonakis et al., 2024; Lee et al., 2024; Kheradmand et al., 2024; Liu et al., 2025a; Svitov et al., 2024).

Redundancy reduction and training efficiency constitute subtopics of efficient 3DGS, involving inference-time and training-time acceleration, with partial overlap between the two. For inference, approaches include vector quantization (Lee et al., 2024), knowledge distillation (Fan et al., 2024), and entropy coding (Huang et al., 2025). For training, existing efforts mainly target backward acceleration and loss redesign (Mallick et al., 2024). These topics fall beyond the scope of this work: for fair comparison, we maintain the same loss function and refrain from using additional compression techniques.

**Optimizer and Regularization**  The development of adaptive first-order optimizers can be traced back to RProp (Riedmiller & Braun, 1993). AdaGrad (Duchi et al., 2011) adapts the learning rate of features by estimated geometry and assigns larger learning rate to infrequent features. RMSProp (Hinton et al., 2012) further stabilizes training by normalizing updates with an exponential moving average of squared gradients. Building on this, Adam (Kinga et al., 2015) incorporates momentum into RMSProp through an exponential average of gradients, quickly becoming the default optimizer for modern DNNs (Vaswani et al., 2017). Numerous refinements and extensions to Adam have since been proposed (You et al., 2019; Chen et al., 2023). Regularization has long been recognized as a key technique to improve model generalization and has also been employed as a tool to facilitate more effective optimization (Srivastava et al., 2014; Shalev-Shwartz & Ben-David, 2014; Andriushchenko et al., 2023; Brown et al., 2020; Radford et al., 2021). Importantly, Loshchilov & Hutter (2017) demonstrated that L2 regularization and weight decay are not equivalent, and further showed that weight decay provides a more appropriate formulation for Adam.

**Optimization for 3DGS**  Sparse Adam (Mallick et al., 2024) introduces asynchronous updates that improve efficiency, yet the insufficient recognition of update-step coupling results in degraded performance. 3DGS-LM (Höllein et al., 2025) proposes a tailored Levenberg-Marquardt optimizer for acceleration. However, it still relies on Adam for the initialization or densification. Although Adam has become a de facto indispensable optimizer for 3DGS, our understanding of its behavior in

Table 12: Quantitative results for 3DGS-MCMC + Sparse Adam with different opacity regularization settings on Mip-NeRF 360.

| Methods | Cite | $\lambda_o$ | Outdoor | | | | Indoor | | | | All | | | |
|---|---|---|---|---|---|---|---|---|---|---|---|---|---|---|
| | | | PSNR | SSIM | LPIPS | $\Delta N_a$ | PSNR | SSIM | LPIPS | $\Delta N_a$ | PSNR | SSIM | LPIPS | $\Delta N_a$ |
| | MC2 | 0.01 | 25.160 | 0.754 | 0.212 | 3.92% | 31.546 | 0.930 | 0.183 | 4.74% | 27.998 | 0.832 | 0.199 | 4.28% |
| $\|o\|_1$ | MC101 | 0.1 | 17.379 | 0.400 | 0.604 | -77.8% | 29.957 | 0.906 | 0.220 | -40.7% | 22.969 | 0.625 | 0.433 | -59.3% |
| | MC102 | 0.001 | 24.585 | 0.731 | 0.220 | 5.18% | 31.342 | 0.930 | 0.179 | 8.1% | 27.588 | 0.819 | 0.202 | 6.28% |
| | MC103 | 0.1 | 24.298 | 0.714 | 0.242 | 5.79% | 30.687 | 0.921 | 0.193 | -2.08% | 27.138 | 0.806 | 0.220 | 2.83% |
| AdamW o | MC104 | 1 | 24.155 | 0.698 | 0.288 | 3.80% | 30.326 | 0.917 | 0.203 | -7.47% | 26.898 | 0.795 | 0.250 | 5.43% |
| | MC105 | 10 | 17.031 | 0.350 | 0.654 | -19.8% | 20.134 | 0.680 | 0.457 | -11.6% | 18.410 | 0.497 | 0.567 | -16.1% |
| AdamW o | MC106 | 0.1 | 24.239 | 0.713 | 0.242 | 5.77% | 30.992 | 0.928 | 0.182 | 9.66% | 27.240 | 0.808 | 0.215 | 7.50% |
| + clip | MC107 | 1 | 24.381 | 0.713 | 0.260 | 5.08% | 30.850 | 0.922 | 0.194 | 8.76% | 27.256 | 0.806 | 0.231 | 6.71% |
| (max 10) | MC108 | 10 | 19.358 | 0.440 | 0.560 | 2.69% | 21.901 | 0.727 | 0.436 | 7.99% | 20.488 | 0.568 | 0.505 | 5.04% |

this context remains limited. A similar situation arises for regularization: despite the widespread use of various regularization techniques in 3DGS, there is currently little work examining the interplay between regularization and the optimizer, even though this has already been explored in the deep neural networks (Loshchilov & Hutter, 2017).

# I  PREPARATION FOR DECOUPLED ATTRIBUTE REGULARIZATION

## I.1  HYPERPARAMETER ADJUSTMENT

This section discusses the influence of hyperparameters on regularization. For both opacity and scaling regularization in 3DGS-MCMC, a default value of 0.01 is adopted. Here, we vary only the opacity hyperparameter by scaling it up or down by a factor of 10, with the results summarized in Table 12. When increased to 0.1, the stronger regularization leads to severe degradation in reconstruction quality. Conversely, reducing it to 0.001 weakens the regularization effect to the extent that the primitive reallocation process is significantly disrupted, also resulting in quality loss. These results suggest that simply tuning fixed hyperparameters is insufficient to effectively control the strength of regularization.

## I.2  ADAMW STYLE DECOUPLING

The AdamW-style decoupling implies an equal penalty applied to attributes, which can be formally expressed in Eq. 15 and Eq. 16. This formulation is equivalent to imposing a constant penalty on opacity (Rota Bulò et al., 2025). Within the 3DGS-MCMC framework, we apply AdamW-style decoupling to opacity and evaluate it under three hyperparameter settings corresponding to different strengths. To disentangle the effect of magnitude, we also experiment with constraining the magnitude of the AdamW-style decoupling, as given in Eq. 17.

The results are summarized in Table 12. Consistent with the findings in Appendix Sec. I.1, excessively large hyperparameters hinder convergence, while overly small values render the regularization ineffective. A key limitation of AdamW-style decoupling is that it enforces uniform penalties across all primitives, despite their varying importance.

$$m(\theta)'_t = \beta'_1 \times m(\theta)'_{t-1} + (1 - \beta'_1) \times \frac{\nabla \ell(\theta)}{N_I} \quad v(\theta)'_t = \beta'_2 \times v(\theta)'_{t-1} + (1 - \beta'_2) \times (\frac{\nabla \ell(\theta)}{N_I})^2 \quad (15)$$

$$\theta_{t+1} = \theta_t - \eta \times (\frac{\hat{m}(\theta)'_t}{\sqrt{\hat{v}(\theta)'_t} + \epsilon} + \lambda \nabla \mathcal{R}(\theta)) \quad (16)$$

$$\theta_{t+1} = \theta_t - \eta \times (\frac{\hat{m}(\theta)'_t}{\sqrt{\hat{v}(\theta)'_t} + \epsilon} + \min(\lambda \nabla \mathcal{R}(\theta), \mathcal{C}_t)) \quad (17)$$

### I.3 ADDITIONAL DISCUSSION REGARDING THE FORMULATION IN EQ.8

A thorough decoupling—i.e., activation via the second moment of $\nabla \mathcal{R}$—is unreasonable: in addition to incurring extra computational and memory costs, it reduces regularization to a trivial attribute-dependent penalty (e.g., higher opacity automatically incurs stronger regularization).

### I.4 DISCUSSION RELATED TO IMPLICIT UPDATE

**Ineffective when stable:** Implicit updates allow primitives that are invisible from the current viewpoint to be updated based on their optimizer moments. Our reasoning, grounded in experimental results, is as follows. When primitives are in a stable state—more precisely, near saddle points, which are abundant in 3DGS when no external perturbation is applied (Wang et al., 2025)—implicit updates are largely ineffective while incurring additional computational cost. In vanilla 3DGS, we implemented AIU (see GS4 and GS3 in Sec. K), and observed negligible improvement.

**Less controllable when unstable:** In contrast, when primitives are unstable, such as during the early training phase, e.g., densification, where newly generated primitives often have zero optimizer moments, or when the update step is relatively large, or when optimizer moments are rescaled, implicit updates can become problematic. We design several experiments to highlight their uncontrollability. For instance, adopting AIU during densification in vanilla 3DGS with Sparse Adam can lead to more primitives but worse performance, as shown by GS6 in Appendix Sec. K, as even small update steps for invisible primitives introduce instability. According to Obsidian 1, Sparse Adam tends to be stable, causing fewer primitives to satisfy the gradient condition. AIU disrupts this stability: when primitives overfit certain views due to implicit updates, they induce larger gradients in other views, triggering unnecessary densification. The additional primitives do not necessarily improve rendering quality. In the Appendix Sec.K, we provide multiple AIU configurations: although in some cases the quality approaches that of vanilla 3DGS, the more common results are in worse quality, confirming the uncontrollable nature of implicit updates.

Another related phenomenon is the emergence of dead primitives and the increase in reallocated primitives (see Sec.5.1). The comparison between GS1 and GS3 in Table 1, with full results provided in Appendix Sec. K, shows that Adam produces more dead primitives than Sparse Adam (0.232M vs. 0.048M). A natural conjecture is that when the update step, from a certain viewpoint, encourages primitives to become dead primitives, the implicit update accelerates this process, thereby producing more dead primitives. We provide several comparative experiments in Appendix Sec. K. (1) MC2 vs. MC3 in Appendix Sec. K, comparing the original 3DGSMCMC with Sparse Adam against the original 3DGSMCMC with Sparse Adam + AIU. (2) Similar comparisons can be found in GS3 and GS4, examining the effect of AIU on dead primitives when Sparse Adam is applied after densification ends. We observe that using AIU results in additional dead primitives. (3) Figure 3 likewise shows that adding AIU leads to more reallocated primitives during the densification stage of 3DGS-MCMC. For certain scenes, such as Kitchen, the corresponding reconstruction quality decreases, further reflecting the increased number of dead primitives. This also indicates that the phenomenon is not part of a normal optimization process; in Sec. 5.1, we relate this behavior to over-effective regularization. (4) The comparison between MC6 and MC7 in Appendix Sec. K similarly shows that adding AIU after densification produces additional dead primitives as well; note that in our application RSR is only added during densification. (5) When training MaskGaussian with AdamW-GS, MaskGaussian will not suffer from the quality risk.

## J IMPLEMENTATION DETAILS

### J.1 ADAMW-GS

Specifically, each parameter of the optimizer can be expressed in the form:

Since both opacity and scaling are strictly positive, the gradient of the L1 regularization reduces to a constant value of $+1$.

For opacity:

$$o = \sigma(\tau) = \frac{1}{1 + e^{-\tau}} \quad \nabla \sigma(\tau) = \sigma(\tau)(1 - \sigma(\tau)) \tag{18}$$

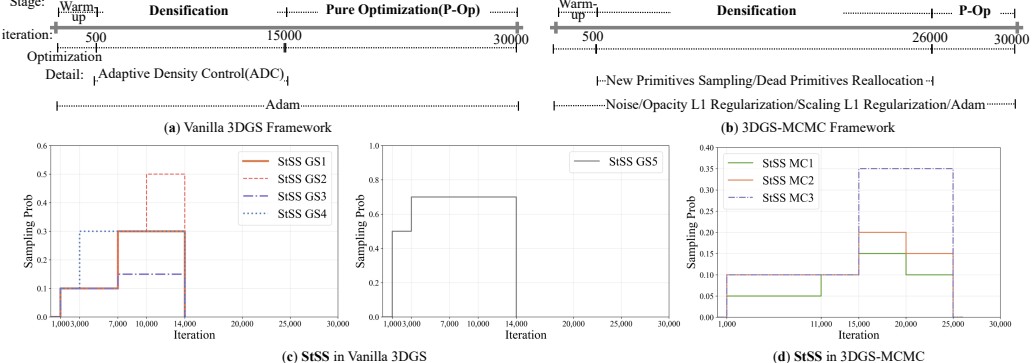

Figure 9: **a-b:**: This illustrates the training pipeline of 3DGS and 3DGS-MCMC. Since our method encourages more primitive reallocations, we postpone the densification stage by 1000 iterations compared to the original 3DGS-MCMC. In contrast, the original 3DGS-MCMC at 25k iterations produces almost no dead primitives. **c-d:** This shows the StSS used in this paper.

$$\tau_{t+1} = \tau_t - \eta_\tau \times \left[ \frac{\hat{m}(\tau)'_t}{\sqrt{\hat{v}(\tau)'_t} + \epsilon} + \min(\lambda_o \frac{\nabla\sigma(\tau)/N_I}{\sqrt{\hat{v}(\tau)'_t} + \epsilon}, \mathcal{C}_t) \right] \tag{19}$$

For scaling:

$$s = \exp(\kappa) \quad \nabla\exp(\kappa) = \exp(\kappa) \tag{20}$$

$$\kappa_{t+1} = \kappa_t - \eta_\kappa \times \left[ \frac{\hat{m}(\kappa)'_t}{\sqrt{\hat{v}(\kappa)'_t} + \epsilon} + \min(\lambda_s \frac{\nabla\exp(\kappa)/N_I}{\sqrt{\hat{v}(\kappa)'_t} + \epsilon}, \mathcal{C}_t) \right] \tag{21}$$

For position:

$$\mathcal{R}_\mu = \eta_\mu \cdot \sigma(-\lambda_\mu(o - \lambda_t)) \cdot \boldsymbol{\Sigma}\gamma \quad \gamma \sim \mathcal{N}(0, I) \tag{22}$$

$$\mu_{t+1} = \mu_t - \eta_\mu \times \left[ \frac{\hat{m}(\mu)'_t}{\sqrt{\hat{v}(\mu)'_t} + \epsilon} + \frac{\eta_{\mathcal{R}_o}}{\eta_\mu} \mathcal{R}_\mu \right] \tag{23}$$

For the other:

$$\theta_{t+1} = \theta_t - \eta_\theta \times \left[ \frac{\hat{m}(\theta)'_t}{\sqrt{\hat{v}(\theta)'_t} + \epsilon} \right] \tag{24}$$

### J.2 HYPERPARAMETER SELECTION

For vanilla 3DGS(Kerbl et al., 2023) and 3DGS-MCMC(Kheradmand et al., 2024), all pipeline parameters follow the original settings, with noise parameters identical to those in 3DGS-MCMC. Operation for the opacity is similar as (Rota Bulò et al., 2025).

$$\theta_{t+1} = \theta_t - \eta \times \left[ \frac{\hat{m}(\theta)'_t}{\sqrt{\hat{v}(\theta)'_t} + \epsilon} + \min(\lambda_\theta \frac{\nabla\mathcal{R}(\theta)/N_I}{\sqrt{\hat{v}(\theta)'_t} + \epsilon}, \mathcal{C}_t) \right] \tag{25}$$

**StSS in RSR** We fix the sampling interval at 100. Currently, StSS is manually set. StSS samples the current set of primitives, followed by RSR rescaling. The configuration of StSS must take into account the generation and reallocation behavior in the pipeline, as these processes typically introduce additional primitives with zero state. When such zero-state primitives constitute a large proportion of the population, a smaller sampling ratio is generally preferred. In the main text, Figure 3 and Figure 2 respectively show the number of reallocated primitives for 3DGS-MCMC and the evolution of the primitive count under different scenes for 3DGS. Understanding these curves provides useful guidance for designing an effective StSS strategy.

In general, the differences between these schedules arise from two main factors: (1) variations in the underlying pipelines, e.g., 3DGS versus 3DGS-MCMC, which naturally require distinct schedules; and (2) variations across scenes, where different schedules are needed to obtain the best performance on indoor versus outdoor datasets. The discrepancies introduced by the pipelines themselves are unavoidable. Nevertheless, for each pipeline we design a conservative schedule, denoted as StSSGS1

and StSSMC1, under which both 3DGS and 3DGS-MCMC outperform their original configurations. For outdoor scenes, which typically involve higher primitive generation or reallocation rates, we find that a more aggressive schedule further improves performance. All StSS schedules used in this paper are illustrated in Figure 9. Appendix Sec.K summarizes the results obtained by applying different StSS configurations across various pipelines and scenes.

**Scaling in RSR**   The three existing or designed operations provide guidance for selecting the parameters $\alpha_1$ and $\alpha_2$ in RSR. (1) The states of newly generated primitives in 3DGS, or reallocated primitives in 3DGS-MCMC, can be viewed as naturally constrained by an RSR with $\alpha_1 = \alpha_2 = 0$. (2) To properly control the magnitude of the update step $\frac{\hat{m}(\theta)'_t}{\sqrt{\hat{v}(\theta)'_t} + \epsilon}$, it is necessary to impose the relation $\alpha_2 = \alpha_1^2$. (3) A very small $\alpha_2$ is consistent with the design philosophy of DAR, as we rely on such a small value to rescale the second moment and thereby activate regularization. Therefore, when $\alpha_1$ and $\alpha_2$ are chosen to be sufficiently small while satisfying $\alpha_2 = \alpha_1^2$, the configuration remains stable. We conducted a search over different combinations of $\alpha_1$ and $\alpha_2$. For larger configurations—for example, $\alpha_1 = 0.5$, $\alpha_2 = 0.25$—RSR becomes less effective, and some combinations even lead to gradient explosion. For smaller values of $\alpha_1$ and $\alpha_2$, the performance differences are minor, and we find that $\alpha_1 = 0.2$, $\alpha_2 = 0.04$ generally performs best. In Appendix Sec.K, we report results for both $\alpha_1 = \alpha_2 = 0$ and $\alpha_1 = 0.2$, $\alpha_2 = 0.04$, cited as MC19 and MC18. **Hence, we adopt $\alpha_1 = 0.2$, $\alpha_2 = 0.04$ as the default in all experiments.**

**DAR**   We use two DAR variants: opacity and scaling regularization. Regularization is constrained to the densification stage, *as no clear benefit* is observed in P-Op. This is consistent with the smaller max magnitude of adaptive gradients in P-Op (Figure 1). Opacity regularization starts after 3000 iterations, since a large percent of generated primitives initially have zero moment, motivating a delayed start.

For the two hyperparameters, we select values by considering activation derivatives, original settings, and Adam's parameters. We set $\lambda_o = 0.001$ and, since scaling activations are two orders larger than opacity, $\lambda_s = 1e-5$. Comparative DAR experiments with $\lambda_s = 0.001$ and $\lambda_s = 1e-5$ show that $1e-5$ performs better (see experiments MC14 and MC17 in Appendix Sec. K). *In our application, only the most significant digit of $N_I$ is retained and then scaled down by one order of magnitude ($N_I/10$).*

For opacity, $C_t$ is set to 10, guided by the max magnitude of adaptive gradients in Figure 1. Without $C_t$, over-effective regularization occurs (see MC9 in Appendix Sec. K). Smaller values (e.g., 5 or 1) reduce reallocated primitives, leading to *slight* drops in PSNR and SSIM, though the MCMC framework still explores effectively. For scaling, we adopt $C_t = 10$ by analogy with opacity, which yields good results; scaling is less sensitive to $C_t$ (e.g., $C_t = 1$ shows little change in reconstruction quality; see MC8–MC17 in Appendix Sec. K).

# K   ALL RESULTS

More rendering visualization can be found in Fig.10.

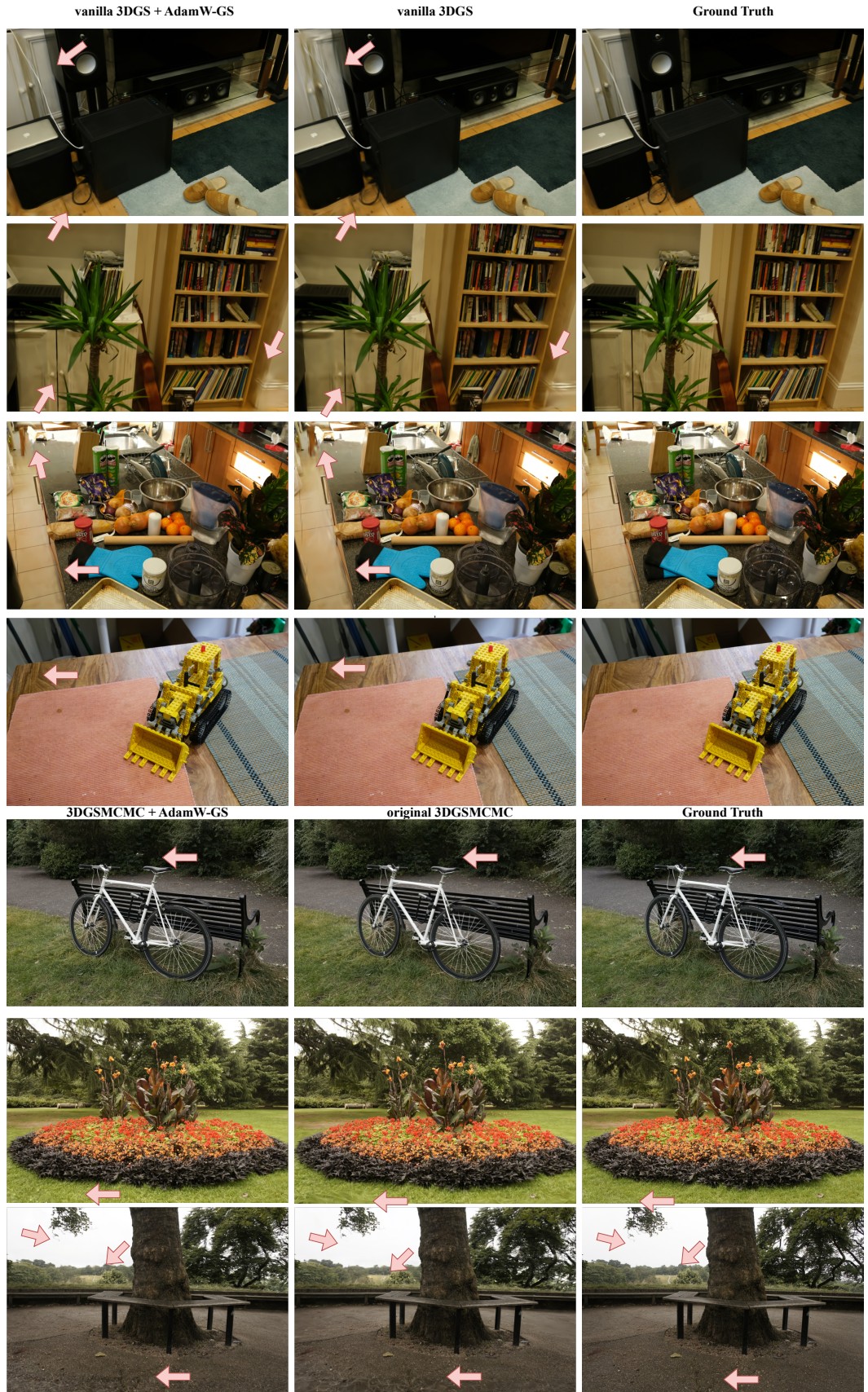

Figure 10: Rendering visualization comparison.

## K.1 BICYCLE FROM MIPNERF-360

Table 13: Quantitative results on the Bicycle from Mip-NeRF 360 with different components in the vanilla 3DGS pipeline. For AIU, the brackets denote its active iteration range (e.g., [15,30] means 15k–30k iterations). ProAIU is the sampling probability of primitives, and $\eta_{\mathrm{AIU}}$ constrains the extra update step applied to sampled primitives. $[\cdot][\cdot]$ denote the scale value and the iteration at which it takes effect. For example, [0.5,0.1][0.1,3] means $\eta_{\mathrm{AIU}} = 0.5$ after 0.1k iterations, $\eta_{\mathrm{AIU}} = 0.1$ after 3k, and $\eta_{\mathrm{AIU}} = 0$ before 0.1k. In Ours, AdamW-GS uses the same configuration but different StSS settings. Sparse denotes using Sparse Adam or Adam, and Half means using Adam in densification but Sparse Adam in P-Op. $^{\mathrm{RSR}}$ here means only adding RSR to the pipeline.

| cite | Sparse | AIU | Pro$_{\mathrm{AIU}}$ | $\eta_{\mathrm{AIU}}$ | noise | reset | Ours | PSNR | SSIM | LPIPS | $N_a$/m | $N_d$/m | $\Delta N_a$ |
|---|---|---|---|---|---|---|---|---|---|---|---|---|---|
| GS1 | x | - | - | - | - | ✓ | - | 25.238 | 0.765 | 0.211 | 5.438 | 0.548 | - |
| GS2 | ✓ | - | - | - | - | ✓ | - | 25.142 | 0.752 | 0.235 | 4.052 | 0.100 | -25.5% |
| GS3 | Half | - | - | - | - | ✓ | - | 25.260 | 0.766 | 0.211 | 5.879 | 0.109 | 8.11% |
| GS0 | ✓ | - | - | - | - | ✓ | StSSGS1$^{\mathrm{RSR}}$ | 25.337 | 0.770 | 0.213 | 3.929 | 0.038 | -28.4% |
| GS4 | Half | [15,30] | 0.5 | [0.1][15] | - | ✓ | - | 25.284 | 0.766 | 0.210 | 5.832 | 0.176 | 7.24% |
| GS15 | Half | [15,30] | 1.0 | [0.1][15] | - | ✓ | - | 25.256 | 0.766 | 0.210 | 5.651 | 0.285 | 3.91% |
| GS16 | ✓ | [0.1,15] | 0.5 | [0.1][0.1] | - | ✓ | - | 25.145 | 0.756 | 0.227 | 5.245 | 0.105 | -3.54% |
| GS17 | ✓ | [0.1,7] | 0.2 | [0.5][0.1] | - | ✓ | - | 25.166 | 0.755 | 0.228 | 5.026 | 0.103 | -7.57% |
| GS18 | ✓ | [0.1,15] | 0.2 | [0.2,0.5,0.1][0.1,1,7] | - | ✓ | - | 25.166 | 0.757 | 0.226 | 5.290 | 0.102 | -2.72% |
| GS19 | ✓ | [0.1,15] | 0.2 | [0.5,0.1][0.1,3] | - | ✓ | - | 25.133 | 0.754 | 0.229 | 4.726 | 0.097 | -13.1% |
| GS5 | ✓ | [0.1,15] | 0.2 | [0.8,0.5,0.1][0.1,3,7] | - | ✓ | - | 25.116 | 0.756 | 0.226 | 5.343 | 0.103 | -1.74% |
| GS6 | ✓ | [0.1,30] | 1.0 | [0.1][0.1] | - | ✓ | - | 25.206 | 0.760 | 0.220 | 6.087 | 0.203 | 11.9% |
| GS9 | ✓ | - | - | - | ✓ | x | StSSGS2 | 25.481 | 0.776 | 0.219 | 3.116 | 0.013 | -42.7% |
| GS7 | ✓ | - | - | - | ✓ | x | StSSGS5 | 25.470 | 0.775 | 0.223 | 3.066 | 0.011 | -43.6% |
| GS10 | ✓ | - | - | - | ✓ | ✓ | StSSGS1 | 25.514 | 0.780 | 0.215 | 2.853 | 0.021 | -47.5% |
| GS8 | ✓ | - | - | - | ✓ | ✓ | StSSGS2 | 25.454 | 0.778 | 0.219 | 2.717 | 0.018 | -50.0% |

Table 14: Quantitative results on the Bicycle from Mip-NeRF 360 with different components in the 3DGS-MCMC pipeline. AIU definitions follow the table description above (Table 13). For RSR, the sampler defaults to uniform (interval = 100) except StSS. $\mathcal{R}$ denotes the form of regularization, $\lambda$ its hyperparameter, and max corresponds to $\mathcal{C}_t$ in the paper; other parameters remain default. $^0$ in StSSMC1 denotes using $\alpha_1 = \alpha_2 = 0$ here.

| cite | Sparse | AIU | Pro$_{\mathrm{AIU}}$/$\eta_{\mathrm{AIU}}$ | RSR | $\mathcal{R}_o$ | $\lambda_o$/max$_o$ | $\mathcal{R}_s$ | $\lambda_s$/max$_s$ | PSNR | SSIM | LPIPS | $N_a$/m | $N_d$/m | $\Delta N_a$ |
|---|---|---|---|---|---|---|---|---|---|---|---|---|---|---|
| MC1 | x | - | - | - | L1 | 0.01/- | L1 | 0.01/- | 25.537 | 0.794 | 0.181 | 5.189 | 0.790 | -4.57% |
| MC2 | ✓ | - | - | - | L1 | 0.01/- | L1 | 0.01/- | 25.630 | 0.793 | 0.178 | 5.835 | 0.145 | 7.30% |
| MC3 | ✓ | [0.1,30] | 0.1/0.1 | - | L1 | 0.01/- | L1 | 0.01/- | 25.649 | 0.794 | 0.177 | 5.819 | 0.161 | 7.0% |
| MC4 | ✓ | [3,30] | 0.1/0.1 | StSSMC1 | L1 | 0.01/- | L1 | 0.01/- | 25.746 | 0.803 | 0.167 | 5.645 | 0.335 | 3.80% |
| MC5 | ✓ | [26,30] | 0.01/1 | StSSMC1 | DAR | 0.001/10 | DAR | $10^{-5}$/10 | 24.930 | 0.798 | 0.163 | 5.764 | 0.216 | 5.99% |
| MC6 | ✓ | [26,30] | 0.01/1 | StSSMC2 | DAR | 0.001/10 | DAR | $10^{-5}$/10 | 25.298 | 0.801 | 0.161 | 5.789 | 0.191 | 6.45% |
| MC18 | ✓ | - | - | StSSMC1$^0$ | L1 | 0.01/- | L1 | 0.01/- | 25.779 | 0.804 | 0.170 | 5.734 | 0.246 | 5.44% |
| MC19 | ✓ | - | - | StSSMC1 | L1 | 0.01/- | L1 | 0.01/- | 25.777 | 0.803 | 0.166 | 5.722 | 0.258 | 5.22% |
| MC20 | ✓ | - | - | StSSMC2 | L1 | 0.01/- | L1 | 0.01/- | 25.539 | 0.795 | 0.180 | 5.189 | 0.791 | -4.57% |
| MC9 | ✓ | - | - | StSSMC1 | DAR | 0.001/- | L1 | 0.01/- | 25.766 | 0.803 | 0.167 | 5.715 | 0.265 | 5.09% |
| MC10 | ✓ | - | - | StSSMC1 | DAR | 0.001/10 | L1 | 0.01/- | 25.768 | 0.804 | 0.160 | 5.840 | 0.139 | 7.4% |
| MC11 | ✓ | - | - | StSSMC2 | DAR | 0.001/10 | L1 | 0.01/- | 25.813 | 0.807 | 0.158 | 5.835 | 0.144 | 7.31% |
| MC12 | ✓ | - | - | StSSMC1 | DAR | 0.001/5 | L1 | 0.01/- | 25.753 | 0.804 | 0.159 | 5.843 | 0.137 | 7.44% |
| MC13 | ✓ | - | - | StSSMC1 | DAR | 0.001/1 | L1 | 0.01/- | 25.666 | 0.800 | 0.162 | 5.871 | 0.109 | 7.96% |
| MC14 | ✓ | - | - | StSSMC1 | DAR | 0.001/10 | DAR | 0.001/10 | 25.727 | 0.803 | 0.159 | 5.873 | 0.107 | 7.99% |
| MC15 | ✓ | - | - | StSSMC1 | DAR | 0.001/10 | DAR | 0.001/1 | 25.743 | 0.804 | 0.159 | 5.863 | 0.117 | 7.81% |
| MC16 | ✓ | - | - | StSSMC1 | DAR | 0.001/10 | DAR | $10^{-5}$/1 | 25.791 | 0.806 | 0.158 | 5.807 | 0.173 | 6.78% |
| MC17 | ✓ | - | - | StSSMC1 | DAR | 0.001/10 | DAR | $10^{-5}$/10 | 25.781 | 0.806 | 0.158 | 5.807 | 0.173 | 6.78% |
| MC7 | ✓ | - | - | StSSMC2 | DAR | 0.001/10 | DAR | $10^{-5}$/10 | 25.826 | 0.807 | 0.158 | 5.799 | 0.181 | 6.63% |
| MC8 | ✓ | - | - | StSSMC3 | DAR | 0.001/10 | DAR | $10^{-5}$/10 | 25.874 | 0.809 | 0.158 | 5.781 | 0.199 | 6.30% |

## K.2 FLOWERS FROM MIPNERF-360

Table 15: Quantitative results on the Flowers from Mip-NeRF 360 with different components in the vanilla 3DGS pipeline. For AIU, the brackets denote its active iteration range (e.g., [15,30] means 15k–30k iterations). ProAIU is the sampling probability of primitives, and $\eta_{AIU}$ constrains the extra update step applied to sampled primitives. [·][·] denote the scale value and the iteration at which it takes effect. For example, [0.5,0.1][0.1,3] means $\eta_{AIU} = 0.5$ after 0.1k iterations, $\eta_{AIU} = 0.1$ after 3k, and $\eta_{AIU} = 0$ before 0.1k. In Ours, AdamW-GS uses the same configuration but different StSS settings. Sparse denotes using Sparse Adam or Adam, and Half means using Adam in densification but Sparse Adam in P-Op. $^{RSR}$ here means only adding RSR to the pipeline.

| cite | Sparse | AIU | Pro$_{AIU}$ | $\eta_{AIU}$ | noise | reset | Ours | PSNR | SSIM | LPIPS | $N_a$/m | $N_d$/m | $\Delta N_a$ |
|---|---|---|---|---|---|---|---|---|---|---|---|---|---|
| GS1 | x | - | - | - | - | ✓ | - | 21.527 | 0.605 | 0.336 | 3.421 | 0.239 | - |
| GS2 | ✓ | - | - | - | - | ✓ | - | 21.463 | 0.598 | 0.346 | 2.737 | 0.030 | -19.9% |
| GS3 | Half | - | - | - | - | ✓ | - | 21.589 | 0.605 | 0.337 | 3.616 | 0.033 | 5.70% |
| GS0 | ✓ | - | - | - | - | ✓ | StSSGS1$^{RSR}$ | 21.698 | 0.610 | 0.336 | 2.506 | 0.011 | -26.7% |
| GS4 | Half | [15,30] | 0.5 | [0.1][15] | - | ✓ | - | 21.598 | 0.606 | 0.336 | 3.581 | 0.056 | 4.67% |
| GS15 | Half | [15,30] | 1.0 | [0.1][15] | - | ✓ | - | 21.585 | 0.605 | 0.336 | 3.518 | 0.098 | 2.83% |
| GS16 | ✓ | [0.1,15] | 0.5 | [0.1][0.1] | - | ✓ | - | 21.488 | 0.600 | 0.343 | 3.198 | 0.032 | -6.51% |
| GS17 | ✓ | [0.1,7] | 0.2 | [0.5][0.1] | - | ✓ | - | 21.535 | 0.600 | 0.342 | 3.153 | 0.032 | -7.83% |
| GS18 | ✓ | [0.1,15] | 0.2 | [0.2,0.5,0.1][0.1,1,7] | - | ✓ | - | 21.563 | 0.601 | 0.342 | 3.240 | 0.032 | -5.29% |
| GS19 | ✓ | [0.1,15] | 0.2 | [0.5,0.1][0.1,3] | - | ✓ | - | 21.524 | 0.599 | 0.344 | 2.940 | 0.029 | -14.0% |
| GS5 | ✓ | [0.1,15] | 0.2 | [0.8,0.5,0.1][0.1,3,7] | - | ✓ | - | 21.548 | 0.601 | 0.342 | 3.330 | 0.003 | -2.66% |
| GS6 | ✓ | [0.1,30] | 1.0 | [0.1][0.1] | - | ✓ | - | 21.551 | 0.601 | 0.340 | 3.599 | 0.077 | 5.20% |
| GS9 | ✓ | - | - | - | ✓ | x | StSSGS2 | 21.762 | 0.609 | 0.343 | 2.026 | 0.004 | -40.7% |
| GS7 | ✓ | - | - | - | ✓ | x | StSSGS5 | 21.718 | 0.606 | 0.349 | 1.951 | 0.003 | -42.9% |
| GS10 | ✓ | - | - | - | ✓ | ✓ | StSSGS1 | 21.708 | 0.612 | 0.339 | 1.917 | 0.007 | -43.9% |
| GS8 | ✓ | - | - | - | ✓ | ✓ | StSSGS2 | 21.711 | 0.610 | 0.344 | 1.832 | 0.005 | -46.4% |

Table 16: Quantitative results on the Flowers from Mip-NeRF 360 with different components in the 3DGS-MCMC pipeline. AIU definitions follow the table description above (Table 15). For RSR, the sampler defaults to uniform (interval = 100) except StSS. $\mathcal{R}$ denotes the form of regularization, $\lambda$ its hyperparameter, and max corresponds to $\mathcal{C}_t$ in the paper; other parameters remain default. $^0$ in StSSMC1 denotes using $\alpha_1 = \alpha_2 = 0$ here.

| cite | Sparse | AIU | Pro$_{AIU}$/$\eta_{AIU}$ | RSR | $\mathcal{R}_o$ | $\lambda_o$/max$_o$ | $\mathcal{R}_s$ | $\lambda_s$/max$_s$ | PSNR | SSIM | LPIPS | $N_a$/m | $N_d$/m | $\Delta N_a$ |
|---|---|---|---|---|---|---|---|---|---|---|---|---|---|---|
| MC1 | x | - | - | - | L1 | 0.01/- | L1 | 0.01/- | 22.028 | 0.641 | 0.296 | 3.352 | 0.248 | -2.01% |
| MC2 | ✓ | - | - | - | L1 | 0.01/- | L1 | 0.01/- | 21.966 | 0.635 | 0.304 | 3.534 | 0.066 | 3.30% |
| MC3 | ✓ | [0.1,30] | 0.1/0.1 | - | L1 | 0.01/- | L1 | 0.01/- | 22.072 | 0.638 | 0.301 | 3.529 | 0.071 | 3.15% |
| MC4 | ✓ | [3,30] | 0.1/0.1 | StSSMC1 | L1 | 0.01/- | L1 | 0.01/- | 22.039 | 0.651 | 0.274 | 3.486 | 0.114 | 0.19% |
| MC5 | ✓ | [26,30] | 0.01/1 | StSSMC1 | DAR | 0.001/10 | DAR | $10^{-5}$/10 | 21.750 | 0.650 | 0.271 | 3.520 | 0.08 | 2.89% |
| MC6 | ✓ | [26,30] | 0.01/1 | StSSMC2 | DAR | 0.001/10 | DAR | $10^{-5}$/10 | 21.511 | 0.646 | 0.275 | 3.524 | 0.076 | 3.01% |
| MC18 | ✓ | - | - | StSSMC1$^0$ | L1 | 0.01/- | L1 | 0.01/- | 22.195 | 0.649 | 0.292 | 3.510 | 0.090 | 2.60% |
| MC19 | ✓ | - | - | StSSMC1 | L1 | 0.01/- | L1 | 0.01/- | 22.038 | 0.650 | 0.277 | 3.524 | 0.076 | 3.01% |
| MC20 | ✓ | - | - | StSSMC2 | L1 | 0.01/- | L1 | 0.01/- | 22.017 | 0.641 | 0.295 | 3.352 | 0.248 | -2.01% |
| MC9 | ✓ | - | - | StSSMC1 | DAR | 0.001/- | L1 | 0.01/- | 21.288 | 0.588 | 0.349 | - | - | - |
| MC10 | ✓ | - | - | StSSMC1 | DAR | 0.001/10 | L1 | 0.01/- | 21.887 | 0.650 | 0.268 | 3.551 | 0.049 | 3.80% |
| MC11 | ✓ | - | - | StSSMC2 | DAR | 0.001/10 | L1 | 0.01/- | 22.017 | 0.656 | 0.268 | 3.548 | 0.052 | 3.71% |
| MC12 | ✓ | - | - | StSSMC1 | DAR | 0.001/5 | L1 | 0.01/- | 21.876 | 0.650 | 0.266 | 3.554 | 0.046 | 3.88% |
| MC13 | ✓ | - | - | StSSMC1 | DAR | 0.001/1 | L1 | 0.01/- | 21.726 | 0.646 | 0.264 | 3.565 | 0.035 | 4.20% |
| MC14 | ✓ | - | - | StSSMC1 | DAR | 0.001/10 | DAR | 0.001/10 | 21.923 | 0.650 | 0.269 | 3.563 | 0.037 | 4.15% |
| MC15 | ✓ | - | - | StSSMC1 | DAR | 0.001/10 | DAR | 0.001/1 | 21.962 | 0.650 | 0.270 | 3.561 | 0.039 | 4.15% |
| MC16 | ✓ | - | - | StSSMC1 | DAR | 0.001/10 | DAR | $10^{-5}$/1 | 21.936 | 0.651 | 0.271 | 3.537 | 0.063 | 3.39% |
| MC17 | ✓ | - | - | StSSMC1 | DAR | 0.001/10 | DAR | $10^{-5}$/10 | 22.034 | 0.653 | 0.270 | 3.539 | 0.061 | 3.44% |
| MC7 | ✓ | - | - | StSSMC2 | DAR | 0.001/10 | DAR | $10^{-5}$/10 | 22.003 | 0.654 | 0.267 | 3.535 | 0.065 | 3.33% |
| MC8 | ✓ | - | - | StSSMC3 | DAR | 0.001/10 | DAR | $10^{-5}$/10 | 22.108 | 0.658 | 0.268 | 3.525 | 0.075 | 3.04% |

### K.3 GARDEN FROM MIPNERF-360

Table 17: Quantitative results on the Garden from Mip-NeRF 360 with different components in the vanilla 3DGS pipeline. For AIU, the brackets denote its active iteration range (e.g., [15,30] means 15k–30k iterations). ProAIU is the sampling probability of primitives, and $\eta_{AIU}$ constrains the extra update step applied to sampled primitives. $[\cdot][\cdot]$ denote the scale value and the iteration at which it takes effect. For example, [0.5,0.1][0.1,3] means $\eta_{AIU} = 0.5$ after 0.1k iterations, $\eta_{AIU} = 0.1$ after 3k, and $\eta_{AIU} = 0$ before 0.1k. In Ours, AdamW-GS uses the same configuration but different StSS settings. . Sparse denotes using Sparse Adam or Adam, and Half means using Adam in densification but Sparse Adam in P-Op. $^{RSR}$ here means only adding RSR to the pipeline.

| cite | Sparse | AIU | Pro$_{AIU}$ | $\eta_{AIU}$ | noise | reset | Ours | PSNR | SSIM | LPIPS | $N_a$/m | $N_d$/m | $\Delta N_a$ |
|---|---|---|---|---|---|---|---|---|---|---|---|---|---|
| GS1 | x | - | - | - | - | ✓ | - | 27.334 | 0.865 | 0.108 | 5.614 | 0.318 | - |
| GS2 | ✓ | - | - | - | - | ✓ | - | 27.237 | 0.860 | 0.117 | 4.064 | 0.059 | -27.6% |
| GS3 | Half | - | - | - | - | ✓ | - | 27.409 | 0.866 | 0.107 | 5.878 | 0.059 | 4.70% |
| GS0 | ✓ | - | - | - | - | ✓ | StSSGS1$^{RSR}$ | 27.318 | 0.865 | 0.109 | 3.026 | 0.019 | -46.9% |
| GS4 | Half | [15,30] | 0.5 | [0.1][15] | - | ✓ | - | 27.436 | 0.866 | 0.107 | 5.827 | 0.112 | 3.79% |
| GS15 | Half | [15,30] | 1.0 | [0.1][15] | - | ✓ | - | 27.455 | 0.866 | 0.107 | 5.672 | 0.265 | 1.03% |
| GS16 | ✓ | [0.1,15] | 0.5 | [0.1][0.1] | - | ✓ | - | 27.268 | 0.862 | 0.114 | 5.138 | 0.068 | -8.47% |
| GS17 | ✓ | [0.1,7] | 0.2 | [0.5][0.1] | - | ✓ | - | 27.304 | 0.862 | 0.113 | 5.225 | 0.066 | -6.92% |
| GS18 | ✓ | [0.1,15] | 0.2 | [0.2,0.5,0.1][0.1,1,7] | - | ✓ | - | 27.298 | 0.862 | 0.113 | 5.363 | 0.069 | -4.47% |
| GS19 | ✓ | [0.1,15] | 0.2 | [0.5,0.1][0.1,3] | - | ✓ | - | 27.234 | 0.861 | 0.114 | 4.652 | 0.061 | -17.1% |
| GS5 | ✓ | [0.1,15] | 0.2 | [0.8,0.5,0.1][0.1,3,7] | - | ✓ | - | 27.352 | 0.863 | 0.112 | 5.548 | 0.069 | -1.17% |
| GS6 | ✓ | [0.1,30] | 1.0 | [0.1][0.1] | - | ✓ | - | 27.263 | 0.862 | 0.112 | 5.877 | 0.162 | 4.68% |
| GS9 | ✓ | - | - | - | ✓ | x | StSSGS2 | 27.438 | 0.864 | 0.123 | 2.076 | 0.006 | -63.0% |
| GS7 | ✓ | - | - | - | ✓ | x | StSSGS5 | 27.590 | 0.864 | 0.127 | 1.983 | 0.005 | -64.7% |
| GS10 | ✓ | - | - | - | ✓ | ✓ | StSSGS1 | 27.370 | 0.863 | 0.122 | 2.145 | 0.016 | -61.8% |
| GS8 | ✓ | - | - | - | ✓ | ✓ | StSSGS2 | 27.170 | 0.862 | 0.124 | 2.075 | 0.012 | -63.0% |

Table 18: Quantitative results on the Garden from Mip-NeRF 360 with different components in the 3DGS-MCMC pipeline. AIU definitions follow the table description above (Table 17). For RSR, the sampler defaults to uniform (interval = 100) except StSS. $\mathcal{R}$ denotes the form of regularization, $\lambda$ its hyperparameter, and max corresponds to $\mathcal{C}_t$ in the paper; other parameters remain default. $^0$ in StSSMC1 denotes using $\alpha_1 = \alpha_2 = 0$ here.

| cite | Sparse | AIU | Pro$_{AIU}$/$\eta_{AIU}$ | RSR | $\mathcal{R}_o$ | $\lambda_o$/max$_o$ | $\mathcal{R}_s$ | $\lambda_s$/max$_s$ | PSNR | SSIM | LPIPS | $N_a$/m | $N_d$/m | $\Delta N_a$ |
|---|---|---|---|---|---|---|---|---|---|---|---|---|---|---|
| MC1 | x | - | - | - | L1 | 0.01/- | L1 | 0.01/- | 27.862 | 0.879 | 0.094 | 5.466 | 0.433 | -2.63% |
| MC2 | ✓ | - | - | - | L1 | 0.01/- | L1 | 0.01/- | 27.889 | 0.878 | 0.096 | 5.785 | 0.115 | 3.04% |
| MC3 | ✓ | [0.1,30] | 0.1/0.1 | - | L1 | 0.01/- | L1 | 0.01/- | 27.896 | 0.878 | 0.095 | 5.770 | 0.13 | 2.77% |
| MC4 | ✓ | [3,30] | 0.1/0.1 | StSSMC1 | L1 | 0.01/- | L1 | 0.01/- | 28.032 | 0.883 | 0.089 | 5.627 | 0.273 | 0.23% |
| MC5 | ✓ | [26,30] | 0.01/1 | StSSMC1 | DAR | 0.001/10 | DAR | $10^{-5}$/10 | 25.235 | 0.864 | 0.107 | 5.645 | 0.255 | 0.55% |
| MC6 | ✓ | [26,30] | 0.01/1 | StSSMC2 | DAR | 0.001/10 | DAR | $10^{-5}$/10 | 24.993 | 0.825 | 0.149 | 5.651 | 0.249 | 0.65% |
| MC18 | ✓ | - | - | StSSMC1$^0$ | L1 | 0.01/- | L1 | 0.01/- | 28.010 | 0.882 | 0.090 | 5.726 | 0.174 | 1.99% |
| MC19 | ✓ | - | - | StSSMC1 | L1 | 0.01/- | L1 | 0.01/- | 28.028 | 0.882 | 0.089 | 5.705 | 0.195 | 1.62% |
| MC20 | ✓ | - | - | StSSMC2 | L1 | 0.01/- | L1 | 0.01/- | 27.877 | 0.879 | 0.094 | 5.714 | 0.186 | 1.78% |
| MC9 | ✓ | - | - | StSSMC1 | DAR | 0.001/- | L1 | 0.01/- | 28.037 | 0.882 | 0.093 | 5.632 | 0.268 | 0.32% |
| MC10 | ✓ | - | - | StSSMC1 | DAR | 0.001/10 | L1 | 0.01/- | 28.087 | 0.884 | 0.088 | 5.752 | 0.148 | 2.24% |
| MC11 | ✓ | - | - | StSSMC2 | DAR | 0.001/10 | L1 | 0.01/- | 28.065 | 0.884 | 0.088 | 5.740 | 0.160 | 2.24% |
| MC12 | ✓ | - | - | StSSMC1 | DAR | 0.001/5 | L1 | 0.01/- | 28.072 | 0.884 | 0.088 | 5.756 | 0.144 | 2.52% |
| MC13 | ✓ | - | - | StSSMC1 | DAR | 0.001/1 | L1 | 0.01/- | 28.021 | 0.883 | 0.089 | 5.807 | 0.093 | 3.43% |
| MC14 | ✓ | - | - | StSSMC1 | DAR | 0.001/10 | DAR | 0.001/10 | 28.045 | 0.883 | 0.088 | 5.805 | 0.095 | 3.40% |
| MC15 | ✓ | - | - | StSSMC1 | DAR | 0.001/10 | DAR | 0.001/1 | 28.018 | 0.883 | 0.088 | 5.796 | 0.104 | 3.24% |
| MC16 | ✓ | - | - | StSSMC1 | DAR | 0.001/10 | DAR | $10^{-5}$/1 | 28.081 | 0.884 | 0.088 | 5.725 | 0.175 | 1.97% |
| MC17 | ✓ | - | - | StSSMC1 | DAR | 0.001/10 | DAR | $10^{-5}$/10 | 28.104 | 0.884 | 0.088 | 5.726 | 0.174 | 1.99% |
| MC7 | ✓ | - | - | StSSMC2 | DAR | 0.001/10 | DAR | $10^{-5}$/10 | 28.109 | 0.884 | 0.088 | 5.714 | 0.186 | 1.78% |
| MC8 | ✓ | - | - | StSSMC3 | DAR | 0.001/10 | DAR | $10^{-5}$/10 | 28.144 | 0.885 | 0.088 | 5.687 | 0.213 | 1.30% |

### K.4 STUMP FROM MIPNERF-360

Table 19: Quantitative results on the Stump from Mip-NeRF 360 with different components in the vanilla 3DGS pipeline. For AIU, the brackets denote its active iteration range (e.g., [15,30] means 15k–30k iterations). ProAIU is the sampling probability of primitives, and $\eta_{\mathrm{AIU}}$ constrains the extra update step applied to sampled primitives. $[\cdot][\cdot]$ denote the scale value and the iteration at which it takes effect. For example, [0.5,0.1][0.1,3] means $\eta_{\mathrm{AIU}} = 0.5$ after 0.1k iterations, $\eta_{\mathrm{AIU}} = 0.1$ after 3k, and $\eta_{\mathrm{AIU}} = 0$ before 0.1k. In Ours, AdamW-GS uses the same configuration but different StSS settings. Sparse denotes using Sparse Adam or Adam, and Half means using Adam in densification but Sparse Adam in P-Op. $^{\mathrm{RSR}}$ here means only adding RSR to the pipeline.

| cite | Sparse | AIU | Pro$_{\mathrm{AIU}}$ | $\eta_{\mathrm{AIU}}$ | noise | reset | Ours | PSNR | SSIM | LPIPS | $N_a$/m | $N_d$/m | $\Delta N_a$ |
|---|---|---|---|---|---|---|---|---|---|---|---|---|---|
| GS1 | x | - | - | - | - | ✓ | - | 26.606 | 0.772 | 0.216 | 4.542 | 0.232 | - |
| GS2 | ✓ | - | - | - | - | ✓ | - | 26.511 | 0.762 | 0.233 | 3.751 | 0.026 | -17.4% |
| GS3 | Half | - | - | - | - | ✓ | - | 26.730 | 0.774 | 0.215 | 4.774 | 0.029 | 5.10% |
| GS0 | ✓ | - | - | - | - | ✓ | StSSGS1$^{\mathrm{RSR}}$ | 26.624 | 0.782 | 0.210 | 3.644 | 0.016 | -19.7% |
| GS4 | Half | [15,30] | 0.5 | [0.1][15] | - | ✓ | - | 26.710 | 0.773 | 0.215 | 4.928 | 0.094 | 8.49% |
| GS15 | Half | [15,30] | 1.0 | [0.1][15] | - | ✓ | - | 26.759 | 0.774 | 0.214 | 4.596 | 0.270 | 1.18% |
| GS16 | ✓ | [0.1,15] | 0.5 | [0.1][0.1] | - | ✓ | - | 26.534 | 0.766 | 0.226 | 4.974 | 0.029 | 9.51% |
| GS17 | ✓ | [0.1,7] | 0.2 | [0.5][0.1] | - | ✓ | - | 26.475 | 0.764 | 0.227 | 5.010 | 0.030 | 10.3% |
| GS18 | ✓ | [0.1,15] | 0.2 | [0.2,0.5,0.1][0.1,1,7] | - | ✓ | - | 26.611 | 0.768 | 0.224 | 4.811 | 0.026 | 5.94% |
| GS19 | ✓ | [0.1,15] | 0.2 | [0.5,0.1][0.1,3] | - | ✓ | - | 26.633 | 0.767 | 0.226 | 4.830 | 0.026 | 6.34% |
| GS5 | ✓ | [0.1,15] | 0.2 | [0.8,0.5,0.1][0.1,3,7] | - | ✓ | - | 26.630 | 0.767 | 0.224 | 5.165 | 0.026 | 13.7% |
| GS6 | ✓ | [0.1,30] | 1.0 | [0.1][0.1] | - | ✓ | - | 26.601 | 0.768 | 0.222 | 5.274 | 0.125 | 16.1% |
| GS9 | ✓ | - | - | - | ✓ | x | StSSGS2 | 27.175 | 0.796 | 0.211 | 2.894 | 0.014 | -36.3% |
| GS7 | ✓ | - | - | - | ✓ | x | StSSGS5 | 27.271 | 0.798 | 0.211 | 2.677 | 0.009 | -41.0% |
| GS10 | ✓ | - | - | - | ✓ | ✓ | StSSGS1 | 27.033 | 0.800 | 0.206 | 2.567 | 0.023 | -43.5% |
| GS8 | ✓ | - | - | - | ✓ | ✓ | StSSGS2 | 27.045 | 0.800 | 0.206 | 2.470 | 0.020 | -45.6% |

Table 20: Quantitative results on the Stump from Mip-NeRF 360 with different components in the 3DGS-MCMC pipeline. AIU definitions follow the table description above (Table 19). For RSR, the sampler defaults to uniform (interval = 100) except StSS. $\mathcal{R}$ denotes the form of regularization, $\lambda$ its hyperparameter, and max corresponds to $\mathcal{C}_t$ in the paper; other parameters remain default. $^0$ in StSSMC1 denotes using $\alpha_1 = \alpha_2 = 0$ here.

| cite | Sparse | AIU | Pro$_{\mathrm{AIU}}$/$\eta_{\mathrm{AIU}}$ | RSR | $\mathcal{R}_o$ | $\lambda_o$/max$_o$ | $\mathcal{R}_s$ | $\lambda_s$/max$_s$ | PSNR | SSIM | LPIPS | $N_a$/m | $N_d$/m | $\Delta N_a$ |
|---|---|---|---|---|---|---|---|---|---|---|---|---|---|---|
| MC1 | x | - | - | - | L1 | 0.01/- | L1 | 0.01/- | 27.204 | 0.805 | 0.183 | 4.245 | 0.555 | -6.53% |
| MC2 | ✓ | - | - | - | L1 | 0.01/- | L1 | 0.01/- | 27.293 | 0.805 | 0.182 | 4.718 | 0.082 | 3.84% |
| MC3 | ✓ | [0.1,30] | 0.1/0.1 | - | L1 | 0.01/- | L1 | 0.01/- | 27.273 | 0.805 | 0.182 | 4.698 | 0.102 | 3.43% |
| MC4 | ✓ | [3,30] | 0.1/0.1 | StSSMC1 | L1 | 0.01/- | L1 | 0.01/- | 27.131 | 0.806 | 0.177 | 4.535 | 0.265 | -0.154% |
| MC5 | ✓ | [26,30] | 0.01/1 | StSSMC1 | DAR | 0.001/10 | DAR | $10^{-5}$/10 | 20.854 | 0.687 | 0.284 | 4.621 | 0.179 | 1.73% |
| MC6 | ✓ | [26,30] | 0.01/1 | StSSMC2 | DAR | 0.001/10 | DAR | $10^{-5}$/10 | 25.509 | 0.782 | 0.195 | 4.615 | 0.185 | 1.60% |
| MC18 | ✓ | - | - | StSSMC1$^0$ | L1 | 0.01/- | L1 | 0.01/- | 27.064 | 0.805 | 0.179 | 4.620 | 0.180 | 1.71% |
| MC19 | ✓ | - | - | StSSMC1 | L1 | 0.01/- | L1 | 0.01/- | 27.172 | 0.807 | 0.176 | 4.631 | 0.169 | 1.93% |
| MC20 | ✓ | - | - | StSSMC2 | L1 | 0.01/- | L1 | 0.01/- | 27.170 | 0.796 | 0.190 | 4.673 | 0.127 | 2.88% |
| MC9 | ✓ | - | - | StSSMC1 | DAR | 0.001/- | L1 | 0.01/- | 26.943 | 0.796 | 0.191 | 4.620 | 0.180 | 1.71% |
| MC10 | ✓ | - | - | StSSMC1 | DAR | 0.001/10 | L1 | 0.01/- | 27.041 | 0.800 | 0.178 | 4.713 | 0.087 | 3.76% |
| MC11 | ✓ | - | - | StSSMC2 | DAR | 0.001/10 | L1 | 0.01/- | 27.064 | 0.803 | 0.176 | 4.705 | 0.095 | 3.58% |
| MC12 | ✓ | - | - | StSSMC1 | DAR | 0.001/5 | L1 | 0.01/- | 27.070 | 0.799 | 0.179 | 4.712 | 0.088 | 3.74% |
| MC13 | ✓ | - | - | StSSMC1 | DAR | 0.001/1 | L1 | 0.01/- | 26.853 | 0.793 | 0.184 | 4.741 | 0.059 | 4.38% |
| MC14 | ✓ | - | - | StSSMC1 | DAR | 0.001/10 | DAR | 0.001/10 | 26.966 | 0.797 | 0.180 | 4.724 | 0.076 | 4.00% |
| MC15 | ✓ | - | - | StSSMC1 | DAR | 0.001/10 | DAR | 0.001/1 | 27.108 | 0.801 | 0.177 | 4.717 | 0.083 | 3.85% |
| MC16 | ✓ | - | - | StSSMC1 | DAR | 0.001/10 | DAR | $10^{-5}$/1 | 27.088 | 0.801 | 0.178 | 4.683 | 0.117 | 3.10% |
| MC17 | ✓ | - | - | StSSMC1 | DAR | 0.001/10 | DAR | $10^{-5}$/10 | 27.101 | 0.802 | 0.178 | 4.681 | 0.119 | 3.06% |
| MC7 | ✓ | - | - | StSSMC2 | DAR | 0.001/10 | DAR | $10^{-5}$/10 | 27.159 | 0.805 | 0.175 | 4.673 | 0.127 | 2.88% |
| MC8 | ✓ | - | - | StSSMC3 | DAR | 0.001/10 | DAR | $10^{-5}$/10 | 27.245 | 0.808 | 0.174 | 4.650 | 0.150 | 2.37% |

## K.5 TREEHILL FROM MIPNERF-360

Table 21: Quantitative results on the TreeHill from Mip-NeRF 360 with different components in the vanilla 3DGS pipeline. For AIU, the brackets denote its active iteration range (e.g., [15,30] means 15k–30k iterations). ProAIU is the sampling probability of primitives, and $\eta_{\mathrm{AIU}}$ constrains the extra update step applied to sampled primitives. $[\cdot][\cdot]$ denote the scale value and the iteration at which it takes effect. For example, [0.5,0.1][0.1,3] means $\eta_{\mathrm{AIU}} = 0.5$ after 0.1k iterations, $\eta_{\mathrm{AIU}} = 0.1$ after 3k, and $\eta_{\mathrm{AIU}} = 0$ before 0.1k. In Ours, AdamW-GS uses the same configuration but different StSS settings. Sparse denotes using Sparse Adam or Adam, and Half means using Adam in densification but Sparse Adam in P-Op. $^{\mathrm{RSR}}$ here means only adding RSR to the pipeline.

| cite | Sparse | AIU | Pro$_{\mathrm{AIU}}$ | $\eta_{\mathrm{AIU}}$ | noise | reset | Ours | PSNR | SSIM | LPIPS | $N_a$/m | $N_d$/m | $\Delta N_a$ |
|---|---|---|---|---|---|---|---|---|---|---|---|---|---|
| GS1 | x | - | - | - | - | ✓ | - | 22.534 | 0.633 | 0.326 | 3.547 | 0.228 | - |
| GS2 | ✓ | - | - | - | - | ✓ | - | 22.477 | 0.629 | 0.340 | 2.701 | 0.027 | -23.8% |
| GS3 | Half | - | - | - | - | ✓ | - | 22.558 | 0.634 | 0.327 | 3.820 | 0.031 | 7.69% |
| GS0 | ✓ | - | - | - | - | ✓ | StSSGS1$^{\mathrm{RSR}}$ | 22.447 | 0.638 | 0.320 | 2.887 | 0.016 | -18.6% |
| GS4 | Half | [15,30] | 0.5 | [0.1][15] | - | ✓ | - | 22.538 | 0.633 | 0.327 | 3.732 | 0.052 | 5.21% |
| GS15 | Half | [15,30] | 1.0 | [0.1][15] | - | ✓ | - | 22.577 | 0.634 | 0.325 | 3.726 | 0.083 | 5.04% |
| GS16 | ✓ | [0.1,15] | 0.5 | [0.1][0.1] | - | ✓ | - | 22.480 | 0.630 | 0.335 | 3.488 | 0.030 | -1.66% |
| GS17 | ✓ | [0.1,7] | 0.2 | [0.5][0.1] | - | ✓ | - | 22.633 | 0.631 | 0.335 | 3.304 | 0.029 | -6.85% |
| GS18 | ✓ | [0.1,15] | 0.2 | [0.2,0.5,0.1][0.1,1,7] | - | ✓ | - | 22.509 | 0.631 | 0.334 | 3.490 | 0.030 | -1.60% |
| GS19 | ✓ | [0.1,15] | 0.2 | [0.5,0.1][0.1,3] | - | ✓ | - | 22.537 | 0.630 | 0.335 | 3.131 | 0.026 | -11.7% |
| GS5 | ✓ | [0.1,15] | 0.2 | [0.8,0.5,0.1][0.1,3,7] | - | ✓ | - | 22.558 | 0.631 | 0.334 | 3.553 | 0.029 | 0.169% |
| GS6 | ✓ | [0.1,30] | 1.0 | [0.1][0.1] | - | ✓ | - | 22.484 | 0.631 | 0.331 | 4.074 | 0.080 | 14.8% |
| GS9 | ✓ | - | - | - | ✓ | x | StSSGS2 | 22.791 | 0.646 | 0.322 | 2.616 | 0.003 | -26.2% |
| GS7 | ✓ | - | - | - | ✓ | x | StSSGS5 | 22.691 | 0.644 | 0.328 | 2.553 | 0.002 | -28.0% |
| GS10 | ✓ | - | - | - | ✓ | ✓ | StSSGS1 | 22.905 | 0.650 | 0.322 | 2.335 | 0.006 | -34.1% |
| GS8 | ✓ | - | - | - | ✓ | ✓ | StSSGS2 | 22.891 | 0.651 | 0.325 | 2.231 | 0.005 | -37.1% |

Table 22: Quantitative results on the TreeHill from Mip-NeRF 360 with different components in the 3DGS-MCMC pipeline. AIU definitions follow the table description above (Table 21). For RSR, the sampler defaults to uniform (interval = 100) except StSS. $\mathcal{R}$ denotes the form of regularization, $\lambda$ its hyperparameter, and max corresponds to $\mathcal{C}_t$ in the paper; other parameters remain default. $^0$ in StSSMC1 denotes using $\alpha_1 = \alpha_2 = 0$ here.

| cite | Sparse | AIU | Pro$_{\mathrm{AIU}}$/$\eta_{\mathrm{AIU}}$ | RSR | $\mathcal{R}_o$ | $\lambda_o$/max$_o$ | $\mathcal{R}_s$ | $\lambda_s$/max$_s$ | PSNR | SSIM | LPIPS | $N_a$/m | $N_d$/m | $\Delta N_a$ |
|---|---|---|---|---|---|---|---|---|---|---|---|---|---|---|
| MC1 | x | - | - | - | L1 | 0.01/- | L1 | 0.01/- | 22.894 | 0.655 | 0.310 | 3.329 | 0.37 | -6.14% |
| MC2 | ✓ | - | - | - | L1 | 0.01/- | L1 | 0.01/- | 23.023 | 0.659 | 0.300 | 3.621 | 0.079 | 2.08% |
| MC3 | ✓ | [0.1,30] | 0.1/0.1 | - | L1 | 0.01/- | L1 | 0.01/- | 23.012 | 0.658 | 0.302 | 3.612 | 0.088 | 1.83% |
| MC4 | ✓ | [3,30] | 0.1/0.1 | StSSMC1 | L1 | 0.01/- | L1 | 0.01/- | 22.858 | 0.663 | 0.284 | 3.555 | 0.145 | 0.22% |
| MC5 | ✓ | [26,30] | 0.01/1 | StSSMC1 | DAR | 0.001/10 | DAR | $10^{-5}$/10 | 22.788 | 0.659 | 0.270 | 3.616 | 0.084 | 1.94% |
| MC6 | ✓ | [26,30] | 0.01/1 | StSSMC2 | DAR | 0.001/10 | DAR | $10^{-5}$/10 | 22.715 | 0.660 | 0.268 | 3.622 | 0.078 | 2.11% |
| MC18 | ✓ | - | - | StSSMC1$^0$ | L1 | 0.01/- | L1 | 0.01/- | 22.930 | 0.663 | 0.291 | 3.591 | 0.109 | 1.21% |
| MC19 | ✓ | - | - | StSSMC1 | L1 | 0.01/- | L1 | 0.01/- | 22.918 | 0.663 | 0.285 | 3.580 | 0.120 | 0.93% |
| MC20 | ✓ | - | - | StSSMC2 | L1 | 0.01/- | L1 | 0.01/- | 22.953 | 0.654 | 0.312 | 3.323 | 0.377 | -6.32% |
| MC9 | ✓ | - | - | StSSMC1 | DAR | 0.001/- | L1 | 0.01/- | - | - | - | - | - | - |
| MC10 | ✓ | - | - | StSSMC1 | DAR | 0.001/10 | L1 | 0.01/- | 22.771 | 0.657 | 0.270 | 3.653 | 0.047 | 2.98% |
| MC11 | ✓ | - | - | StSSMC2 | DAR | 0.001/10 | L1 | 0.01/- | 22.740 | 0.660 | 0.268 | 3.654 | 0.046 | 3.01% |
| MC12 | ✓ | - | - | StSSMC1 | DAR | 0.001/5 | L1 | 0.01/- | 22.684 | 0.655 | 0.271 | 3.654 | 0.046 | 3.01% |
| MC13 | ✓ | - | - | StSSMC1 | DAR | 0.001/1 | L1 | 0.01/- | 22.533 | 0.652 | 0.274 | 3.663 | 0.037 | 3.27% |
| MC14 | ✓ | - | - | StSSMC1 | DAR | 0.001/10 | DAR | 0.001/10 | 22.690 | 0.655 | 0.270 | 3.663 | 0.037 | 3.27% |
| MC15 | ✓ | - | - | StSSMC1 | DAR | 0.001/10 | DAR | 0.001/1 | 22.692 | 0.656 | 0.270 | 3.661 | 0.039 | 3.21% |
| MC16 | ✓ | - | - | StSSMC1 | DAR | 0.001/10 | DAR | $10^{-5}$/1 | 22.800 | 0.658 | 0.270 | 3.638 | 0.062 | 2.56% |
| MC17 | ✓ | - | - | StSSMC1 | DAR | 0.001/10 | DAR | $10^{-5}$/10 | 22.806 | 0.658 | 0.270 | 3.639 | 0.061 | 2.59% |
| MC7 | ✓ | - | - | StSSMC2 | DAR | 0.001/10 | DAR | $10^{-5}$/10 | 22.836 | 0.661 | 0.268 | 3.637 | 0.063 | 2.53% |
| MC8 | ✓ | - | - | StSSMC3 | DAR | 0.001/10 | DAR | $10^{-5}$/10 | 22.866 | 0.662 | 0.265 | 3.630 | 0.07 | 2.34% |

## K.6 ROOM FROM MIPNERF-360

Table 23: Quantitative results on the Room from Mip-NeRF 360 with different components in the vanilla 3DGS pipeline. For AIU, the brackets denote its active iteration range (e.g., [15,30] means 15k–30k iterations). ProAIU is the sampling probability of primitives, and $\eta_{\mathrm{AIU}}$ constrains the extra update step applied to sampled primitives. [·][·] denote the scale value and the iteration at which it takes effect. For example, [0.5,0.1][0.1,3] means $\eta_{\mathrm{AIU}} = 0.5$ after 0.1k iterations, $\eta_{\mathrm{AIU}} = 0.1$ after 3k, and $\eta_{\mathrm{AIU}} = 0$ before 0.1k. In Ours, AdamW-GS uses the same configuration but different StSS settings. Sparse denotes using Sparse Adam or Adam, and Half means using Adam in densification but Sparse Adam in P-Op. [RSR] here means only adding RSR to the pipeline.

| cite | Sparse | AIU | Pro$_{\mathrm{AIU}}$ | $\eta_{\mathrm{AIU}}$ | noise | Ours | PSNR | SSIM | LPIPS | $N_a$/m | $N_d$/m | $\Delta N_a$ |
|---|---|---|---|---|---|---|---|---|---|---|---|---|
| GS1 | x | - | - | - | - | - | 31.500 | 0.920 | 0.221 | 1.374 | 0.202 | - |
| GS2 | ✓ | - | - | - | - | - | 31.096 | 0.917 | 0.231 | 1.131 | 0.031 | -17.6% |
| GS3 | Half | - | - | - | - | - | 31.607 | 0.921 | 0.220 | 1.537 | 0.041 | 11.8% |
| GS0 | ✓ | - | - | - | - | StSSGS1$^{\mathrm{RSR}}$ | 31.541 | 0.921 | 0.220 | 0.994 | 0.017 | -28.8% |
| GS4 | Half | [15,30] | 0.5 | [0.1][15] | - | - | 31.719 | 0.921 | 0.220 | 1.496 | 0.111 | 8.87% |
| GS15 | Half | [15,30] | 1.0 | [0.1][15] | - | - | 31.614 | 0.921 | 0.220 | 1.466 | 0.113 | 6.69% |
| GS16 | ✓ | [0.1,15] | 0.5 | [0.1][0.1] | - | - | 31.389 | 0.919 | 0.228 | 1.447 | 0.034 | 5.31% |
| GS17 | ✓ | [0.1,7] | 0.2 | [0.5][0.1] | - | - | 31.539 | 0.919 | 0.227 | 1.441 | 0.036 | 2.25% |
| GS18 | ✓ | [0.1,15] | 0.2 | [0.2,0.5,0.1] [0.1,1,7] | - | - | 31.379 | 0.918 | 0.227 | 1.473 | 0.037 | 7.20% |
| GS19 | ✓ | [0.1,15] | 0.2 | [0.5,0.1] [0.1,3] | - | - | 31.476 | 0.919 | 0.228 | 1.342 | 0.034 | -2.32% |
| GS5 | ✓ | [0.1,15] | 0.2 | [0.8,0.5,0.1] [0.1,3,7] | - | - | 31.509 | 0.919 | 0.226 | 1.504 | 0.036 | 9.46% |
| GS6 | ✓ | [0.1,30] | 1.0 | [0.1][0.1] | - | - | 31.259 | 0.917 | 0.228 | 1.549 | 0.066 | 12.7% |
| GS13 | ✓ | - | - | - | - | StSSGS3 | 31.768 | 0.920 | 0.227 | 0.617 | 0.006 | -55.1% |
| GS7/GS8 | ✓ | - | - | - | - | StSSGS1 | 31.741 | 0.921 | 0.222 | 0.624 | 0.005 | -54.6% |
| GS14 | ✓ | - | - | - | - | StSSGS4 | 31.753 | 0.920 | 0.222 | 0.632 | 0.005 | -54.0% |

Table 24: Quantitative results on the Room from Mip-NeRF 360 with different components in the 3DGS-MCMC pipeline. AIU definitions follow the table description above (Table 23). For RSR, the sampler defaults to uniform (interval = 100) except StSS. $\mathcal{R}$ denotes the form of regularization, $\lambda$ its hyperparameter, and max corresponds to $\mathcal{C}_t$ in the paper; other parameters remain default. [0] in StSSMC1 denotes using $\alpha_1 = \alpha_2 = 0$ here.

| cite | Sparse | AIU | Pro$_{\mathrm{AIU}}$/$\eta_{\mathrm{AIU}}$ | RSR | $\mathcal{R}_o$ | $\lambda_o$/max$_o$ | $\mathcal{R}_s$ | $\lambda_s$/max$_s$ | PSNR | SSIM | LPIPS | $N_a$/m | $N_d$/m | $\Delta N_a$ |
|---|---|---|---|---|---|---|---|---|---|---|---|---|---|---|
| MC1 | x | - | - | - | L1 | 0.01/- | L1 | 0.01/- | 32.034 | 0.927 | 0.210 | 1.320 | 0.250 | -3.93% |
| MC2 | ✓ | - | - | - | L1 | 0.01/- | L1 | 0.01/- | 32.417 | 0.929 | 0.209 | 1.472 | 0.098 | 7.13% |
| MC3 | ✓ | [0.1,30] | 0.1/0.1 | - | L1 | 0.01/- | L1 | 0.01/- | 32.493 | 0.930 | 0.207 | 1.438 | 0.132 | 4.65% |
| MC4 | ✓ | [3,30] | 0.1/0.1 | StSSMC1 | L1 | 0.01/- | L1 | 0.01/- | 32.498 | 0.930 | 0.204 | 1.351 | 0.219 | -1.67% |
| MC6 | ✓ | [26,30] | 0.01/1 | StSSMC1 | DAR | 0.001/10 | DAR | $10^{-5}$/10 | 19.068 | 0.734 | 0.387 | 1.505 | 0.065 | 9.53% |
| MC18 | ✓ | - | - | StSSMC1$^0$ | L1 | 0.01/- | L1 | 0.01/- | 32.328 | 0.929 | 0.207 | 1.428 | 0.142 | 3.93% |
| MC19 | ✓ | - | - | StSSMC1 | L1 | 0.01/- | L1 | 0.01/- | 32.514 | 0.930 | 0.205 | 1.430 | 0.140 | 4.07% |
| MC20 | ✓ | - | - | StSSMC2 | L1 | 0.01/- | L1 | 0.01/- | 31.179 | 0.922 | 0.213 | 1.126 | 0.444 | -18.4% |
| MC9 | ✓ | - | - | StSSMC1 | DAR | 0.001/- | L1 | 0.01/- | 32.498 | 0.930 | 0.204 | 1.351 | 0.219 | -1.67% |
| MC10 | ✓ | - | - | StSSMC1 | DAR | 0.001/10 | L1 | 0.01/- | 32.600 | 0.933 | 0.196 | 1.536 | 0.034 | 11.7% |
| MC11 | ✓ | - | - | StSSMC2 | DAR | 0.001/10 | L1 | 0.01/- | 32.542 | 0.933 | 0.197 | 1.535 | 0.035 | 11.7% |
| MC12 | ✓ | - | - | StSSMC1 | DAR | 0.001/5 | L1 | 0.01/- | 32.611 | 0.933 | 0.197 | 1.536 | 0.034 | 11.7% |
| MC13 | ✓ | - | - | StSSMC1 | DAR | 0.001/1 | L1 | 0.01/- | 32.317 | 0.932 | 0.197 | 1.546 | 0.024 | 12.5% |
| MC14 | ✓ | - | - | StSSMC1 | DAR | 0.001/10 | DAR | 0.001/10 | 32.547 | 0.933 | 0.195 | 1.528 | 0.042 | 11.2% |
| MC15 | ✓ | - | - | StSSMC1 | DAR | 0.001/10 | DAR | 0.001/1 | 32.586 | 0.933 | 0.196 | 1.528 | 0.042 | 11.2% |
| MC16 | ✓ | - | - | StSSMC1 | DAR | 0.001/10 | DAR | $10^{-5}$/1 | 32.647 | 0.933 | 0.197 | 1.515 | 0.055 | 10.2% |
| MC8/MC7 | ✓ | - | - | StSSMC1 | DAR | 0.001/10 | DAR | $10^{-5}$/10 | 32.651 | 0.933 | 0.196 | 1.515 | 0.055 | 10.2% |
| MC21 | ✓ | - | - | StSSMC2 | DAR | 0.001/10 | DAR | $10^{-5}$/10 | 32.647 | 0.933 | 0.197 | 1.512 | 0.058 | 10.0% |

## K.7 COUNTER FROM MIPNERF-360

Table 25: Quantitative results on the Counter from Mip-NeRF 360 with different components in the vanilla 3DGS pipeline. For AIU, the brackets denote its active iteration range (e.g., [15,30] means 15k–30k iterations). ProAIU is the sampling probability of primitives, and $\eta_{\text{AIU}}$ constrains the extra update step applied to sampled primitives. [·][·] denote the scale value and the iteration at which it takes effect. For example, [0.5,0.1][0.1,3] means $\eta_{\text{AIU}} = 0.5$ after 0.1k iterations, $\eta_{\text{AIU}} = 0.1$ after 3k, and $\eta_{\text{AIU}} = 0$ before 0.1k. In Ours, AdamW-GS uses the same configuration but different StSS settings. Sparse denotes using Sparse Adam or Adam, and Half means using Adam in densification but Sparse Adam in P-Op. [RSR] here means only adding RSR to the pipeline.

| cite | Sparse | AIU | Pro$_{\text{AIU}}$ | $\eta_{\text{AIU}}$ | noise | Ours | PSNR | SSIM | LPIPS | $N_a$/m | $N_d$/m | $\Delta N_a$ |
|---|---|---|---|---|---|---|---|---|---|---|---|---|
| GS1 | x | - | - | - | - | - | 29.046 | 0.909 | 0.201 | 1.092 | 0.097 | - |
| GS2 | ✓ | - | - | - | - | - | 28.979 | 0.907 | 0.205 | 1.005 | 0.034 | -7.96% |
| GS3 | Half | - | - | - | - | - | 29.065 | 0.909 | 0.201 | 1.115 | 0.076 | 2.10% |
| GS0 | ✓ | - | - | - | - | StSSGS1$^{\text{RSR}}$ | 29.026 | 0.909 | 0.202 | 0.768 | 0.011 | -30.6% |
| GS4 | Half | [15,30] | 0.5 | [0.1][15] | - | - | 29.060 | 0.909 | 0.201 | 1.124 | 0.067 | 2.93% |
| GS15 | Half | [15,30] | 1.0 | [0.1][15] | - | - | 29.079 | 0.909 | 0.201 | 1.067 | 0.116 | -2.28% |
| GS16 | ✓ | [0.1,15] | 0.5 | [0.1][0.1] | - | - | 28.947 | 0.908 | 0.204 | 1.163 | 0.035 | 6.50% |
| GS17 | ✓ | [0.1,7] | 0.2 | [0.5][0.1] | - | - | 28.974 | 0.908 | 0.204 | 1.175 | 0.036 | 7.60% |
| GS18 | ✓ | [0.1,15] | 0.2 | [0.2,0.5,0.1][0.1,1,7] | - | - | 29.010 | 0.908 | 0.203 | 1.199 | 0.036 | 9.97% |
| GS19 | ✓ | [0.1,15] | 0.2 | [0.5,0.1][0.1,3] | - | - | 28.972 | 0.908 | 0.203 | 1.118 | 0.033 | 2.38% |
| GS5 | ✓ | [0.1,15] | 0.2 | [0.8,0.5,0.1][0.1,3,7] | - | - | 28.998 | 0.908 | 0.202 | 1.221 | 0.036 | 11.81% |
| GS6 | ✓ | [0.1,30] | 1.0 | [0.1][0.1] | - | - | 28.974 | 0.908 | 0.202 | 1.257 | 0.058 | 15.1% |
| GS13 | ✓ | - | - | - | - | StSSGS3 | 29.084 | 0.907 | 0.204 | 0.579 | 0.005 | -46.9% |
| GS7/GS8 | ✓ | - | - | - | - | StSSGS1 | 29.112 | 0.907 | 0.206 | 0.535 | 0.004 | -51.0% |
| GS14 | ✓ | - | - | - | - | StSSGS4 | 29.077 | 0.907 | 0.203 | 0.535 | 0.003 | -51.0% |

Table 26: Quantitative results on the Counter from Mip-NeRF 360 with different components in the 3DGS-MCMC pipeline. AIU definitions follow the table description above (Table 25). For RSR, the sampler defaults to uniform (interval = 100) except StSS. $\mathcal{R}$ denotes the form of regularization, $\lambda$ its hyperparameter, and max corresponds to $\mathcal{C}_t$ in the paper; other parameters remain default. $^0$ in StSSMC1 denotes using $\alpha_1 = \alpha_2 = 0$ here.

| cite | Sparse | AIU | Pro$_{\text{AIU}}$/$\eta_{\text{AIU}}$ | RSR | $\mathcal{R}_o$ | $\lambda_o$/max$_o$ | $\mathcal{R}_s$ | $\lambda_s$/max$_s$ | PSNR | SSIM | LPIPS | $N_a$/m | $N_d$/m | $\Delta N_a$ |
|---|---|---|---|---|---|---|---|---|---|---|---|---|---|---|
| MC1 | x | - | - | - | L1 | 0.01/- | L1 | 0.01/- | 29.229 | 0.914 | 0.195 | 1.084 | 0.106 | -0.73% |
| MC2 | ✓ | - | - | - | L1 | 0.01/- | L1 | 0.01/- | 29.180 | 0.912 | 0.198 | 1.148 | 0.042 | 5.12% |
| MC3 | ✓ | [0.1,30] | 0.1/0.1 | - | L1 | 0.01/- | L1 | 0.01/- | 29.194 | 0.914 | 0.196 | 1.140 | 0.05 | 4.39% |
| MC4 | ✓ | [3,30] | 0.1/0.1 | StSSMC1 | L1 | 0.01/- | L1 | 0.01/- | 29.345 | 0.916 | 0.190 | 1.098 | 0.092 | 0.54% |
| MC6 | ✓ | [26,30] | 0.01/1 | StSSMC1 | DAR | 0.001/10 | DAR | $10^{-5}$/10 | 28.791 | 0.916 | 0.186 | 1.148 | 0.042 | 5.12% |
| MC18 | ✓ | - | - | StSSMC1$^0$ | L1 | 0.01/- | L1 | 0.01/- | 29.274 | 0.914 | 0.196 | 1.160 | 0.03 | 6.22% |
| MC19 | ✓ | - | - | StSSMC1 | L1 | 0.01/- | L1 | 0.01/- | 29.291 | 0.914 | 0.191 | 1.151 | 0.039 | 5.40% |
| MC20 | ✓ | - | - | StSSMC2 | L1 | 0.01/- | L1 | 0.01/- | 29.069 | 0.911 | 0.199 | 0.967 | 0.223 | -11.4% |
| MC9 | ✓ | - | - | StSSMC1 | DAR | 0.001/- | L1 | 0.01/- | 29.075 | 0.910 | 0.202 | 1.13 | 0.060 | 3.47% |
| MC10 | ✓ | - | - | StSSMC1 | DAR | 0.001/10 | L1 | 0.01/- | 29.476 | 0.919 | 0.184 | 1.172 | 0.018 | 7.32% |
| MC11 | ✓ | - | - | StSSMC2 | DAR | 0.001/10 | L1 | 0.01/- | 29.495 | 0.919 | 0.183 | 1.171 | 0.019 | 7.23% |
| MC12 | ✓ | - | - | StSSMC1 | DAR | 0.001/5 | L1 | 0.01/- | 29.470 | 0.919 | 0.184 | 1.143 | 0.047 | 4.67% |
| MC13 | ✓ | - | - | StSSMC1 | DAR | 0.001/1 | L1 | 0.01/- | 29.450 | 0.919 | 0.183 | 1.181 | 0.009 | 8.15% |
| MC14 | ✓ | - | - | StSSMC1 | DAR | 0.001/10 | DAR | 0.001/10 | 29.455 | 0.919 | 0.182 | 1.172 | 0.018 | 7.32% |
| MC15 | ✓ | - | - | StSSMC1 | DAR | 0.001/10 | DAR | 0.001/1 | 29.486 | 0.919 | 0.183 | 1.171 | 0.019 | 7.23% |
| MC16 | ✓ | - | - | StSSMC1 | DAR | 0.001/10 | DAR | $10^{-5}$/1 | 29.475 | 0.919 | 0.184 | 1.163 | 0.027 | 6.50% |
| MC8/MC7 | ✓ | - | - | StSSMC1 | DAR | 0.001/10 | DAR | $10^{-5}$/10 | 29.515 | 0.919 | 0.183 | 1.163 | 0.027 | 6.50% |
| MC21 | ✓ | - | - | StSSMC2 | DAR | 0.001/10 | DAR | $10^{-5}$/10 | 29.506 | 0.919 | 0.184 | 1.161 | 0.029 | 6.31% |

## K.8 KITCHEN FROM MIPNERF-360

Table 27: Quantitative results on the Kitchen from Mip-NeRF 360 with different components in the vanilla 3DGS pipeline. For AIU, the brackets denote its active iteration range (e.g., [15,30] means 15k–30k iterations). ProAIU is the sampling probability of primitives, and $\eta_{\mathrm{AIU}}$ constrains the extra update step applied to sampled primitives. [·][·] denote the scale value and the iteration at which it takes effect. For example, [0.5,0.1][0.1,3] means $\eta_{\mathrm{AIU}} = 0.5$ after 0.1k iterations, $\eta_{\mathrm{AIU}} = 0.1$ after 3k, and $\eta_{\mathrm{AIU}} = 0$ before 0.1k. In Ours, AdamW-GS uses the same configuration but different StSS settings. Sparse denotes using Sparse Adam or Adam, and Half means using Adam in densification but Sparse Adam in P-Op. $^{\mathrm{RSR}}$ here means only adding RSR to the pipeline.

| cite | Sparse | AIU | Pro$_{\mathrm{AIU}}$ | $\eta_{\mathrm{AIU}}$ | noise | Ours | PSNR | SSIM | LPIPS | $N_a$/m | $N_d$/m | $\Delta N_a$ |
|------|--------|-----|------|------|-------|------|------|------|-------|------|------|------|
| GS1 | x | - | - | - | - | - | 31.505 | 0.928 | 0.127 | 1.699 | 0.116 | - |
| GS2 | ✓ | - | - | - | - | - | 31.053 | 0.926 | 0.131 | 1.621 | 0.036 | -4.59% |
| GS3 | Half | - | - | - | - | - | 31.564 | 0.929 | 0.127 | 1.774 | 0.040 | 4.41% |
| GS0 | ✓ | - | - | - | - | StSSGS1$^{\mathrm{RSR}}$ | 31.072 | 0.922 | 0.133 | 0.930 | 0.013 | -45.2% |
| GS4 | Half | [15,30] | 0.5 | [0.1][15] | - | - | 31.531 | 0.929 | 0.127 | 1.721 | 0.089 | 1.29% |
| GS15 | Half | [15,30] | 1.0 | [0.1][15] | - | - | 31.509 | 0.929 | 0.127 | 1.602 | 0.237 | -5.70% |
| GS16 | ✓ | [0.1,15] | 0.5 | [0.1][0.1] | - | - | 31.110 | 0.926 | 0.130 | 1.766 | 0.035 | 3.94% |
| GS17 | ✓ | [0.1,7] | 0.2 | [0.5][0.1] | - | - | 31.389 | 0.927 | 0.129 | 1.740 | 0.037 | 2.41% |
| GS18 | ✓ | [0.1,15] | 0.2 | [0.2,0.5,0.1] [0.1,1,7] | - | - | 30.997 | 0.926 | 0.129 | 1.769 | 0.035 | 4.12% |
| GS19 | ✓ | [0.1,15] | 0.2 | [0.5,0.1] [0.1,3] | - | - | 31.330 | 0.927 | 0.129 | 1.724 | 0.035 | 1.47% |
| GS5 | ✓ | [0.1,15] | 0.2 | [0.8,0.5,0.1] [0.1,3,7] | - | - | 31.355 | 0.927 | 0.129 | 1.772 | 0.036 | 4.29% |
| GS6 | ✓ | [0.1,30] | 1.0 | [0.1][0.1] | - | - | 31.404 | 0.928 | 0.128 | 1.782 | 0.069 | 4.88% |
| GS13 | ✓ | - | - | - | - | StSSGS3 | 31.770 | 0.927 | 0.131 | 0.692 | 0.005 | -59.3% |
| GS7/GS8 | ✓ | - | - | - | - | StSSGS1 | 31.728 | 0.927 | 0.132 | 0.667 | 0.004 | -60.7% |
| GS14 | ✓ | - | - | - | - | StSSGS4 | 31.741 | 0.926 | 0.133 | 0.667 | 0.004 | -60.7% |

Table 28: Quantitative results on the Kitchen from Mip-NeRF 360 with different components in the 3DGS-MCMC pipeline. AIU definitions follow the table description above (Table 27). For RSR, the sampler defaults to uniform (interval = 100) except StSS. $\mathcal{R}$ denotes the form of regularization, $\lambda$ its hyperparameter, and max corresponds to $\mathcal{C}_t$ in the paper; other parameters remain default. $^0$ in StSSMC1 denotes using $\alpha_1 = \alpha_2 = 0$ here.

| cite | Sparse | AIU | Pro$_{\mathrm{AIU}}$/$\eta_{\mathrm{AIU}}$ | RSR | $\mathcal{R}_o$ | $\lambda_o$/max$_o$ | $\mathcal{R}_s$ | $\lambda_s$/max$_s$ | PSNR | SSIM | LPIPS | $N_a$/m | $N_d$/m | $\Delta N_a$ |
|------|--------|-----|------|-----|------|------|------|------|------|------|-------|------|------|------|
| MC1 | x | - | - | - | L1 | 0.01/- | L1 | 0.01/- | 32.173 | 0.933 | 0.122 | 1.661 | 0.149 | -2.23% |
| MC2 | ✓ | - | - | - | L1 | 0.01/- | L1 | 0.01/- | 32.079 | 0.934 | 0.122 | 1.753 | 0.057 | 3.17% |
| MC3 | ✓ | [0.1,30] | 0.1/0.1 | - | L1 | 0.01/- | L1 | 0.01/- | 32.286 | 0.934 | 0.122 | 1.743 | 0.067 | 2.58% |
| MC4 | ✓ | [3,30] | 0.1/0.1 | StSSMC1 | L1 | 0.01/- | L1 | 0.01/- | 31.765 | 0.909 | 0.137 | 1.681 | 0.129 | -1.05% |
| MC6 | ✓ | [26,30] | 0.01/1 | StSSMC1 | DAR | 0.001/10 | DAR | $10^{-5}$/10 | 21.787 | 0.820 | 0.234 | 1.729 | 0.081 | 1.76% |
| MC18 | ✓ | - | - | StSSMC1$^0$ | L1 | 0.01/- | L1 | 0.01/- | 32.363 | 0.346 | 0.120 | 1.72 | 0.09 | 1.23% |
| MC19 | ✓ | - | - | StSSMC1 | L1 | 0.01/- | L1 | 0.01/- | 32.289 | 0.934 | 0.119 | 1.714 | 0.096 | 0.88% |
| MC20 | ✓ | - | - | StSSMC2 | L1 | 0.01/- | L1 | 0.01/- | 31.924 | 0.931 | 0.124 | 1.495 | 0.315 | -12.0% |
| MC9 | ✓ | - | - | StSSMC1 | DAR | 0.001/- | L1 | 0.01/- | 32.342 | 0.934 | 0.121 | 1.703 | 0.107 | 2.35% |
| MC10 | ✓ | - | - | StSSMC1 | DAR | 0.001/10 | L1 | 0.01/- | 32.570 | 0.937 | 0.117 | 1.765 | 0.045 | 3.88% |
| MC11 | ✓ | - | - | StSSMC2 | DAR | 0.001/10 | L1 | 0.01/- | 32.397 | 0.935 | 0.118 | 1.763 | 0.047 | 3.76% |
| MC12 | ✓ | - | - | StSSMC1 | DAR | 0.001/5 | L1 | 0.01/- | 32.443 | 0.936 | 0.118 | 1.767 | 0.043 | 4.00% |
| MC13 | ✓ | - | - | StSSMC1 | DAR | 0.001/1 | L1 | 0.01/- | 32.458 | 0.935 | 0.118 | 1.786 | 0.024 | 5.12% |
| MC14 | ✓ | - | - | StSSMC1 | DAR | 0.001/10 | DAR | 0.001/10 | 32.466 | 0.936 | 0.117 | 1.766 | 0.044 | 3.94% |
| MC15 | ✓ | - | - | StSSMC1 | DAR | 0.001/10 | DAR | 0.001/1 | 32.173 | 0.936 | 0.118 | 1.766 | 0.044 | 3.94% |
| MC16 | ✓ | - | - | StSSMC1 | DAR | 0.001/10 | DAR | $10^{-5}$/1 | 32.521 | 0.936 | 0.117 | 1.756 | 1.756 | 3.35% |
| MC8/MC7 | ✓ | - | - | StSSMC1 | DAR | 0.001/10 | DAR | $10^{-5}$/10 | 32.546 | 0.937 | 0.117 | 1.755 | 1.755 | 3.29% |
| MC21 | ✓ | - | - | StSSMC2 | DAR | 0.001/10 | DAR | $10^{-5}$/10 | 32.298 | 0.936 | 0.117 | 1.753 | 0.057 | 3.17% |

### K.9 BONSAI FROM MIPNERF-360

Table 29: Quantitative results on the Bonsai from Mip-NeRF 360 with different components in the vanilla 3DGS pipeline. For AIU, the brackets denote its active iteration range (e.g., [15,30] means 15k–30k iterations). ProAIU is the sampling probability of primitives, and $\eta_{\text{AIU}}$ constrains the extra update step applied to sampled primitives. $[\cdot][\cdot]$ denote the scale value and the iteration at which it takes effect. For example, [0.5,0.1][0.1,3] means $\eta_{\text{AIU}} = 0.5$ after 0.1k iterations, $\eta_{\text{AIU}} = 0.1$ after 3k, and $\eta_{\text{AIU}} = 0$ before 0.1k. In Ours, AdamW-GS uses the same configuration but different StSS settings. Sparse denotes using Sparse Adam or Adam, and Half means using Adam in densification but Sparse Adam in P-Op. $^{\text{RSR}}$ here means only adding RSR to the pipeline.

| cite | Sparse | AIU | Pro$_{\text{AIU}}$ | $\eta_{\text{AIU}}$ | noise | Ours | PSNR | SSIM | LPIPS | $N_a$/m | $N_d$/m | $\Delta N_a$ |
|---|---|---|---|---|---|---|---|---|---|---|---|---|
| GS1 | x | - | - | - | - | - | 32.269 | 0.943 | 0.207 | 1.159 | 0.115 | - |
| GS2 | ✓ | - | - | - | - | - | 31.610 | 0.931 | 0.216 | 1.230 | 0.016 | 6.12% |
| GS3 | Half | - | - | - | - | - | 32.325 | 0.943 | 0.206 | 1.256 | 0.017 | 8.36% |
| GS0 | ✓ | - | - | - | - | StSSGS1$^{\text{RSR}}$ | 32.284 | 0.942 | 0.205 | 1.003 | 0.011 | -13.4% |
| GS4 | Half | [15,30] | 0.5 | [0.1][15] | - | - | 32.382 | 0.943 | 0.206 | 1.230 | 0.037 | 6.12% |
| GS15 | Half | [15,30] | 1.0 | [0.1][15] | - | - | 32.406 | 0.943 | 0.206 | 1.208 | 0.072 | 4.22% |
| GS16 | ✓ | [0.1,15] | 0.5 | [0.1][0.1] | - | - | 32.243 | 0.942 | 0.209 | 1.360 | 0.016 | 17.3% |
| GS17 | ✓ | [0.1,7] | 0.2 | [0.5][0.1] | - | - | 32.106 | 0.942 | 0.208 | 1.353 | 0.017 | 16.7% |
| GS18 | ✓ | [0.1,15] | 0.2 | [0.2,0.5,0.1] [0.1,1,7] | - | - | 32.293 | 0.943 | 0.208 | 1.370 | 0.015 | 18.2% |
| GS19 | ✓ | [0.1,15] | 0.2 | [0.5,0.1] [0.1,3] | - | - | 32.260 | 0.941 | 0.209 | 1.283 | 0.015 | 10.6% |
| GS5 | ✓ | [0.1,15] | 0.2 | [0.8,0.5,0.1] [0.1,3,7] | - | - | 32.260 | 0.943 | 0.208 | 1.378 | 0.016 | 18.8% |
| GS6 | ✓ | [0.1,30] | 1.0 | [0.1][0.1] | - | - | 32.251 | 0.942 | 0.208 | 1.428 | 0.034 | 23.2% |
| GS13 | ✓ | - | - | - | - | StSSGS3 | 32.409 | 0.942 | 0.204 | 0.807 | 0.004 | -30.3% |
| GS7/GS8 | ✓ | - | - | - | - | StSSGS1 | 32.251 | 0.942 | 0.205 | 0.747 | 0.004 | -35.5% |
| GS14 | ✓ | - | - | - | - | StSSGS4 | 32.279 | 0.941 | 0.206 | 0.735 | 0.003 | -36.5% |

Table 30: Quantitative results on the Bonsai from Mip-NeRF 360 with different components in the 3DGS-MCMC pipeline. AIU definitions follow the table description above (Table 29). For RSR, the sampler defaults to uniform (interval = 100) except StSS. $\mathcal{R}$ denotes the form of regularization, $\lambda$ its hyperparameter, and max corresponds to $\mathcal{C}_t$ in the paper; other parameters remain default. $^0$ in StSSMC1 denotes using $\alpha_1 = \alpha_2 = 0$ here.

| cite | Sparse | AIU | Pro$_{\text{AIU}}$/$\eta_{\text{AIU}}$ | RSR | $\mathcal{R}_o$ | $\lambda_o$/max$_o$ | $\mathcal{R}_s$ | $\lambda_s$/max$_s$ | PSNR | SSIM | LPIPS | $N_a$/m | $N_d$/m | $\Delta N_a$ |
|---|---|---|---|---|---|---|---|---|---|---|---|---|---|---|
| MC1 | x | - | - | - | L1 | 0.01/- | L1 | 0.01/- | 32.572 | 0.946 | 0.200 | 1.101 | 0.169 | -5.00% |
| MC2 | ✓ | - | - | - | L1 | 0.01/- | L1 | 0.01/- | 32.508 | 0.945 | 0.203 | 1.200 | 0.070 | 3.53% |
| MC3 | ✓ | [0.1,30] | 0.1/0.1 | - | L1 | 0.01/- | L1 | 0.01/- | 32.578 | 0.946 | 0.202 | 1.191 | 0.079 | 2.76% |
| MC4 | ✓ | [3,30] | 0.1/0.1 | StSSMC1 | L1 | 0.01/- | L1 | 0.01/- | 32.742 | 0.947 | 0.198 | 1.130 | 0.140 | -2.50% |
| MC6 | ✓ | [26,30] | 0.01/1 | StSSMC1 | DAR | 0.001/10 | DAR | $10^{-5}$/10 | 29.919 | 0.922 | 0.213 | 1.204 | 0.066 | 3.88% |
| MC18 | ✓ | - | - | StSSMC1$^0$ | L1 | 0.01/- | L1 | 0.01/- | 32.527 | 0.946 | 0.201 | 1.205 | 0.065 | 3.96% |
| MC19 | ✓ | - | - | StSSMC1 | L1 | 0.01/- | L1 | 0.01/- | 32.645 | 0.947 | 0.199 | 1.201 | 0.069 | 3.62% |
| MC20 | ✓ | - | - | StSSMC2 | L1 | 0.01/- | L1 | 0.01/- | 32.417 | 0.944 | 0.204 | 0.959 | 0.311 | -17.2% |
| MC9 | ✓ | - | - | StSSMC1 | DAR | 0.001/- | L1 | 0.01/- | 32.513 | 0.943 | 0.206 | 1.184 | 0.086 | 2.15% |
| MC10 | ✓ | - | - | StSSMC1 | DAR | 0.001/10 | L1 | 0.01/- | 33.006 | 0.950 | 0.191 | 1.221 | 0.049 | 5.34% |
| MC11 | ✓ | - | - | StSSMC2 | DAR | 0.001/10 | L1 | 0.01/- | 32.990 | 0.950 | 0.189 | 1.234 | 0.036 | 6.47% |
| MC12 | ✓ | - | - | StSSMC1 | DAR | 0.001/5 | L1 | 0.01/- | 32.990 | 0.950 | 0.189 | 1.237 | 0.033 | 6.72% |
| MC13 | ✓ | - | - | StSSMC1 | DAR | 0.001/1 | L1 | 0.01/- | 32.921 | 0.950 | 0.190 | 1.252 | 0.018 | 8.02% |
| MC14 | ✓ | - | - | StSSMC1 | DAR | 0.001/10 | DAR | 0.001/10 | 32.916 | 0.950 | 0.189 | 1.233 | 0.037 | 6.38% |
| MC15 | ✓ | - | - | StSSMC1 | DAR | 0.001/10 | DAR | 0.001/1 | 32.906 | 0.950 | 0.189 | 1.232 | 0.038 | 6.29% |
| MC16 | ✓ | - | - | StSSMC1 | DAR | 0.001/10 | DAR | $10^{-5}$/1 | 32.990 | 0.950 | 0.190 | 1.220 | 0.050 | 5.26% |
| MC8/MC7 | ✓ | - | - | StSSMC1 | DAR | 0.001/10 | DAR | $10^{-5}$/10 | 33.022 | 0.950 | 0.190 | 1.220 | 0.050 | 5.26% |
| MC21 | ✓ | - | - | StSSMC2 | DAR | 0.001/10 | DAR | $10^{-5}$/10 | 33.007 | 0.950 | 0.190 | 1.219 | 0.051 | 5.17% |

