# OpenReview forum: "A Step to Decouple Optimization in 3DGS"
_ICLR.cc/2026/Conference — ICLR 2026 Poster_

### Official Review · Reviewer_odb6 · 2025-10-27

**Soundness:** 3
**Presentation:** 3
**Contribution:** 3
**Rating:** 6
**Confidence:** 2

**Summary:**

The authors analyzed Adam optimizer for the problem of 3DGS optimization, provided several observations and modifications on the optimizer to be more effective, summarized as AdamW-GS. Specifically, they made the observation that the previously proposed sparse Adam is effective at eliminating dead gaussians, but also lead to performance degradation. To resolve this, they have proposed steps such as Re-State Regularization (RSR) and Decoupled Attribute Regularization (DAR). RSR aims at better activating regularization by rescaling the moment, by introducing two additional scaler. DAR aims to decouple the main objective function (photometric loss) and regularization (on opacity and scale), such that they do not influence each other and cause over/under regularization. Experiments demonstrate that AdamW-GS consistently improves reconstruction/NVS results compared to 3DGS/3DGS-MCMC.

**Strengths:**

The authors tackle an important problem in determining the appropriate optimization function on the problem of 3DGS reconstruction. While 3DGS has been influential, the optimization behavior can easily get stuck either due to overfitting/floaters or underfitting (blobs).

The proposed AdamW-GS shows consistent improvements compared to 3DGS and 3DGS-MCMC, with faster training process and fewer (dead) gaussian primitives.

I am not an expert in optimization, but the experiments, observations, and modifications seem reasonable and generalizable enough to me.

**Weaknesses:**

The modification in AdamW-GS introduces additional hyperparameters, with no obvious intuition in how to adjust them. The RSR parameters, for example, are obtained through grid search. While observations show that the regularization term is sometimes over/under effective, it is unclear to me how to determine over/under fitting based on a single run and tune hyperparameters accordingly.

I find the table listing of MC1, MC2, etc. to be confusing and difficult to cross reference and understand what the authors are doing.

A few papers on automated efficient 3DGS may be worth citing:

PUP-3DGS (Hanson et al., CVPR25) - determines dead gaussians/gaussians with low contribution to loss
LP-3DGS (Zhang et al. NIPS24) - automatically reduces gaussian counts during training, similar to AdamW-GS
3DGS-LM (Höllein et al. ICCV25) - Optimization with Levenberg-Marquardt instead of Adam

**Questions:**

See Weakness - how would authors propose for tuning the additional hyperparameters in practice?

Since Taming 3DGS is extensively referenced, has it been properly compared to this work?

---

> ### Author Response · Authors · 2025-11-21
>
> Thanks for your valuable comments. Below we provide our responses, and we will incorporate the relevant parts into the manuscript. **The revised version has been updated; we mark the newly added information with blue text.**
>
> **For weakness 1: (The corresponding analyses in this part have been added in our paper.)**
>
> We organize our discussion of parameter selection, justification, and robustness into 4 parts: (1) the RSR rescaling parameters $\alpha_1$ and $\alpha_2$; (2) the RSR StSS schedule; (3) the clipping and regularization hyperparameters in DAR; and (4) the applicability of the overall method.
> 1. **Basis for the RSR rescaling parameters $\alpha_1$ and $\alpha_2$, some of which we have mentioned in paper**:
> - When $\alpha_1$ and $\alpha_2$ are chosen to be sufficiently small while satisfying $\alpha_2 = \alpha_1^2$, this setting is safe: it is easy to verify, since the states of the generative primitives in 3DGS or the reallocated primitives in 3DGS-MCMC can be viewed as being naturally constrained by an RSR with $\alpha_1 = \alpha_2 = 0$.
> - When we need to control the magnitude of the update step $\frac{\hat{m}}{\sqrt{\hat{v}}+\epsilon}$, it is necessary to enforce $\alpha_2 = \alpha_1^2$.
> - Our RSR is motivated by the observation that implicit update of Adam naturally rescales the state; since each primitive typically appears only in a subset of views, this rescaling factor tends to be small. We also visualize this discrepancy in our paper.
> - We report experiments with $\alpha_1 = \alpha_2 = 0$ and $\alpha_1 = 0.2, \alpha_2 = 0.04$. These settings are described in the Hyperparameter Selection section, and all experiments cited as MC18/19 in the final All Results section show them. The results indicate that the performance gap between the two configurations is small, with $\alpha_1 = 0.2, \alpha_2 = 0.04$ yielding slightly better results.
> - A very small $\alpha_2$ is consistent with the design of DAR, as we rely on such a small $\alpha_2$ to rescale the second moment and thereby activate the regularization. As discussed in the penultimate paragraph of Sec. 4.3, the designs of RSR and DAR are intrinsically coupled and mutually influence each other.
> - All our experiments use the setting $\alpha_1 = 0.2, \alpha_2 = 0.04$, and all newly added experiments also adopt this configuration. The results show that this choice is robust.
> - We use grid search, but our choice is grounded in analytical considerations and empirical evidence.
> - In summary, the choice of $\alpha_1$ and $\alpha_2$ is grounded in both empirical evidence and practical considerations: when $\alpha_1$ and $\alpha_2$ are chosen to be small and satisfy $\alpha_2 = \alpha_1^2$, this setting is safe, whereas parameter choices that violate this relation are inherently unreasonable. Based on extensive experiments, we recommend $\alpha_1 = 0.2, \alpha_2 = 0.04$.
> 2. **Suggestion for choosing the StSS schedule**
> - The differences between these schedules mainly arise from two factors: (1) **differences in the underlying pipelines**, which lead 3DGS and 3DGS-MCMC to adopt entirely different schedules; and (2) **differences across scenes**, where we use different schedules to obtain the best results on indoor or outdoor scenes.  Fundamentally, these discrepancies are induced by variations in the **absolute increment and its proportion of primitive generation or reallocation across pipelines and scenes**. When these quantities are high, many primitives already remain in a zero state, in which case we prefer to use a smaller StSS rate. Consulting **Figure 3** (reallocation primitive number change for 3DGSMCMC) and **Figures 2** (primitives number change for 3DGS) **in our original paper provides further insight into our choice of the StSS schedule**.
> - The schedule differences caused by intrinsic pipeline differences are **unavoidable**. Nevertheless, for each pipeline we design a **conservative schedule**, denoted StSSGS1 and StSSMC1, under which both 3DGS and 3DGS-MCMC **achieve better performance than their original counterparts**. For outdoor scenes, which typically exhibit higher primitive generation or reallocation increment, we find that **a more aggressive schedule yields better results**. All experiments are reported in the last section (All Results) of the original paper, where we annotate the corresponding "Ours" or "RSR" entries with the respective StSS code.
> - Regarding the sensitivity to the schedule, this has already been discussed in the paragraph (Side Effect in Coupling) of Section 5 (Result and Analysis). Thanks to our decoupling of the regularization, using a stronger schedule **does not lead to a pronounced degradation in reconstruction quality**, and the performance remains superior to the original method.

---

> ### Author Response · Authors · 2025-11-21
>
> 3. **Basis for the clipping in DAR and its regularization hyperparameters**
> - Selecting a better regularization hyperparameters is _a common issue in the design of regularization_; similar problem/discussions can be found in issue \#26 of the 3DGSMCMC GitHub repository. In the All Results section, we present experiments (MC14/15) showing that scaling regularization in the DAR formulation exhibits a certain degree of robustness for relatively large hyperparameters, without causing a noticeable degradation.
> - The clipping here is introduced to align the magnitude of the decoupled regularization with Adam and the empirical magnitudes observed in practice. So, **the number choice is reasonable**.
> 4. **the applicability of the overall method**
> - Except for the StSS schedule, all other hyperparameters are **kept identical across all baselines and datasets**. To further demonstrate the generalization ability of our method, we additionally report results on **3 new baselines** (MaskGaussian, Taming-3DGS, and DBS) and **new dataset** (OMMO). **The results and analysis for Taming-3DGS are reported in our respond to Question below**, while the analyses for MaskGaussian, DBS and OMMO can be found in our response to reviewer A2qo, weakness 1 and 3.
> - The **findings obtained from our exploration of the optimization also exhibit a generalizable** nature, and the experimental behavior of MaskGaussian further corroborates the observations presented in our paper. ( see our responds to reviewer A2qo, weakness 1 )
>
> ---
>
> ---
>
> **For weakness 2**:
>
> We thank the reviewer for the valuable comments. We **will add further clarifications** to the relevant tables and main text to **improve the readability** of the paper.
>
> **For weakness 3 about citing:**
>
> We have added the relevant citations. Related information can be found in the Related Work section of the main paper or in the appendix.
>
> ---
>
> **For Question about Taming-3DGS: (The corresponding experiments and analyses have been added in our paper).**
>
> We additionally provide experiments on **Taming-3DGS** (both the final count mode with a larger budget and the multiplier mode with a smaller budget as specified in the official script), each trained without or with our proposed AdamW-GS.
> - For Taming-3DGS (both the final count  mode and the multiplier mode, see **Table 1 below**), our method consistently **reconstructs higher-quality scenes more _rapidly_ while using fewer active primitives**. In the revised paper (will be updated later), Figure 6 compares the renderings of the original Taming-3DGS and Taming-3DGS with AdamW-GS under both modes, showing that our approach better **preserves fine details**. Leveraging the properties of our method, we **further propose an modified variant of Taming-3DGS**: since **our method can rapidly penalize redundant primitives**, we replace the conservative pruning operation in Taming-3DGS with the pruning operation of vanilla 3DGS, yielding **Taming-3DGS-p**. Comparing the original Taming-3DGS and Taming-3DGS-p with AdamW-GS in the final count mode, our method not only **improves reconstruction quality** but also **reduces training time to 40%** and **further lowers memory usage**, with a final $N_P$ of only 2.16 million versus 3.20 million for the original Taming-3DGS.
>
> ---
> ---
>
>
> **Table 1**: Taming 3DGS w/ or w/o AdamW-GS (numbers are reported in million, time cost is reported in mins.)
>
> | Pipeline                    | AdamW-GS | All :   |    |    |      |       | All       | Indoor: |  |  | | Indoor | Outdoor: |  |  |  | Outdoor |
> | ----- | ---- | --- | ----- | ---- | ---- | ------ | --- | ---- | ---- | ---- | ---- | ---- | --- | ---- | ---- | ---- | ----- |
> |                             |           | PSNR$\uparrow$  | SSIM$\uparrow$ | LPIPS$\downarrow$ | $N_p$        | $N_a$        | Time      | PSNR  | SSIM  | LPIPS | $N_p$     | $N_a$     | PSNR   | SSIM   | LPIPS  | $N_p$      | $N_a$     |
> | Taming-3DGS (multiplier)    | x         | 27.386 | 0.796 | 0.258  | 0.668     | 0.620     | 7.44      | 31.025 | 0.918  | 0.205  | 0.357  | 0.329  | 24.479  | 0.698   | 0.299   | 0.916   | 0.853   |
> |                             | $\checkmark$         | 27.537 | 0.799 | 0.254  | 0.646     | 0.575     | 5.27      | 31.268 | 0.920  | 0.200  | 0.357  | 0.313  | 24.552  | 0.703   | 0.297   | 0.877   | 0.785   |
> | Taming-3DGS (final count)   | x         | 27.912 | 0.822 | 0.207  | 3.205     | 2.609     | 20.30     | 31.603 | 0.928  | 0.181  | 1.377  | 1.144  | 25.949  | 0.736   | 0.228   | 4.667   | 3.776   |
> |                             | $\checkmark$         | 28.034 | 0.826 | 0.207  | 3.109     | **1.847** | **10.46** | 31.720 | 0.928  | 0.180  | 1.408  | 0.759  | 25.805  | 0.744   | 0.229   | 4.469   | 2.717   |
> | Taming-3DGS-p (final count) | $\checkmark$         | 28.038 | 0.828 | 0.205  | **2.160** | **2.160** | **8.44**  | 31.724 | 0.930  | 0.177  | 0.804  | 0.804  | 25.089  | 0.745   | 0.227   | 3.346   | 3.246   |

---

> ### Author Response · Authors · 2025-11-27
>
> Dear Reviewer,
>
> I hope this message finds you well. As the discussion period is drawing to a close, I wanted to ensure we have addressed all your concerns satisfactorily. If there are any remaining points you would like us to consider, please let us know at your convenience. Your feedback has been extremely valuable to us, and we would be very happy to provide additional clarification or make further revisions that could help strengthen the manuscript.
>
> Thank you again for your time and effort in reviewing our work.

---

### Official Review · Reviewer_AiQe · 2025-11-01

**Soundness:** 3
**Presentation:** 3
**Contribution:** 3
**Rating:** 6
**Confidence:** 4

**Summary:**

The paper re-examines optimization in 3D Gaussian Splatting (3DGS) and argues that two kinds of coupling: (i) update-step coupling from synchronous Adam that induces
"implicit updates" on invisible primitives, and (ii) gradient coupling where regularization gradients are mixed into Adam's moments. It proposes a three-part redesign: Sparse Adam for view-conditioned updates, Re-State Regularization (RSR) to periodically rescale moments, and Decoupled Attribute Regularization (DAR) that removes $\nabla R$ from moment updates and applies a preconditioned, clipped regularization step. The recoupled system ("AdamW-GS") is shown to improve PSNR while reducing total runtime.

**Strengths:**

+ The motivation is well-explained, and provides carefully analysis (with figures and measurements) to show the synchronous Adam rescales moments even with zero gradients, producing implicit updates.
+ Sparse Adam + RSR + DAR is easy to implement, integrates with both vanilla 3DGS and 3DGS-MCMC
+ Experimental results show AdamW-GS improves quality and cuts runtime

**Weaknesses:**

+ The proposed AdamW-GS optimizer is far more complex to tune than the standard Adam. It introduces numerous new hyperparameters, such as moment scaling factors ($\alpha_1$, $\alpha_2$) for RSR 15, and regularization scales ($\lambda$) and clipping values ($\mathcal{C}_t$) for DAR 16. This may pose challenges to directly applied hyper-parameter settings on MipNerf360 to other scenes.

+ Related to first point, the crucial RSR component relies on a "hand-crafted" milestone-based schedule (StSS) to function. This schedule is manually designed for the test datasets (see Fig. 6) and is unlikely to generalize to new scenes or datasets without being re-tuned

+ The claim of removing primitives "without introducing additional pruning components" is nuanced. The method works by aggressively pushing opacity below the existing 1/255 threshold, at which point the original 3DGS rendering pipeline already excludes them.

+ Some important ablations are missing. For example, the contribution of DAR is not isolated in the ablation studies. To prove DAR’s advantage isn’t just a scale/clip effect, the paper should include step-size-matched baselines (e.g., L1 with λ tuned to match DAR’s average regularization step; AdamW-style constant penalty with matched effective step). Current comparisons don’t fully disentangle mechanism vs magnitude.

**Questions:**

+ The "hand-crafted" StSS schedule  is the method's most significant liability. How sensitive is the performance to this schedule? Could authors provide additional experiments on different scenes with the same schedule?

+ The paper notes that Sparse Adam alone degrades performance, and the RSR component is required to "fix" it. This suggests a tight, possibly brittle, inter-dependency. How robust is the method to the RSR hyperparameters? If the $\alpha$ values are slightly misconfigured, does the optimization fail entirely?

+ Regarding the ablation studies for DAR, it remains unclear whether the observed benefits stem from DAR's adaptive mechanism (i.e., scaling by $\sqrt{\hat{v}}$) or simply from its resultant effective magnitude. Please include baselines where (a) L1 and (b) constant-penalty (AdamW-style) are tuned to match DAR's average/max regularization step (with and without $C_t$), to rule out scale effects.

+ Given that the noise-based exploration mechanism is scene-dependent and can be harmful, what alternative exploration strategies could be integrated? Could the RSR's moment-rescaling be adapted to also induce exploration (e.g., by adding noise during rescaling) rather than just activating regularization?

---

> ### Author Response · Authors · 2025-11-21
>
> Thanks for your valuable comments. Below we provide our responses, and we will incorporate the relevant parts into the manuscript. **The revised version has been updated; we mark the newly added information with blue text.**
>
> **For Question 1: (The corresponding analyses in this part have been added in the our paper.)**
>
> The differences between these schedules mainly arise from two factors: (1) **differences in the underlying pipelines**, which lead 3DGS and 3DGS-MCMC to adopt entirely different schedules; and (2) **differences across scenes**, where we use different schedules to obtain the best results on indoor or outdoor scenes. Fundamentally, these discrepancies are induced by variations in the **absolute increment and its proportion of primitive generation or reallocation across pipelines and scenes**. When these quantities are high, many primitives already remain in a zero state, in which case we prefer to use a smaller StSS rate. Consulting **Figure 3** (reallocation primitive number change for 3DGSMCMC) and **Figures 2** (primitives number change for 3DGS) in our original paper provides further insight into our choice of the StSS schedule.
>
> The schedule differences caused by intrinsic pipeline differences are unavoidable. Nevertheless, for each pipeline we **design a conservative schedule**, denoted StSSGS1 and StSSMC1, under which both 3DGS and 3DGS-MCMC achieve better performance than their original counterparts. For outdoor scenes, which typically exhibit higher primitive generation or reallocation increment, we find that _a more aggressive schedule yields better results_. All experiments are reported in the last section (All Results) of the original paper, where we annotate the corresponding "Ours" or "RSR" entries with the respective StSS code.
>
> Regarding the sensitivity to the schedule, this has already been discussed  in the paragraph (Side Effect in Coupling) of Section 5 (Result and Analysis). Thanks to our decoupling of the regularization, **using a stronger schedule does not lead to a pronounced degradation in reconstruction quality**, and the performance remains superior to the original method.
>
> To further demonstrate the generalization ability of our method, we **additionally report results on new baselines** (MaskGaussian, Taming-3DGS, DBS) and **new dataset** (OMMO). A detailed analysis is provided in our response to reviewer A2qo weakness 1. Here, we present the experimental results for MaskGaussian (see our responds to weakness 3 below).
>
> ---
>
> **For Question 2: (The corresponding analyses  in this part have been added in our paper)**
>
> For the RSR-related parameters, the main hyperparameters are those of StSS and the coefficients $\alpha_1$ and $\alpha_2$ in Eq. 5 of the paper. We have already answer the sensitivity of the StSS parameters in our response to **Question 1**, so here we focus on $\alpha_1$ and $\alpha_2$. (Some points have been in the paper.)
> 1. When $\alpha_1$ and $\alpha_2$ are chosen to be sufficiently small while satisfying $\alpha_2 = \alpha_1^2$, this setting is safe: it is easy to verify, since the states of the new generated primitives in 3DGS or the reallocated primitives in 3DGS-MCMC can be viewed as being naturally constrained by an RSR with $\alpha_1 = \alpha_2 = 0$.
> 2. When we need to control the magnitude of the update step $\frac{\hat{m}}{\sqrt{\hat{v}}+\epsilon}$, it is necessary to enforce $\alpha_2 = \alpha_1^2$.
> 3. Our RSR is motivated by the observation that implicit update of Adam naturally rescales the state; since each primitive typically appears only in a subset of views, this rescaling factor tends to be small. We also visualize this discrepancy in our paper.
> 4. We report experiments with $\alpha_1 = \alpha_2 = 0$ and $\alpha_1 = 0.2, \alpha_2 = 0.04$. These settings are described in the Hyperparameter Selection section, and all experiments cited as MC18/19 in the final All Results section show them. The results indicate that the performance gap between the two configurations is small, with $\alpha_1 = 0.2, \alpha_2 = 0.04$ yielding slightly better results.
> 5. A very small $\alpha_2$ is consistent with the design of DAR, as we rely on such a small $\alpha_2$ to rescale the second moment and thereby activate the regularization. As discussed in the penultimate paragraph of Sec. 4.3, the designs of RSR and DAR are intrinsically coupled and mutually influence each other.
> 6. All our experiments use the setting $\alpha_1 = 0.2, \alpha_2 = 0.04$, and all newly added experiments also adopt this configuration. The results show that this choice is robust.
> 7. In summary, the choice of $\alpha_1$ and $\alpha_2$ is grounded in both empirical evidence and practical considerations: when $\alpha_1$ and $\alpha_2$ are chosen to be small and satisfy $\alpha_2 = \alpha_1^2$, this setting is safe, whereas parameter choices that violate this relation are inherently unreasonable. Based on extensive experiments, we recommend $\alpha_1 = 0.2, \alpha_2 = 0.04$.

---

> > ### Author Response · Authors · 2025-11-27
> >
> > Dear Reviewer,
> >
> > I hope this message finds you well. As the discussion period is drawing to a close, I wanted to ensure we have addressed all your concerns satisfactorily. If there are any remaining points you would like us to consider, please let us know at your convenience. Your feedback has been extremely valuable to us, and we would be very happy to provide additional clarification or make further revisions that could help strengthen the manuscript.
> >
> > Thank you again for your time and effort in reviewing our work.

---

> ### Author Response · Authors · 2025-11-21
>
> **For Question 3: (The corresponding experiments and  analyses  in this part have been added in the our paper)**
> 1. First, the step magnitude in L1 is **controlled by Adam**[1] . Direct modification of the gradients is beyond the scope of this work. Owing to the chain rule, we cannot simply choose a constant to directly clip the gradients, and there is currently no theory that provides a principled way to construct such a constraint function.
> 2. We additionally apply a clipping function to the AdamW-style decoupling consistent with our method, **using the same clipping range**. As shown in Table 1, introducing this clipping constraint does not change the results.
>
> **We analyze the current limitations of these two approaches in our paper** (see Sec.4.3 for a detailed discussion):
>
> 3. for L1: under coupling, the mismatch between $\nabla \mathcal{R}$ and $\nabla \ell$ distorts the intended amplification of the regularization. We empirically demonstrate this instability in Sec. 5.1.
> 4. for AdamW-style decoupling: its main issue is that it is independent of the primitive state.
> 5. By contrast, our design is inherently conditioned on the primitive state, and the clipping operation is introduced to align the magnitude of the decoupled regularization with Adam and the empirical magnitudes observed in practice. We give four detailed benifits analysis of DAR in the paper Sec.4.3.
>
> [1] A method for stochastic optimization
>
> ---
>
> **Table 1**: AdamW-style with clip ($\lambda_o<0.1$ for AdamW$_o$ + clip is ineffective as well.)
>
> | Methods          | $\lambda_o$ | All    |       |       | Indoor |       |       | Outdoor |       |       |
> | ---------------- | ----------- | ------ | ----- | ----- | ------ | ----- | ----- | ------- | ----- | ----- |
> |                  |             | PSNR   | LPIPS | SSIM  | PSNR   | SSIM  | LPIPS | PSNR    | SSIM  | LPIPS |
> | $\vert o \vert_1$        | 0.01        | 27.998 | 0.832 | 0.199 | 31.546 | 0.930 | 0.183 | 25.160  | 0.754 | 0.212 |
> | AdamW$_o$ + clip: | 0.1         | 27.240 | 0.215 | 0.808 | 30.992 | 0.928 | 0.182 | 24.239  | 0.713 | 0.242 |
> |                  | 1           | 27.256 | 0.231 | 0.806 | 30.850 | 0.926 | 0.194 | 24.381  | 0.713 | 0.260 |
> |                  | 10          | 20.488 | 0.505 | 0.568 | 21.901 | 0.727 | 0.436 | 19.358  | 0.440 | 0.560 |
>
> ---
>
> **Table 2**: more experiments related to our methods
>
> | Pipeline                         | All:   |       |        |          | Indoor: |       |        |          | Outdoor: |       |        |          |
> | -------------------------------- | ------ | ----- | ------ | -------- | ------- | ----- | ------ | -------- | -------- | ----- | ------ | -------- |
> |                                  | PSNR$\uparrow$  | SSIM$\uparrow$ | LPIPS$\downarrow$ | $\Delta N_a$      | PSNR$\uparrow$   | SSIM$\uparrow$ | LPIPS$\downarrow$ | $\Delta N_a$     | PSNR$\uparrow$    | SSIM$\uparrow$ | LPIPS$\downarrow$ | $\Delta N_a$      |
> | vanilla 3DGS                     | 27.506 | 0.815 | 0.216  | (3.098m) | 31.080  | 0.925 | 0.189  | (1.331m) | 24.648   | 0.728 | 0.239  | (4.512m) |
> | + Only RSR                       | 27.483 | 0.818 | 0.217  | -28.6%   | 30.981  | 0.924 | 0.190  | -29.5%   | 24.685   | 0.733 | 0.238  | -27.9%   |
> | + Adam W-GS (GS8)                | 27.678 | 0.822 | 0.220  | -49.3%   | 31.209  | 0.925 | 0.191  | -50.4%   | 24.854   | 0.740 | 0.243  | -48.4%   |
> | + Adam W-GS (GS8) + ABE          | 27.751 | 0.822 | 0.220  | -41.1%   | 31.304  | 0.925 | 0.191  | -44.3%   | 24.909   | 0.740 | 0.243  | -38.5%   |
> | + Adam W-GS (GS8) + Longer Densi | 27.715 | 0.824 | 0.218  | -48.4%   | 31.288  | 0.925 | 0.190  | -53.2%   | 24.857   | 0.744 | 0.240  | -44.6%   |

---

> ### Author Response · Authors · 2025-11-21
>
> **For Question 4: (The corresponding experiments and analyses in this part have been added in our paper.)**
>
> **Alternative exploration strategies**: Existing work offers limited discussion of exploration strategies for such 3DGS pipeline, and we believe this is a promising direction for further study.
>
> We propose **two new choices**:
> 1. we introduce Adaptive Bound-Expanding Split (**ABE-Split**)[2], which enhances exploration by **adding additional cross-region primitives**;
> 2. we leverage the densification mechanism itself by **extending the densification phase** from 15000 to 25000 iterations. Thanks to our method's ability to rapidly penalize redundant primitives, this **does not pose a risk of memory explosion**. The corresponding results are summarized in Table 2 above as "**AdamW-GS (GS8) + ABE**" and "**AdamW-GS (GS8) + Longer Densi.**" All outdoor experiments are conducted on top of the original noise regularization.
>
> **Experiments**:
> 1. **ABE-Split**: ABE-Split improves reconstruction quality on both indoor and outdoor scenes. Some cases show a clear gain; for example, on the _Room_ scene from MipNeRF-360, the PSNR/SSIM **improve from 31.500 dB / 0.920 for vanilla 3DGS to 32.121 dB / 0.923**.
> 2. **Extending the densification phase** also has a positive effect on indoor scenes and **further reduces the number of primitives**.
>
> **More discussion on RSR**: We conduct an **additional ablation study using only Sparse Adam and RSR**, with results reported in **Table 2 above** (**Only RSR**). The experiments show that RSR provides a certain degree of exploration, but this effect is limited. In Sec. 5.1 (Robust Hyperparameter Test and Autonomously Redundancy Removal), we state that **the overall method enhances exploration**, which stems from: (1) RSR; (2) the influence of DAR around saddle points; and (3) improved gradient flow within primitive groups once redundant primitives are reduced.
>
> Exploration strategies are not the primary focus of this work, and we leave DAR-style exploration strategies for future investigation.
>
> [2] Relaxing Accurate Initialization Constraint for  3D Gaussian Splatting
>
> ---
>
> **For weakness1:**
> We have discussed several RSR-related parameters in our responses to **Question 1 and Question 2**. Apart from StSS, which differs across pipelines, all other parameters are kept identical and unchanged. For StSS, we provide two conservative configurations for each pipeline, under which AdamW-GS consistently outperforms the original methods.
> As for selecting the better regularization hyperparameters, this is a common issue in the design of regularization; similar discussions of different hyperparameter settings can be found in issue \#26 of the 3DGSMCMC GitHub repository. The **three additional baselines** (which can be found in our respond to reviewer A2qo weakness 1) and **an extra dataset** (which can be found in our respond to reviewer A2qo weakness 3) we include **further demonstrate the robustness of our chosen parameters.**
>
> ---
>
> **For weakness3**:
> Our intention here is to emphasize that **our method is compatible with additional pruning components**. Accordingly, we include experiments on MaskGaussian with our proposed AdamW-GS. The results can be found in **Table 3 below**. We find that the behavior of MaskGaussian can be **explained by our observations**, and that our method **further improves its pruning effectiveness**.
> We attribute the phenomenon discussed in Sec.5.1 ("MaskGaussian suffers from potential reconstruction quality risk") to the characteristics of Adam analyzed in Observation 1 of Sec.4.1. **When MaskGaussian is trained with AdamW-GS, it can prune an additional 7% of primitives on indoor scenes without incurring this potential reconstruction-quality risk.**
>
> ---
>
> **Table 3**: vanilla 3DGS and MasskGaussion w/ or w/o  AdamW-GS
>
> | Pipeline     | AdamW-GS | All    | All   | All    | All      | Indoor | Indoor | Indoor | Indoor     | Outdoor | Outdoor | Outdoor | Outdoor  |
> | ----- | ----- | ------ | ----- | ------ | -------- | ------ | ------ | ------ | -------- | ------- | ------- | ------- | -------- |
> |              |           | PSNR$\uparrow$  | SSIM$\uparrow$ | LPIPS$\downarrow$ | $\Delta N_a$      | PSNR$\uparrow$  | SSIM$\uparrow$  | LPIPS$\downarrow$ | $\Delta N_a$  | PSNR$\uparrow$   | SSIM$\uparrow$   | LPIPS$\downarrow$  | $\Delta N_a$      |
> | vanilla 3DGS | x         | 27.506 | 0.815 | 0.216  | (3.098m) | 31.080 | 0.925  | 0.189  | (1.331m)   | 24.648  | 0.728   | 0.239   | (4.512m) |
> |              | $\checkmark$         | 27.678 | 0.822 | 0.220  | -49.3%   | 31.209 | 0.925  | 0.191  | -50.4%     | 24.854  | 0.740   | 0.243   | -48.4%   |
> | MaskGaussian | x         | 27.485 | 0.815 | 0.219  | -53.1%   | 30.988 | 0.924  | 0.192  | -61.5%     | 24.683  | 0.728   | 0.240   | -46.4%   |
> |              | $\checkmark$          | 27.721 | 0.821 | 0.221  | -57.2%   | 31.199 | 0.925  | 0.193  | **-68.6%** | 24.939  | 0.739   | 0.244   | -48.1%   |

---

### Official Review · Reviewer_A2qo · 2025-11-01

**Soundness:** 3
**Presentation:** 3
**Contribution:** 3
**Rating:** 6
**Confidence:** 2

**Summary:**

This paper revisits optimization in 3D Gaussian Splatting and argues that standard practice (Adam + synchronous updates) introduces two harmful couplings: (i) update-step coupling that rescales optimizer states and updates invisible primitives, and (ii) gradient/regularization coupling that makes effective weight decay depend on Adam’s second moment. It decomposes the training recipe into three parts: Sparse Adam, Re-State Regularization, and Decoupled Attribute Regularization, and then re-couples the beneficial pieces into AdamW-GS. Across vanilla 3DGS and 3DGS-MCMC, the paper reports consistent quality gains and meaningful wall-clock savings. Their method also reduces redundancy while improving rendering performance.

**Strengths:**

* This paper pinpoints that Adam synchronously updates even invisible primitives, rescaling their moments and changing attributes despite zero gradients (coined Implicit Update). Such a study helps the understanding of the 3DGS Optimization procedure.
* On outdoor scenes, the method reduces primitives by 48.4% while improving PSNR by +0.2 dB and SSIM by +0.01; on indoor scenes, it removes 50% of primitives while still gaining +0.1 dB PSNR, unlike MaskGaussian, which removes 61% but drops PSNR by 0.1 dB. The proposed approach strikes a better balance. Further, Table 4 (Deep Blending, Tanks & Temples) shows that AdamW-GS surpasses vanilla baselines.
* Time-cost breakdowns show large step-time and total runtime reductions compared with baselines.

**Weaknesses:**

* The comparison emphasizes MaskGaussian (and RePR), but omits other density-control splatting methods (Compact3DGS, Deformable beta splatting) cited in related work. Without matched-budget comparisons on the same scenes, it’s hard to evaluate the reported ~48–50% primitive reductions with small PSNR/SSIM gains compared with various baselines.
* All the per-scene tables report single PSNR/SSIM/LPIPS numbers with no error bars or seed variance, so it’s unclear whether gains are stable across runs or due to initialization. Given the small rendering quality gap compared with the baselines, it is necessary to have the error bars (at least with several examples) to understand the significance of the improvements.
* Most results emphasize vanilla 3DGS and 3DGS-MCMC setups on standard static benchmarks; there’s little evidence for large-scale indoor scans, or outdoor long-range sequences where visibility patterns and sparsity differ. A dedicated failure case section will help to further study the method.

**Questions:**

I am having a hard time connecting the cits (e.g., GSs, MCs, Ours) when reading the paper. Primarily, I need to frequently read the Appendix and identify the differences in the configuration. It would be better to provide a clearer explanation or include the difference in the table consistently.

---

> ### Author Response · Authors · 2025-11-21
>
> Thanks for your valuable comments. Below we provide our responses, and we will incorporate the relevant parts into the manuscript. **The revised version has been updated; we mark the newly added information with blue text**.
>
> **For weakness 1: (The corresponding experiments and analyses have been added in the our paper.)**
>
> The core motivation of our work is to decouple the 3DGS optimization, characterize the behavior of each component, and remove unnecessary elements from the optimization process. Building on these investigations, we propose AdamW-GS, which simultaneously accelerates training, substantially reduces redundant primitives, and improves reconstruction quality. **We argue that fair discussion and comparison should be based on making principled use of these findings and evaluating, across diverse baselines, the original optimizer and our AdamW-GS**. In fact, the comparisons between the original optimizer and the proposed AdamW-GS across different baselines are inheriently meaning under the same limited upper bound on the number of primitives, and our method typically uses fewer primitives.
>
> We additionally provide experiments on **MaskGaussian[1]**, **Taming-3DGS[2]** (both the final count mode with a larger budget and the multiplier mode with a smaller budget as specified in the official script), and **Deformable Beta Splatting (DBS)[3]**, each trained with either the original optimizer or our proposed AdamW-GS. Since MaskGaussian already discusses Compact3DGS and validates it as a strong algorithm, we do not include further comparisons to Compact3DGS here.
> - For **MaskGaussian** (see Table 1 below), we argue in Sec.5.1 of our paper that "MaskGaussian suffers from a potential reconstruction quality risk," as it removes 61% of the primitives but degrades PSNR by 0.1 dB. We attribute this to the synchronous updating of mask scores, **which is consistent with Observation 1 in Sec.4.1** showing that synchronous Adam updates lead to more dead primitives. To further validate this hypothesis, we apply AdamW-GS to MaskGaussian. As reported in Table , MaskGaussian with AdamW-GS matches the performance of vanilla 3DGS with AdamW-GS **without incurring the aforementioned reconstruction-quality risk**, and, compared to the original MaskGaussian, even **achieves an additional 7% reduction in primitives on indoor scenes**.
> - For **Taming-3DGS** (both the final count  mode and the multiplier mode, see Table 2 below), our method consistently reconstructs higher-quality scenes more rapidly while using fewer active primitives. In the revised paper, Figure 6 compares the renderings of the original Taming-3DGS and Taming-3DGS with AdamW-GS under both modes, showing that our approach better **preserves fine details**. Leveraging the properties of our method, we further propose an modified variant of Taming-3DGS: since our method can rapidly penalize redundant primitives, we replace the conservative pruning operation in Taming-3DGS with the pruning operation of vanilla 3DGS, yielding Taming-3DGS-p. Comparing the original Taming-3DGS and Taming-3DGS-p with AdamW-GS in the final count  mode, our method not only improves reconstruction quality but also **reduces training time to 40%** and further **lowers memory usage**, with a final $N_P$ of only 2.16 million versus 3.20 million for the original Taming-3DGS.
> - For **DBS** (see Table 3 below), similar to 3DGSMCMC, we evaluate DBS with Adam and DBS with AdamW-GS on MipNeRF-360. Across the 9 MipNeRF-360 scenes, our method **improves reconstruction quality** on eight of them. For Treehill, we observe severe overfitting during training, which prematurely triggers early stopping, and the number of dead primitives rapidly approaches zero, rendering our DAR ineffective. Thus, our method may be less effective in scenarios that suffer from severe overfitting. We will include this case in the failure case section of the revised paper.
>
> In summary, our results show that the proposed method exhibits stronger generalization and achieves better performance under the same budget. We offer new insights into the 3DGS optimization process, and we believe our work will help advance the development of 3DGS optimization.
>
> [1] Maskgaussian: Adaptive 3d  gaussian representation from probabilistic masks.
> [2] Taming 3dgs: High-quality radiance fields with limited resources.
> [3] Deformable beta splatting.

---

> ### Author Response · Authors · 2025-11-21
>
> **Table 1**: vanilla 3DGS and MasskGaussion w/ or w/o  AdamW-GS
>
> | Pipeline     | AdamW-GS | All    |    |    |   All   | Indoor |  |  | Indoor     | Outdoor |  |  | Outdoor  |
> | ------------ | --------- | ------ | ----- | ------ | -------- | ------ | ------ | ------ | ---------- | ------- | ------- | ------- | -------- |
> |              |           | PSNR$\uparrow$  | SSIM$\uparrow$  | LPIPS$\downarrow$  | $\Delta N_a$     | PSNR$\uparrow$  | SSIM$\uparrow$  | LPIPS$\downarrow$  | $\Delta N_a$       | PSNR$\uparrow$   | SSIM$\uparrow$   | LPIPS$\downarrow$  | $\Delta N_a$      |
> | vanilla 3DGS | x         | 27.506 | 0.815 | 0.216  | (3.098m) | 31.080 | 0.925  | 0.189  | (1.331m)   | 24.648  | 0.728   | 0.239   | (4.512m) |
> |              | $\checkmark$          | 27.678 | 0.822 | 0.220  | -49.3%   | 31.209 | 0.925  | 0.191  | -50.4%     | 24.854  | 0.740   | 0.243   | -48.4%   |
> | MaskGaussian | x         | 27.485 | 0.815 | 0.219  | -53.1%   | 30.988 | 0.924  | 0.192  | -61.5%     | 24.683  | 0.728   | 0.240   | -46.4%   |
> |              | $\checkmark$          | 27.721 | 0.821 | 0.221  | -57.2%   | 31.199 | 0.925  | 0.193  | **-68.6%** | 24.939  | 0.739   | 0.244   | -48.1%   |
>
> **Table 2**: Taming 3DGS w/ or w/o AdamW-GS (numbers are reported in million, Time are reported in mins.)
>
> | Pipeline                    | AdamW-GS | All    |   |   |       |       | All       | Indoor |  |  | | Indoor | Outdoor |  | |  | Outdoor |
> | --------------------------- | --------- | ------ | ----- | ------ | --------- | --------- | --------- | ------ | ------ | ------ | ------ | ------ | ------- | ------- | ------- | ------- | ------- |
> |                             |           | PSNR$\uparrow$   | SSIM$\uparrow$ | LPIPS$\downarrow$  | $N_p$       | $N_a$       | Time      | PSNR$\uparrow$  | SSIM$\uparrow$  | LPIPS$\downarrow$  | $N_p$     | $N_a$     | PSNR$\uparrow$   | SSIM$\uparrow$   | LPIPS$\downarrow$   | $N_p$      | $N_a$     |
> | Taming-3DGS (multiplier)    | x         | 27.386 | 0.796 | 0.258  | 0.668     | 0.620     | 7.44      | 31.025 | 0.918  | 0.205  | 0.357  | 0.329  | 24.479  | 0.698   | 0.299   | 0.916   | 0.853   |
> |                             | $\checkmark$          | 27.537 | 0.799 | 0.254  | 0.646     | 0.575     | 5.27      | 31.268 | 0.920  | 0.200  | 0.357  | 0.313  | 24.552  | 0.703   | 0.297   | 0.877   | 0.785   |
> | Taming-3DGS (final count)   | x         | 27.912 | 0.822 | 0.207  | 3.205     | 2.609     | 20.30     | 31.603 | 0.928  | 0.181  | 1.377  | 1.144  | 25.949  | 0.736   | 0.228   | 4.667   | 3.776   |
> |                             | $\checkmark$          | 28.034 | 0.826 | 0.207  | 3.109     | **1.847** | **10.46** | 31.720 | 0.928  | 0.180  | 1.408  | 0.759  | 25.805  | 0.744   | 0.229   | 4.469   | 2.717   |
> | Taming-3DGS-p (final count) |$\checkmark$          | 28.038 | 0.828 | 0.205  | **2.160** | **2.160** | **8.44**  | 31.724 | 0.930  | 0.177  | 0.804  | 0.804  | 25.089  | 0.745   | 0.227   | 3.346   | 3.246   |
>
>
> **Table 3**: DBS w/ or w/o AdamW-GS (8 scenes here)
>
> | AdamW-GS | All    |   | All    | Indoor |  | Indoor | Outdoor |  | Outdoor |
> | --------- | ------ | ----- | ------ | ------ | ------ | ------ | ------- | ------- | ------- |
> |           | PSNR$\uparrow$  | SSIM$\uparrow$ | LPIPS$\downarrow$ | PSNR$\uparrow$  | SSIM$\uparrow$  | LPIPS$\downarrow$  | PSNR$\uparrow$   | SSIM$\uparrow$   | LPIPS$\downarrow$   |
> | x        | 29.362 | 0.864 | 0.165  | 32.696 | 0.940  | 0.143  | 26.029  | 0.787   | 0.187   |
> | $\checkmark$        | 29.643 | 0.871 | 0.158  | 33.178 | 0.945  | 0.140  | 26.108  | 0.796   | 0.175   |
>
>
> **Table 4**. 3DGSMCMC or 3DGS on OMMO  (numbers are reported in million)
>
> | AdamW-GS | 3DGSMCMC: |       |        |       |       | 3DGS:   |       |        |       |       |
> | --------- | -------- | ----- | ------ | ----- | ----- | ------ | ----- | ------ | ----- | ----- |
> |           | PSNR$\uparrow$    | SSIM$\uparrow$ | LPIPS$\downarrow$ | $N_p$    | $N_a$    | PSNR$\uparrow$  | SSIM$\uparrow$ | LPIPS$\downarrow$ | $N_p$    | $N_a$    |
> | x         | 30.359   | 0.925 | 0.135  | 1.960 | 1.673 | 30.040 | 0.914 | 0.154  | 1.878 | 1.640 |
> | $\checkmark$         | 30.716   | 0.930 | 0.126  | 1.960 | 1.765 | 30.351 | 0.914 | 0.154  | 1.245 | 1.211 |

---

> ### Author Response · Authors · 2025-11-21
>
> **For weakness 2**:
>
> First, we strictly enforce identical initialization conditions. To further assess the stability of the method: (1) we additionally evaluate three different baselines (MaskGaussion, Taming-3DGS, DBS), as well as the new dataset introduced below, which further corroborates the stability of our method; and (2) we report per-scene results with repeated experiments, as shown in Table 5 and 6 below, where the error is represented as $\pm$ std. The plot has been added in the revised manuscript.
>
>
> **For weakness 3**:
>
> We provide results on the OMMO [4], which contains large-scale outdoor scenes with **long-range sequences**. Our data processing follows the settings described in [5] and its associated Github repository. Except for scenario \#10, where data processing is incompatible, we provide average values to prove that **our method is the better**. The results are summarized in Table 4 which can be found above.
> **We add the failure case section in the revised manuscript.**
>
> [4] A large-scale outdoor multi-modal dataset and benchmark for novel view synthesis and implicit scene reconstruction.
>
> [5]3d gaussian splatting as markov chain  monte carlo.
>
> **For Question**:
>
> We thank the reviewer for the valuable comments. We **add further clarifications** to the relevant tables and main text to **improve the readability of the paper**.
>
> - - -
>
> **Table 5**. vanilla 3DGS with AdamW-GS per scene
>
> | scenes | bicycle           | flowers           | garden             | stump             | treehill          | room              | counter           | kitchen            | bosai             |
> | ------ | ----------------- | ----------------- | ------------------ | ----------------- | ----------------- | ----------------- | ----------------- | ------------------ | ----------------- |
> | PSNR   | 25.451$\pm$ 0.009 | 21.718$\pm$ 0.005 | 27.253 $\pm$ 0.081 | 27.029$\pm$ 0.031 | 22.876$\pm$ 0.015 | 31.845$\pm$ 0.082 | 29.092$\pm$ 0.033 | 31.734 $\pm$ 0.029 | 32.331$\pm$ 0.050 |
> | SSIM   | 0.0778$\pm$ 0.000 | 0.610 $\pm$ 0.000 | 0.862 $\pm$ 0.000  | 0.800$\pm$ 0.000  | 0.650$\pm$ 0.001  | 0.922$\pm$ 0.000  | 0.907$\pm$ 0.000  | 0.927$\pm$ 0.000   | 0.942$\pm$ 0.000  |
> | LPIPS  | 0.221$\pm$ 0.001  | 0.344$\pm$ 0.000  | 0.124 $\pm$ 0.000  | 0.206$\pm$ 0.000  | 0.326$\pm$ 0.001  | 0.221$\pm$ 0.000  | 0.207$\pm$ 0.001  | 0.132$\pm$ 0.000   | 0.205$\pm$ 0.000  |
>
> **Table 6**. 3DGS-MCMC with AdamW-GS per scene
>
> | scenes | bicycle            | flowers           | garden            | stump             | treehill          | room              | counter           | kitchen           | bosai             |
> | ------ | ------------------ | ----------------- | ----------------- | ----------------- | ----------------- | ----------------- | ----------------- | ----------------- | ----------------- |
> | PSNR   | 25.869 $\pm$ 0.004 | 22.076$\pm$ 0.035 | 28.152$\pm$ 0.011 | 27.244$\pm$ 0.012 | 22.854$\pm$ 0.017 | 32.653$\pm$ 0.023 | 29.509$\pm$ 0.013 | 32.542$\pm$ 0.038 | 33.030$\pm$ 0.015 |
> | SSIM   | 0.809$\pm$ 0.000   | 0.658$\pm$ 0.000  | 0.885$\pm$ 0.000  | 0.808$\pm$ 0.000  | 0.662$\pm$ 0.000  | 0.933$\pm$ 0.000  | 0.919$\pm$ 0.000  | 0.937$\pm$ 0.000  | 0.950$\pm$ 0.000  |
> | LPIPS  | 0.157$\pm$ 0.000   | 0.267$\pm$ 0.000  | 0.088$\pm$ 0.000  | 0.174$\pm$ 0.000  | 0.265$\pm$ 0.000  | 0.196$\pm$ 0.000  | 0.183$\pm$ 0.000  | 0.117$\pm$ 0.000  | 0.190$\pm$ 0.000  |

---

> ### Author Response · Authors · 2025-11-27
>
> Dear Reviewer,
>
> I hope this message finds you well. As the discussion period is drawing to a close, I wanted to ensure we have addressed all your concerns satisfactorily. If there are any remaining points you would like us to consider, please let us know at your convenience. Your feedback has been extremely valuable to us, and we would be very happy to provide additional clarification or make further revisions that could help strengthen the manuscript.
>
> Thank you again for your time and effort in reviewing our work.

---

### Author Response · Authors · 2025-11-26
**Summary of Manuscript Revisions**

We sincerely thank the reviewers for their time and valuable comments. In response, we have revised our manuscript, with changes highlighted in **blue**. The main revisions are summarized below:
- **3 Additional pipelines (MaskGaussian, Taming-3DGS, DBS) and 1 Extra dataset (OMMO):**  Additional experiments and analyses for MaskGaussian are reported in the main paper. All these experiments, analyses, brief method descriptions, and visualizations (Figure 6 for Taming-3DGS) is provided in Appendix Sec. E.
- **More Ablation Study**: ablation for RSR can be found Sec. F and the results can be found in Table 11; ablation about regularization with clip can be found in Sec. I.2 and results can be found in Table 12.
- **More Guidance for Hyperparameters:** We provide a more detailed analysis for the StSS and rescaling hyperparameters in RSR in Appendix Sec. J.2.
- **More Exploration Strategies**: We provide the analysis of current methods and 2 feasible extra Exploration Strategies (ABE-split and densification extending) in Appendix Sec. F.
- **Readability of the Paper:** We have added necessary clarifying notes to all cited content (like MC7/MC8) in the main text, appendix, and in the caption of tables as well as figures, and have also improved the presentation of the tables. All major updates have been explicitly highlighted.
- **New Appendix Section G Failure Cases:** We provide visualizations and analyses of the failure cases, as well as the corresponding error bars.
- **Update fot Related Work**: This can be found in Appendix Sec. H More about Related Work.


We once again sincerely appreciate the reviewers' valuable comments and would be pleased to address any further questions or concerns.

---

### Author Response · Authors · 2025-11-30
**Summary of key issues and responds**

We thank the reviewers for their valuable comments. **We have provided detailed, point-by-point responses to all Questions and Weaknesses**; when a Weakness overlaps with a Question, we address it under the corresponding Question. In the _Summary of Manuscript Revisions_, we describe the revisions made to the paper in detail.

Below, **we summarize our main responses to the key issues highlighted by the reviewers**, following the order in which the relevant Questions or Weaknesses appear: (i) **hyperparameter choices** (Reviewer AiQe: Questions 1,2, Weaknesses 1,2; Reviewer odb6: Weakness 1), (ii) **ablation studies** (Reviewer AiQe: Questions 3,4, Weakness 4), (iii) **budget comparisons and additional baselines or datasets** (Reviewer A2qo: Weaknesses 1 and 3; Reviewer odb6: Question 2), and (iv) **readability** (Reviewer A2qo: Question; Reviewer odb6: Weakness).

**(i) hyperparameter**:
1. Although our method introduces additional hyperparameters, **in all experiments across the 5 different pipelines and 4 datasets we only modified the StSS, keeping all others unchanged**. For StSS, we **provide two conservative schedules**. Even with the conservative schedules, our method consistently **outperforms the original pipelines**  (see our response to Reviewer AiQe Question 1).
2. For the StSS in RSR, the rescaling hyperparameters, and the clipping in DAR, we **provide detailed tuning guidelines together with the underlying rationale and supporting evidence** (see our responses to Reviewer odb6 Weakness 1 or Reviewer AiQe Questions 1,2); the corresponding analyses are also summarized in Appendix Sec. J.2.
3. We also **report all related ablation study under different hyperparameters**, demonstrating their effectiveness. Thanks to our decoupling design, our method still outperforms the original approach even when the hyperparameters are not perfectly tuned. All related experiments are provided in Appendix Sec. K.

**(ii) ablation study**:
1. We additionally include **a new experiment of RSR** and **provide a detailed analysis of the exploration** of our current method (see our response to Reviewer AiQe Question 4). Moreover, we **introduce two exploration strategies** that can be directly combined with our method. All related analyses are organized in Appendix Sec. F, with discussion in the main paper.
2. For the ablation on effective magnitude, this clipping is **not effective to the original L1**, since its effective magnitude is inherently distorted by Adam. We **further include three additional experiments showing that clipping is also ineffective for the constant-penalty**. We **discuss the reasons for the failure of these two approaches (L1 and constant-penalty) in Sec. 4.3 and Sec. I, and summarize four benefits of our proposed DAR**. Related explanations can also be found in our response to Reviewer AiQe Question 3.

**(iii) budget comparisons or more baselines or dataset**:
1. We argue that **fair discussion and comparison should be based on making principled use of these findings and evaluating**, across diverse baselines, **the original optimizer and our proposed AdamW-GS**. In fact, the comparisons between the original optimizer and the proposed AdamW-GS across different baselines are inheriently **meaning under the same limited upper bound on the number of primitives**, and **our method typically uses fewer primitives**.
2. We further evaluate our method on **3 additional pipelines and 1 additional dataset**, which together verify the **effectiveness of our approach and the _associated findings_**. All corresponding experiments and analyses have been included in Appendix E, and are also discussed in our responses to Reviewer A2qo Weaknesses 1 and 3.

**(iv) Readability**:
1. We have **added necessary clarifying notes to all cited content** (like MC7/MC8) in the main text, appendix, and in the caption of tables as well as figures, and have also improved the presentation of the tables and figures. All major updates have been explicitly highlighted.

---

### Meta-Review · Area_Chair_hgMq · 2026-01-01

**Summary:**

This paper combines three modules: Sparse Adam, Re-State Regularization, and Decoupled Attribute Regularization to create a new optimization strategy to better solve the 3DGS-MCMC.

All the reviewers share the same concerns: an extensive number of hyperparameters, more ablations, and readability. Most of the concerns are addressed by the authors. While the introduction of new hyperparameters initially raised concerns about complexity, the authors' extensive additional experiments and provided tuning guidelines have mitigated these worries.

However, ensuring the final manuscript is fully readable remains a prerequisite for acceptance.

The AC has a concern about performance gains: while PSNR in 3DGS typically fluctuates by roughly 0.2 dB due to stochasticity, the authors' rebuttal demonstrates remarkably low variance across repeated runs (in Table 5).  The authors should also provide the stability analysis for their baseline methods as well, which is very important but missing in the rebuttal and paper.
This stability is crucial, as it suggests the improvements are a direct result of the proposed optimization method rather than random initialization or noise. Hence, I suggest  other ACs and SAC have a quick check.

**Reviewer Concerns:**

***Hyperparameter Sensitivity and Generalization***: Reviewers AiQe and odb6 expressed concerns regarding the complexity of tuning the new hyperparameters, particularly the milestone-based StSS schedule. The authors addressed this by providing conservative schedules for different pipelines (3DGS and 3DGS-MCMC) and demonstrating that the method is robust even when parameters are not perfectly tuned.

***Ablation and Baselines***. Reviewers requested more rigorous comparisons and a clearer understanding of individual components like Decoupled Attribute Regularization (DAR). In response, the authors added experiments on three additional pipelines (MaskGaussian, Taming-3DGS, and DBS) and a new outdoor dataset (OMMO), showing consistent improvements in reconstruction quality and training speed across various setups.

***Statistical Stability***: Reviewer A2qo noted a lack of error bars and seed variance. The authors revised the manuscript to include per-scene results with standard deviations across repeated experiments, confirming that the performance gains are statistically stable. According to the previous 3DGS papers, usually the variance of PSNR is around 0.2db, which is singificantly higher than the authors provided. Thus, this leaves uncertainty for the AC as well.

Except for the reviews, the AC found the caption in figure 1 is a bit confusing, what is bosai and valune? Are they typos or domain jargons?

**Reviewer Scores:**

This paper received only 3 reviews, and all the reviewers gave 6, however, two of them are very uncertain about their ratings (2).

---

### Decision · Program_Chairs · 2026-01-26

Accept (Poster)